# Electricity-free hydrogen production from the air

Qili Xu[1,2,3,9], Xiaoxue Yao [1,3,9], Hoi Ying Chung[2,9], Xiongyi Liang [4,5,6,9],
Zhi Zhang[1,3], Zhenwen Zhang [1,3], Wai Kin Lo [1,3], Yijun Zeng [1,3],
Xiao Cheng Zeng [4,7] ✉, Yun Hau Ng [2,8] ✉ & Steven Wang [1,3] ✉

The ever-growing demand for electricity and clean water restricts widespread application of hydrogen production via water electrolysis or photocatalytic water splitting. Here, we present a self-sufficient electricity-free air-to-hydrogen system that integrates radiative cooling-enhanced water adsorption with synergistic photocatalysis and photothermal conversion by fabricating spectral selective absorbing/emitting hygroscopic hydrogen evolution nano-fiber membranes to harvest atmospheric moisture and produce hydrogen. Leveraging the nocturnal radiative cooling effect, we expand the operational relative humidity range of nanofiber membranes and enhance both water collection capacity and kinetics. The collected water undergoes efficient gas-phase water splitting for $H_2$ production during the day through photothermal catalytic processes without electrical and liquid water assistance. The hydrogen production rate of the scale-up air-to-$H_2$ system under outdoor natural light reaches 6467.55 $\mu mol \cdot m^{-2} \cdot h^{-1}$. Extrapolating this experimentally validated rate to land-based deployment demonstrates the potential for large-scale hydrogen generation, with practical feasibility dependent on regional humidity and solar conditions. Thus, our approach cost-effectively addresses green-$H_2$ scarcity without demanding natural freshwater and electricity, thereby providing an archetype for global sustainable development.

Producing hydrogen ($H_2$) fuel from sunlight and water through photocatalytic (PC) water splitting offers a promising pathway to carbon neutrality[1–4]. However, this process necessitates a continuous supply of clean freshwater, as impurities and contaminants can compromise the stability and longevity of photocatalytic systems[5,6]. With over 4 billion people facing freshwater scarcity[7,8], relying solely on photocatalytic $H_2$ production fails to adequately address the intertwined issues of energy and water[9]. While numerous studies have explored the use of seawater as a water source for PC $H_2$ production[10–12], the ions present in seawater can be absorbed onto the catalyst surface, potentially affecting its stability or generating undesirable byproducts[13–15]. Additionally, this approach is challenging to implement in inland areas that are far from coastlines. Therefore, there is an urgent need to identify alternative water sources for $H_2$ production that do not compromise clean water supplies and have fewer geographical restrictions.

[1]Department of Mechanical Engineering, City University of Hong Kong, Hong Kong, China. [2]School of Energy and Environment, City University of Hong Kong, Hong Kong, China. [3]Centre for Nature-inspiring Engineering, City University of Hong Kong, Hong Kong, China. [4]Department of Materials Science & Engineering, City University of Hong Kong, Hong Kong, China. [5]Shenzhen Research Institute, City University of Hong Kong, Shenzhen, China. [6]Chengdu Research Institute, City University of Hong Kong, Chengdu, China. [7]Hong Kong Institute for Clean Energy, City University of Hong Kong, Hong Kong, China. [8]Center for Renewable Energy and Storage Technologies (CREST), Clean Energy Research Platform (CERP), Chemical Engineering Program, Physical Science and Engineering Division, King Abdullah University of Science and Technology, Thuwal, Saudi Arabia. [9]These authors contributed equally: Qili Xu, Xiaoxue Yao, Hoi Ying Chung, Xiongyi Liang. ✉e-mail: xzeng26@cityu.edu.hk; yunhau.ng@kaust.edu.sa; Steven.wang@cityu.edu.hk

Atmospheric humidity, which contains approximately 13 trillion tons of water[16], represents one of the most promising sources for clean water[17–19]. Adsorption-based atmospheric water harvesting (AWH) technologies utilize hygroscopic materials such as metal-organic frameworks (MOFs)[19–22] to effectively capture moisture from the air. By supplying heat energy, the adsorbed water can subsequently be desorbed to obtain clean water. Therefore, the alternating adsorption-desorption process of MOFs can provide a reliable source of clean water for PC $H_2$ production.

Furthermore, most PC systems exhibit limited solar spectrum utilization for $H_2$ production[3]. Only photons with energy exceeding the semiconductor bandgap can be harnessed. This limitation means that only ultraviolet (UV) light and a small portion of visible (Vis) light are used, while the majority of solar energy is wasted as heat[9,23,24]. Additionally, many PC systems rely on liquid water as the reaction medium, which presents several challenges. For instance, catalyst components may leach into the water, and liquid water can hinder the absorption of solar radiation[12,25,26]. To address these issues, the heat generated during the photocatalytic process can be harnessed to drive the desorption of AWH, releasing water vapor. Subsequently, gas-phase water vapor can serve as the feedstock for PC reactions. This approach not only enhances solar spectrum utilization but also alleviates the limitations associated with clean liquid-phase water, improving reaction rates and long-term stability.

This study introduces an air-to-$H_2$ system that employs multilayer hygroscopic hydrogen evolution (HHE) nanofiber membranes (NFMs) for efficient green $H_2$ production, with clean water generated as a byproduct (Fig. 1a). Fabricated via electrospinning, the HHE NFMs consist of a photothermal catalytic (PTC) top layer of Pt/TiO$_2$@PVDF and a porous hygroscopic bottom layer of MIL-101(Cr)@PAN (Fig. 1b). Capitalizing on the high solar absorbance and high infrared emissivity of the PTC layer (Fig. 1c), the system achieves radiative cooling-enhanced atmospheric water harvesting (AWH) at night, and

photocatalytic-driven $H_2$ production during the day by decomposing the water vapor released from the hygroscopic layer. Specifically, the distribution of MIL-101(Cr) within the highly porous hygroscopic layer facilitates rapid, spontaneous moisture adsorption from ambient air at night while simultaneously mitigating particle aggregation. Concurrently, the radiative cooling effect of the PTC layer elevates the local humidity, thereby accelerating moisture capture and extending the effective hygroscopic range (Fig. 1d). Under daytime solar irradiation, visible and near-infrared (NIR) wavelengths are converted into thermal energy to desorb water vapor from the hygroscopic layer, while ultraviolet (UV) light is absorbed to activate Pt/TiO$_2$ for the photocatalytic (PC) production of $H_2$. These advantages are substantiated through experimental and theoretical validation. A large-scale prototype system was demonstrated in outdoor operations over 14 days, yielding a maximum $H_2$ production rate of 764.2 mL·m$^{-2}$ under 6 h of natural sunlight. This approach enables the direct, passive production of green $H_2$ from air, entirely independent of liquid water input, thereby offering a promising pathway toward a sustainable future.

## Results and discussion

The adsorption capacity of the hygroscopic layer and the range of humidity it can adsorb are crucial for the performance of HHE system. Generally, the moisture adsorption kinetics of the hygroscopic layer depend on the diffusion of water vapor between and within the MIL-101(Cr) particles embedded in the nanofiber. A porous, flexible green hygroscopic layer is prepared by using a co-spinning method with MIL-101(Cr) and PAN (Figs. 2a and S1). The X-ray diffraction (XRD) patterns clearly indicate that the diffraction peaks of the synthesized MIL@PAN NFMs align well with those of the MIL-101(Cr) crystals, confirming the presence of the MOF in the hygroscopic layer (Fig. S2). Scanning electron microscopy (SEM) images show that the porous hygroscopic layer exhibits a honeycomb-like structure and the merged nanofibers

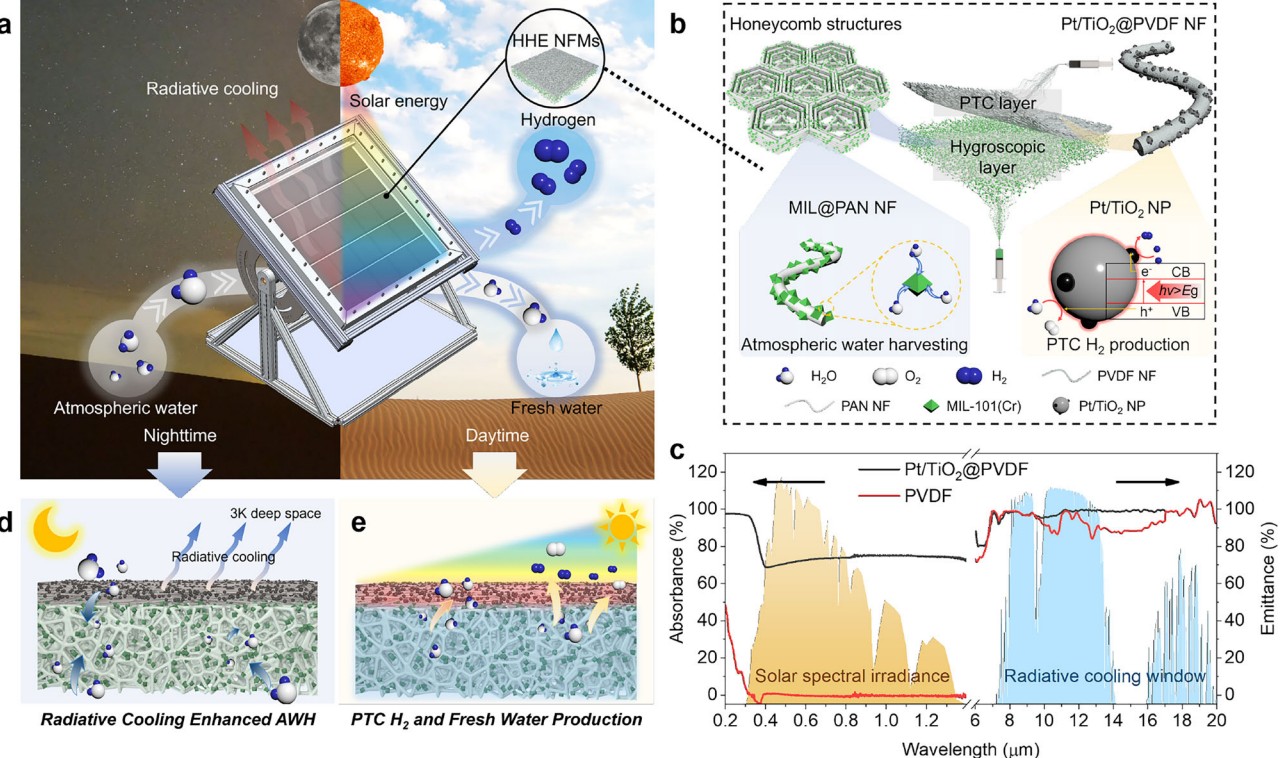

**Fig. 1 | Overview of the HHE system. a** Schematic of HHE system. **b** Schematic structures of HHE NFMs. The Pt/TiO$_2$@PVDF NFMs were directly electrospun on the MIL@PAN NFMs. **c** Absorption spectrum in solar spectral irradiance and emission

spectrum in radiative cooling window of Pt/TiO$_2$@PVDF NFMs and pure PVDF NFMs. **d** The working principle of radiative cooling enhanced AWH. **e** The working principle of PTC $H_2$ and fresh water production.

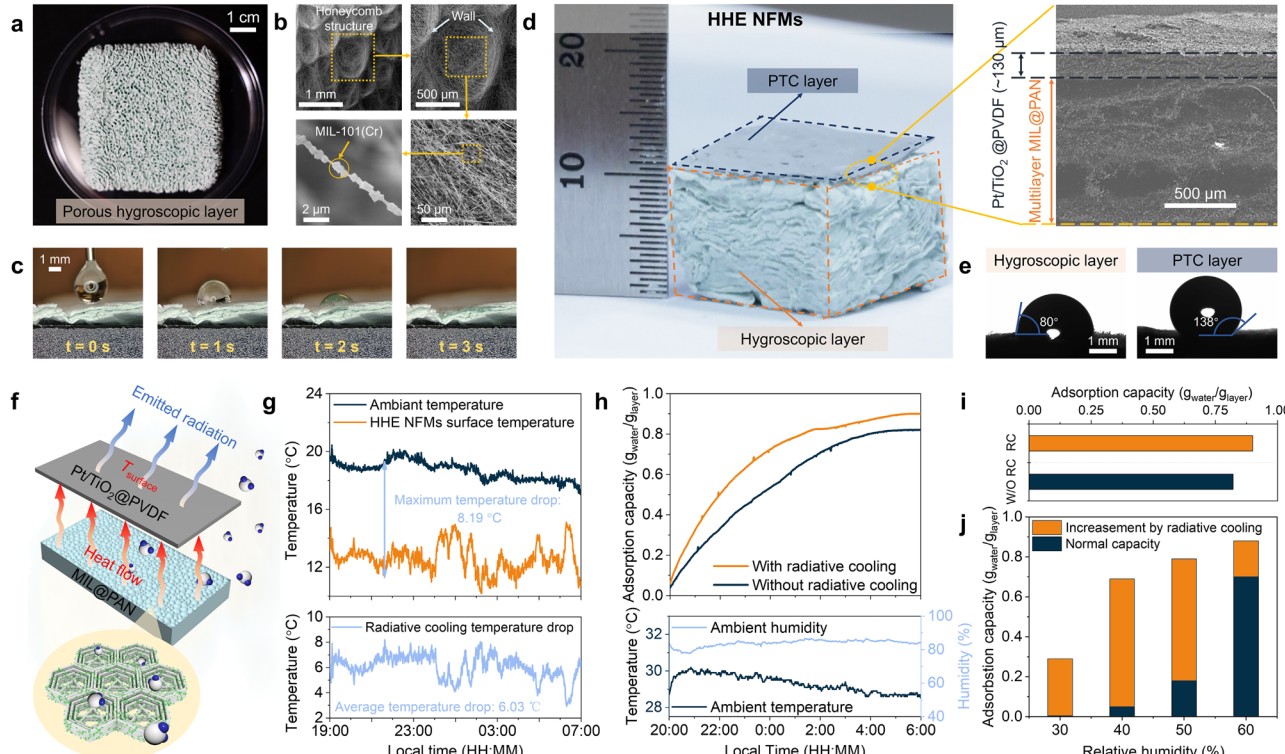

**Fig. 2 | Radiative cooling enhanced atmospheric water harvesting. a** Optical image of porous hygroscopic layer. **b** SEM images at increasing magnifications demonstrating the honeycomb structure of MIL@PAN NFMs. **c** The water adsorption process of porous hygroscopic layer. **d** Optical image of HHE NFMs and its cross-sectional SEM image. **e** Contact angle measurement at various locations of the HHE NFMs: the top surface of the hygroscopic layer (left) and PTC layer (right). **f** Schematic diagram of radiative cooling-enhanced AWH. **g** Real-time temperature and temperature difference between the surface of Pt/TiO$_2$@PVDF NFMs and the ambient during the outdoor experiment. **h** Water uptake capacity comparison of HHE NFMs with/without radiative cooling effect as a function of time during the outdoor test. **i** The maximum water uptake comparison of HHE NFMs with/without radiative cooling during the outdoor test. **j** The theoretical enhancement of water adsorption capacity due to the effects of radiation cooling at an ambient temperature of 25 °C during the indoor test.

form the walls of the honeycomb (Figs. 2b and S3), which results in a high porosity of 0.93 (Fig. S4) and therefore facilitates moisture diffusion[27,28]. The uniformly distributed MIL-101(Cr) particles (Fig. S5) along the nanofibers impart a hierarchical roughness and nanoporous structure to the MIL@PAN NFMs (Fig. 2b). N$_2$ adsorption-desorption measurements (Fig. S6) reveal that the adsorption isotherms exhibit typical Type I characteristics with a notable hysteresis loop, confirming the presence of a significant number of micropores in the MIL@PAN NFMs. The measured pore size distribution indicates that the micropore sizes are centered around 1.479 nm, which is larger than the kinetic diameter of water molecules (0.27–0.32 nm) (Fig. S7), allowing for the adsorption of a large amount of water molecules[29]. Material hydrophilicity (Fig. 2e, left) and capillary effect of porous NFMs promotes rapid moisture contact and penetration. As shown in Fig. 2c, the droplet is quickly absorbed and penetrates both horizontally and vertically, confirming the great water adsorption capability of the hygroscopic layer.

To evaluate the moisture adsorption performance of the hygroscopic layer, the moisture adsorption process was assessed gravimetrically at 25 °C under various humidity levels (Fig. S8). Experimental results show that the dry NFMs exhibit water adsorption capacities of 0.698, 0.765, 0.821, and 0.872 g$_{water}^{-1}$ · g$_{layer}^{-1}$ at 60, 70, 80, and 90% relative humidity (RH), respectively. Furthermore, this hygroscopic layer can rapidly adsorb moisture from the air within the first hour (Fig. S8B). Additionally, this layer demonstrates great cycling stability, retaining its adsorption performance even after multiple cycling tests (Fig. S9). However, as the air humidity decreases, the moisture absorption of the hygroscopic layer drops rapidly. At 50% RH, the absorption capacity is about

0.179 g$_{water}^{-1}$ · g$_{layer}^{-1}$, limiting the applicability of this hygroscopic film under low humidity conditions.

Therefore, to broaden the humidity range for practical applications of the hygroscopic layer and accelerate the moisture absorption process, we further positioned the high infrared emissivity PTC layer on the top layer of the hygroscopic layer (Figs. 2d and S10). The thickness of the porous PTC layer is about 130 μm, which allows the flow of moisture. Besides, as shown in Fig. 2e, the hydrophobic PTC layer and the hydrophilic hygroscopic layer form a Janus structure, which enhances the transfer of moisture. Meanwhile, due to the inherent high emissivity of PVDF, the PTC layer exhibit great emissivity of 0.95 in the atmospheric transparency window (8–13 μm) (Fig. 1c). Essentially, the adsorption-based AWH converts atmospheric moisture into liquid water for storage through phase change, releasing heat to the surroundings. By utilizing the radiative cooling effect of PTC layer, the long-wavelength (infrared, IR) thermal radiation can be emitted from the surface of HHE NFMs to the cold sky to cool it below the ambient temperature during the night (Fig. 2f). Given the constant absolute humidity of the environment, this cooling effect increases the effective RH near the hygroscopic layer[30–32].

The practical radiative cooling effect was tested by recording the surface temperature of Pt/TiO$_2$@PVDF NFMs and ambient temperature on the outdoor roof in real time via thermocouples (Fig. S11). As shown in Fig. 2g, we initiated the test at around 19:00 h local time, with an average nighttime ambient temperature of around 18.6 °C and an average humidity of approximately 50% RH. The PTC layer was positioned to face the clear sky to enable passive radiative cooling, reducing the surface temperature below its ambient. Temperature drops of ~6 °C were consistently observed, with a maximum temperature drop

of 8.19 °C, confirming the good practical radiative cooling property of the PTC layer. Experiments were also conducted to characterize the moisture adsorption performance of the radiative cooling enhanced system in real-world scenarios (Fig. S12). Figure 2h shows a comparison of the water adsorption capacities between the samples with radiative cooling enhancement and those without. Due to the increase in effective RH from the radiative cooling effect, the sample with radiative cooling enhancement exhibits faster adsorption rates and greater total adsorption capacity. At an average humidity of 80% RH, the radiative cooling enhanced sample achieved a water adsorption capacity of 0.90 $g_{water} \cdot g_{layer}^{-1}$ (Fig. 2i), demonstrating nearly 10% improvement in performance compared to their non-radiative cooling enhanced HHE NFMs counterparts ($0.82 g_{water}^{-1} \cdot g_{layer}^{-1}$).

To further quantify the impact of radiative cooling on effective RH under lower humidity conditions, we calculated the effective RH increased by the radiative cooling effect near the surface of the PTC layer under environmental humidity of 50% RH at various temperature (see the "Methods" section and Fig. S13). The results indicate that with a temperature drop of 6.01 °C, the average effective humidity increased by approximately 40%, rising from 50% RH to about 70% RH (Fig. S13). Controlled experiments were also conducted in an environmental chamber. The effective relative humidity after cooling was set as the adsorption environment, and we tested the theoretical water absorption capacity of the radiative cooling-enhanced HHE NFMs under low humidity conditions (30–60% RH). These specific indoor conditions can indirectly simulate and reflect the water collection performance of radiative cooling in outdoor environments. As shown in Fig. 2j, the adsorption range of the hygroscopic layer is significantly broadened. Initially, it is nearly impossible to adsorb moisture at 30% RH, but after radiative cooling enhancement, it can achieve an adsorption capacity of 0.29 $g_{water} \cdot g_{layer}^{-1}$. At 40, 50, and 60% RH under 25 °C, the theoretical adsorption capacities increased by 0.64, 0.61, and 0.18 $g_{water} \cdot g_{layer}^{-1}$, respectively. This indicates that the radiative cooling-enhanced HHE NFMs possess a broader range of moisture adsorption and greater moisture adsorption capacity.

To further achieve photothermal conversion for water vapor release and PC $H_2$ production, Platinum-loaded titanium dioxide (Pt/ $TiO_2$) has been chosen as a PTC material due to its simple synthesis method, good light absorption, superior charge separation, effective photothermal conversion, great catalytic activity, and good chemical stability[33,34]. Pt/$TiO_2$ powder was synthesized by depositing Pt nanoparticles (1 wt%) onto commercial $TiO_2$ using a photodeposition method[33] (see the "Methods" section, and the optimal pt load has been verified as shown in Fig. S14). Figs. S15 and S16 display the SEM and TEM images, along with the XRD pattern of Pt/$TiO_2$ (Fig. S17), confirming the successful synthesis of Pt/$TiO_2$. These catalyst nanoparticles were then co-spun with PVDF to produce Pt/$TiO_2$@PVDF NFMs, which also serves as the PTC layer continuously deposited above the hygroscopic layer. SEM and TEM images of the PTC layer (Figs. S18 and S19) reveal that the interlaced Pt/$TiO_2$@PVDF nanofibers are arranged in layers, creating a porous structure that facilitates vapor escape and provides abundant active sites for vapor decomposition. Comparisons between Pt/$TiO_2$@PVDF nanofibers and pure PVDF nanofibers (Fig. S20) indicate that the surface of the Pt/$TiO_2$@PVDF nanofibers is significantly rougher, due to the uniform distribution of catalyst particles on the surface of PVDF nanofibers. Thus, the PTC layer exhibits high solar absorption with an average absorption rate of 76.6% (Fig. 1c). This enhancement is attributed to the use of high solar absorption Pt/$TiO_2$ nanoparticles and the rough surface of the Pt/ $TiO_2$@PVDF nanofibers, as the rough surface significantly increases the scattering of incident light within the PTC layer, thereby improving solar radiation absorption.

The indoor $H_2$ production tests were first studied in a glass reactor under AM1.5 G (1000 $W \cdot m^{-2}$) irradiation for 6 h with $H_2$ quantification by gas chromatography (GC) (Fig. S21). The vapor generation rate was measured by placing the 2 ×2 cm HHE NFMs on an electronic balance, while the surface temperature of the HHE NFMs was recorded using an infrared radiation meter. Prior to the experiment, the HHE NFMs had already undergone saturated adsorption in a 90% RH environment. The optimal catalyst loading mass of PTC layer and the optimal thickness of hygroscopic layer were found to be 4.0 $mg \cdot cm^{-2}$ and 11 mm, respectively (Fig. S22). With this loading, the temperature on the upper surface of the HHE NFMs increased rapidly from room temperature to 58.0 °C within 10 minutes, subsequently maintaining above 68 °C (Fig. 3a). Heat is then conducted downward, raising the temperature of the hygroscopic layer (Fig. 3b). The water adsorbed in the hygroscopic layer was released by nearly 80% (approximately 120 mg) within 1 h, with a vapor generation rate reaching 0.298 $kg \cdot m^{-2} \cdot h^{-1}$(Fig. 3c). Additionally, 3D models of the HHE NFMs were established based on finite element analysis to simulate the vapor generation process under natural light, and the results align well with the experimental findings (Fig. 3c).

As the Pt/$TiO_2$ nanoparticles fixed on the PVDF NFMs are in intimate contact with the vapor, photogenerated charge carriers participate in the $H_2$ evolution reaction at the active sites of the catalyst under sunlight (Fig. 3d). Since only the solid-phase catalyst and the gas-phase vapor and $H_2$ participate in the reactions within the HHE system, this process can be classified as a biphasic reaction. Under the optimal catalyst loading mass and optimal thickness of HHE NFMs, the maximum $H_2$ production rate of the HHE NFMs during the first hour reaches 2035 $\mu mol \cdot m^{-2} \cdot h^{-1}$ in indoor test (Fig. 3f), which is about four times the $H_2$ production rate (565.7 $\mu mol \cdot m^{-2} \cdot h^{-1}$) under triphasic conditions (liquid-phase water, solid-phase Pt/$TiO_2$ nanoparticles, and gas-phase $H_2$) (Fig. 3f). To further verify that the $H_2$ production performance in a biphasic system surpasses that in a triphasic system, a vapor feeding reaction condition was established (refer to the Method section and Fig. S23A). This setup reliably maintained the relative humidity inside the reactor above 80% RH (as confirmed in Fig. S23B). The results show that the $H_2$ production rate of biphasic system with vapor feeding is about 2543.2 $\mu mol \cdot m^{-2} \cdot h^{-1}$ during the first hour of reaction, which is higher than HHE NFMs biphasic system (2035.2 $\mu mol \cdot m^{-2} \cdot h^{-1}$), as shown in Fig. S24A. However, both of these two biphasic systems show four 4-4.5 times higher than that of the triphasic system. However, the $H_2$ production rate of HHE NFMs biphasic system decreases because of the gradual release of water vapor from hygroscopic layer. And the $H_2$ production rate of biphasic system with vapor feeding keeps nearly constant $H_2$ production rate (Fig. S24B). Therefore, after 4 h of sunlight illumination, the average $H_2$ production rate of biphasic system with vapor feeding is about twice that of HHE NFMs biphasic system under indoor condition. This enhancement can be attributed to the continuous vapor supply in the biphasic system with vapor feeding, which ensures stable $H_2$ production performance, further validating the advantages of biphasic systems over triphasic systems.

$H_2$ production tests of HHE NFMs under different humidity conditions were also simultaneously conducted in indoor environments. As humidity increased, the enhanced moisture adsorption by the HHE NFMs led to a corresponding improvement in $H_2$ production (as shown in Fig. S25). When the HHE NFMs contained no adsorbed water, no $H_2$ was detected in the reactor, confirming that the $H_2$ originated from the decomposition of adsorbed water. To further verify the source of $H_2$, the source of hydrogen protons in the reactants was qualitatively investigated by using $D_2O$ as an isotope tracer instead of $H_2O$. The experimental results show that $D_2$ (m/z = 4) signal is observed after illumination in HHE NFMs which adsorbed $D_2O$ (Fig. S26A). However, only $H_2$ (m/z = 2) signal is found after illumination in HHE NFMs which adsorbed $H_2O$ (Fig. S26B). In addition, $O_2$ was observed after the reaction (Fig. S27), and the isotope detection of $H_2^{18}O$ also found that the reactor produced $^{18}O_2$ (Fig. S28), verifying that both $H_2$ and $O_2$ came from adsorbed water from the air.

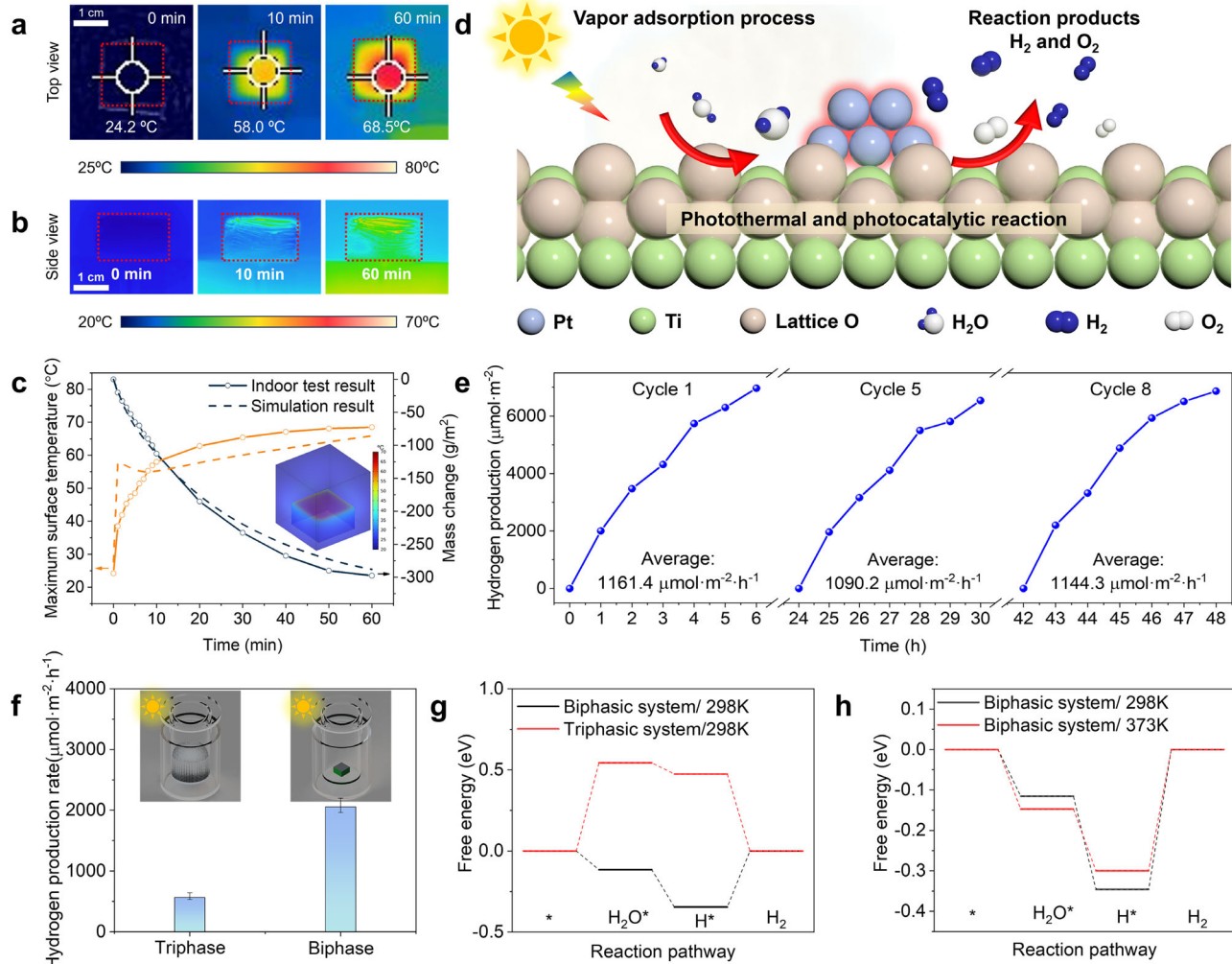

**Fig. 3 | Photothermal catalytic hydrogen production. a, b** IR thermal images revealing the top and side surface temperature of HHE NFMs under 1sun illumination after a fixed time. **c** Surface temperature and mass change of 4 cm² HHE NFMs evolution as a function of time as well as their comparison with the simulation results. Inset figure: Temperature distribution cloud map of a 4 cm² HHE NFM at 60 min. **d** Schematic diagram of PTC vapor decomposition. **e** Time course of H₂ during the indoor cyclic testing of HHE NFMs. **f** H₂ production rate comparison of triphasic and biphasic reaction system in the first hour. Error bars correspond to the standard deviation of three independent measurements. **g** Gibbs energy of a photocatalytic reaction in the triphasic system (298 K) in comparison with the biphasic system (298 K) over the Pt/TiO₂ surface. **h** Gibbs energy of a photocatalytic reaction in the biphasic system with different temperatures over Pt/TiO₂ surface.

To confirm the synergistic effect between photocatalytic and photothermal processes in the HHE system, we measured H₂ production under different light conditions (Fig. S29). The results show that although no H₂ was detected under visible and infrared light, significant liquid water condensed on the reactor walls (Fig. S30). Meanwhile, under UV light, the H₂ production rate was only about 457.7 μmol·m⁻²·h⁻¹ in the first hour, which is much lower than under AM 1.5 G illumination. This supports our design concept that the utilization of full spectrum significantly enhances the overall H₂ production in the HHE system. Subsequent repeatability tests on the HHE NFMs show a stable H₂ production performance, with no noticeable decline observed (Fig. S31). To validate the versatility of the HHE system, we also used Au/TiO₂ catalysts co-spun with PVDF as the PTC layer. While the average H₂ production rate over 6 h is lower than using Pt/TiO₂@PVDF NFMs, it still surpasses the H₂ production rate of the Au/TiO₂ triphasic reaction system (Fig. S32).

The great H₂ production performance of our HHE biphasic system compared to the triphasic reaction system can be attributed to the effects of water state and temperature. These effects can be analyzed from both thermodynamic and kinetic perspectives, involving three key steps: the adsorption of water molecules, the adsorption of

hydrogen atoms, and the PC H₂ evolution reaction. In order to investigate the thermodynamic mechanism of effects of the state of water and temperature, the Gibbs adsorption free energies of two main reaction steps, including the adsorption of water molecule and a hydrogen atom in different states of water and temperatures were calculated by using density functional theory (DFT) method. The calculated adsorption energies of water molecule and hydrogen atom of biphasic system both are obviously lower compared to those of triphasic system under the same temperature (298 K) (Figs. 3g and S33, S34). The potential-determining step (PDS) for the biphasic system is the formation of H₂ gas with an energy change of 0.35 eV, where the PDS turns into the adsorption of H₂O with an uphill barrier of 0.54 eV in the case of triphasic system. This suggests that gas-phase water can lower the thermodynamic barrier in PC reaction process, which agrees well with experimental results (Figs. 3f and S35). On the other hand, the temperature can also affect the Gibbs adsorption free energies. Consistent with experimental results, the higher reaction temperature in the biphasic system leads to a lower barrier for the H₂ formation process, thus accelerating the PC H₂ evolution. Therefore, gas-phase water and elevated temperature can modulate the adsorption of reaction intermediate with an optimal bonding strength, ultimately

boosting the intrinsic catalytic performance. Besides, the temperature increase induced by photothermal effects further accelerates electron transport between Pt and TiO$_2$ and within Pt itself, thereby reducing the recombination of photogenerated carriers and improving the overall photocatalytic H$_2$ evolution efficiency[35–37].

Additionally, from a kinetic perspective, the diffusion coefficient of H$_2$ in the gas phase is significantly greater than in the liquid phase[11,12]. Consequently, during the release process, H$_2$ encounters considerable resistance when crossing liquid water, making it difficult to detach from the photocatalyst surface. In contrast, in the biphasic system, H$_2$ encounters less resistance while diffusing through the gas phase, facilitating its release and collection. As temperature increases, H$_2$ transport resistance decreases, allowing it to desorb more easily from catalytic active sites. This difference further enhances the H$_2$ production efficiency of the biphasic system.

Scaling up the HHE system is essential for practical applications. The size of the HHE NFMs was sequentially scaled up by 5 times and then by 25 times, from 0.0004 m$^2$ to 0.002 m$^2$, and then to 0.01 m$^2$. Then, HHE NFMs of different sizes were arranged in an array within the reactors (Figs. 4a and S36, S37). Under three typical weather conditions (namely sunny, partially cloudy, and overcast), the scaled-up HHE reactor was exposed to natural light to produce H$_2$, with a weather station (Fig. S38) recording temperature, light intensity, and humidity in real-time (Specific definition of different weather condition can be founded in Note S1). In other large-scale photocatalytic reactions, increasing the size typically results in a decrease in reaction rates when conducted under outdoor natural light[5,9,10,12]. However, it is noteworthy that our HHE system demonstrated a substantial increase in H$_2$ production rates under large-scale outdoor natural light conditions. As shown in Fig. 4b, the 0.04 m$^2$ reactor, composed of an array of 0.01 m$^2$ HHE NFMs, yielded H$_2$ production totals of 38,805.3 µmol·m$^{-2}$, 32,103 µmol·m$^{-2}$, and 22,789.7 µmol·m$^{-2}$ under sunny (786 W·m$^{-2}$), partially cloudy (716.5 W·m$^{-2}$), and overcast (397.6 W·m$^{-2}$) conditions, respectively. During the first hour of outdoor test, the maximum H$_2$ production rates can reach 12148.5 µmol·m$^{-2}$·h$^{-1}$, 9286.4 µmol·m$^{-2}$·h$^{-1}$, and 7617.5 µmol·m$^{-2}$·h$^{-1}$, with a maximum solar to H$_2$ efficiency (STH) of 0.10%, 0.07%, and 0.085%, respectively. However, with the decrease of light intensity and the gradual release of moisture in the hygroscopic layer, the H$_2$ production rate gradually decreased. Therefore, the average H$_2$ production rates were 6467.55 µmol·m$^{-2}$·h$^{-1}$, 5350.5 µmol·m$^{-2}$·h$^{-1}$, and 3798.28 µmol·m$^{-2}$·h$^{-1}$ during the 6 h outdoor test, which is 5.71 times, 4.73 times and 3.36 times higher than that under indoor conditions with 1000 W·m$^{-2}$ AM 1.5 G simulated sunlight (Fig. 4c). Due to environmental variability and dimensional scaling of the HHE NFMs, the distinct H$_2$ production performance observed between indoor and outdoor conditions (contrary to trends reported in prior studies) demands a rigorous investigation into the underlying mechanisms.

The enhancement of H$_2$ production performance under outdoor larger-scale system can be attributed to the combined synergistic thermodynamic and kinetic effects. From the thermodynamic perspective, in the outdoor air-to-H$_2$ system, the HHE NFMs exhibit both higher energy input and higher surface temperature of PTC layer (Note S2 and S3). As shown in Fig. 4b, the temperature of the PTC layer under outdoor open area conditions can rise from 45.9 °C to 71.8 °C within 30 s (Fig. S39), reaching a maximum surface temperature of 82.2 °C (average ~76 °C) (Fig. 4d), which is about 14 °C higher than that of indoor conditions. As demonstrated by DFT calculations (Fig. 3h), the increased reaction temperature within the biphasic system lowers the energy barrier for H$_2$ formation, thereby accelerating the H$_2$ evolution rate. Thus, the enhanced surface temperature under outdoor conditions thermodynamically boosts H$_2$ production rates compared to indoor systems. From the kinetic perspective, the enhancement of H$_2$ production rates in HHE NFMs originates from scaling effects. In the indoor condition, a 0.0004 m$^2$ HHE NFMs was enclosed within a glass

reactor, whereas the outdoor system employed a 0.01 m$^2$ HHE NFMs array integrated into a 0.25 m$^2$ reactor. COMSOL simulations of the photothermal water desorption process for HHE NFMs of varying sizes revealed that moisture in the hygroscopic layer escapes not only from the top PTC layer but also from the sides of the hygroscopic layer (Figs. 4e and S40). However, as the HHE NFMs size increases, a greater proportion of vapor escapes through the PTC layer (Figs. 4f and S41). This indicates that larger HHE NFMs dimensions enhance reactant availability by increasing vapor contact with catalysts on the PTC layer, thereby elevating vapor concentration in the PTC layer and kinetically enhancing H$_2$ production rates.

Next, we prepared an array of HHE NFMs with a total area of 0.25 m$^2$ (Fig. S42) for large-scale H$_2$ production of the HHE system (Fig. 5a, b). After 14 days of moisture adsorption and H$_2$ production experiments, we verified the outdoor long-term feasibility and stability of the HHE system (Figs. 5d and S43, Supplementary Movie 1), with each square meter of HHE NFMs gathering up to approximately 0.31 kg of water from the air and 764 mL of H$_2$ under 6 h effective sunlight. During the two-week outdoor H$_2$ production tests, no significant degradation in H$_2$ production performance was observed (Figs. 5d and S44). The observed fluctuations correlated with daily weather variations, confirming that the efficiency remained stable under operational conditions. No detectable changes were observed in HHE NFMs before and after testing, confirming material stability (Figs. S45, S46). Subsequently, after two days of operation, the HHE system was connected to a fuel cell with an output power rate of 2 W, successfully lighting nearly 1000 commercial LED bulbs continuously (Fig. 5c and Supplementary Movie 2), demonstrating the practical applicability of the HHE system. Given its excellent performance, we conducted a comprehensive assessment of the H$_2$ production of the HHE system across terrestrial regions from 60 °N to 60 °S latitude (Fig. S53). Considering local solar intensity and average nighttime humidity conditions, application of HHE system could produce over 4.5 × 10$^4$ GL of H$_2$ per year (approximately 4.1 × 10$^9$ kg) across even just 0.1% of the global land area. Future research could focus on reactor optimization and material synergy to accelerate the moisture capture-to-conversion process, ultimately boosting H$_2$ production and solar utilization efficiency. For example, simplified reactor structures with solar-tracking capabilities will enhance solar energy utilization. Meanwhile, strengthened integration of hygroscopic materials and photocatalysts through precisely engineered pore structures, tailored surface properties, and enhanced catalytic activity will optimize airflow organization, light utilization. Overall, this study introduces a passive air-to-H$_2$ system without requiring any liquid water input, offering potential solutions to energy and water scarcity in developing regions.

## Methods

### Materials

Titanium dioxide (P25 TiO$_2$, purity ≥99.5%), chloroplatinic acid hexahydrate (H$_2$PtCl$_6$ · 6H$_2$O, purity ≥99.9%), gold (III) chloride trihydrate (HAuCl$_4$·3H$_2$O purity ≥99.9%), chromium (III) nitrate nonahydrate (Cr(NO$_3$)$_3$·9H$_2$O, 99.99%), 1,4-benzene dicarboxylic acid (H$_2$BDC, 99%), and absolute ethanol (EtOH, purity ≥99.7%) were obtained from Aladdin Biochemical Technology Co., Ltd., China. Polyacrylonitrile (PAN, Mw = 150,000, Ispin, China) was purchased from Kaneka Co., Ltd., Japan. N,N-Dimethylformamide (DMF, ≥99.5%) was supplied by Macklin Biochemical Co., Ltd., China. All chemicals were used without any further treatment. Ultrapure water (18.2 megaohm·cm) was obtained by using Mili-Q Synthesis System. Deuterium oxide (D$_2$O, 99.9%).

### Synthesis of Pt/TiO$_2$ and Au/TiO$_2$ nanoparticles

Photodeposition was conducted within a Pyrex-topped reaction cell. In a typical process, 200 mg of TiO$_2$ powder was suspended in 20 ml

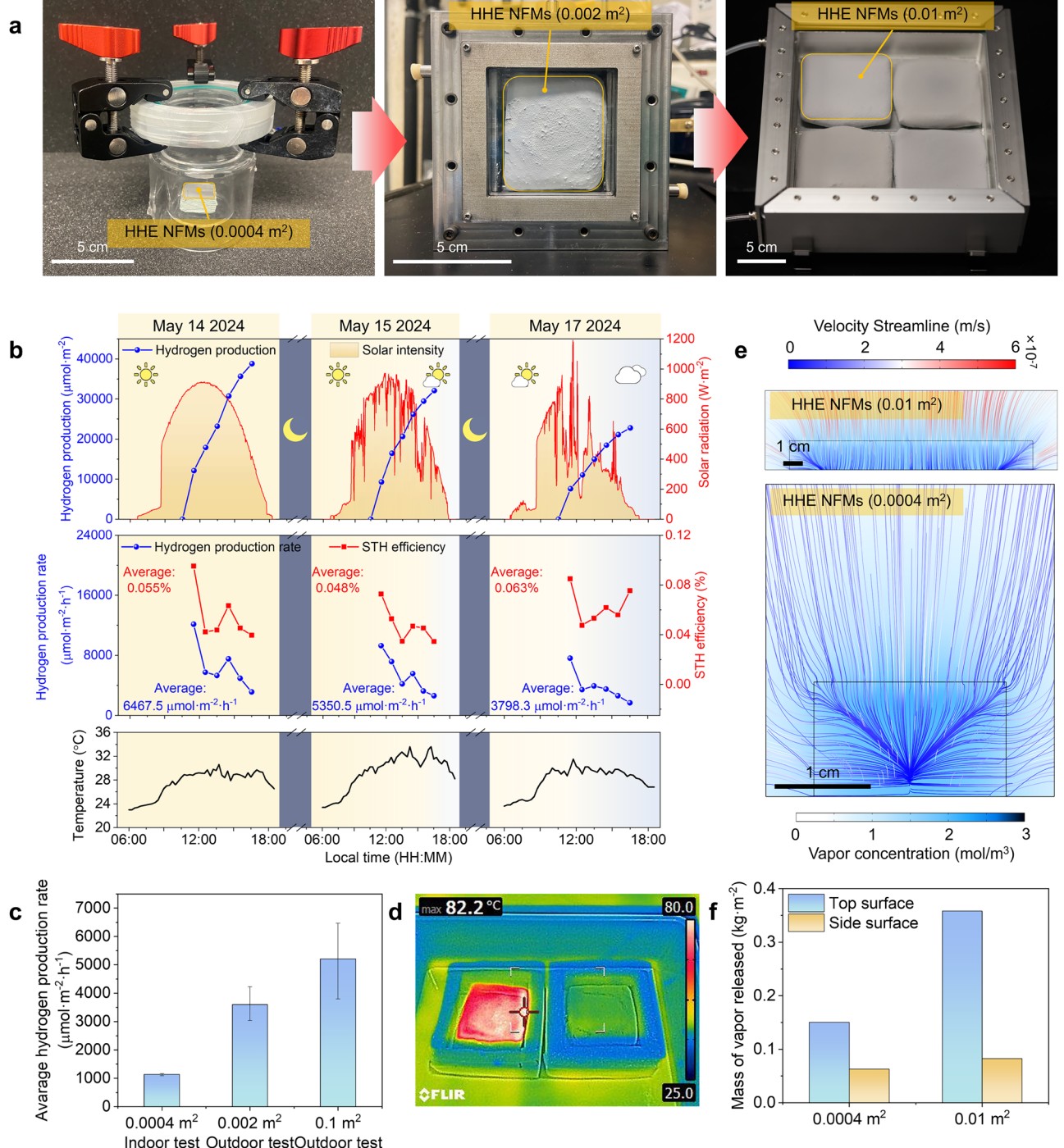

**Fig. 4 | Scaling up of HHE NFMs. a** Scaling up process of HHE NFMs from 0.0004 m² to 0.01 m². **b** H₂ production, solar radiation intensity, H₂ production rate and STH efficiency at corresponding air humidity and temperature under three typical weather condition. **c** Comparison of H₂ production rates between HHE NFMs of different sizes and varying reaction conditions. Error bars correspond to the standard deviation of three independent measurements. **d** IR thermal images revealing the top surface temperature of 0.01m² HHE NFMs with/without PTC layer under natural sunlight. **e** The velocity distribution of water vapor streamlines and concentration cloud maps of HHE NFMs cross-sections of different sizes obtained through 3D numerical simulations. **f** Comparison of water vapor release from the top and side surfaces of HHE NFMs with different sizes obtained through 3D numerical simulation.

water at 25 °C, and the mixture was then ultrasonicated for 30 min. After ultrasonication, a stoichiometric amount of H₂PtCl₆·6H₂O solution was added slowly under vigorous stirring (450 rpm). The amount of Pt metal added was 1 wt%, relative to the TiO₂. Then, the reaction vessel purged with argon and irradiation under a 300 W Xe lamp for 1 h. The photodeposited powders were washed and filtered (300 ml water) before drying at 80 °C for 12 h to obtain dry Pt/TiO₂ nanoparticles. The optimal pt load has been verified. Accordingly, Pt/TiO₂

catalysts with 0.5, 1, and 2 wt% Pt loadings (denoted as 0.5Pt/TiO₂, 1Pt/TiO₂, and 2Pt/TiO₂) by adjusting the amount of H₂PtCl₆ solution during synthesis were prepared and tested. Similarly, the Au/TiO₂ nanoparticles were prepared using the same process.

### Synthesis of MIL-101(Cr) nanoparticles
The reactant mixture with a molar composition of 1 Cr(NO₃)₃·9H₂O: 1 H₂BDC: 2.87 HNO₃: 267 H₂O was poured into a Teflon-lined stainless

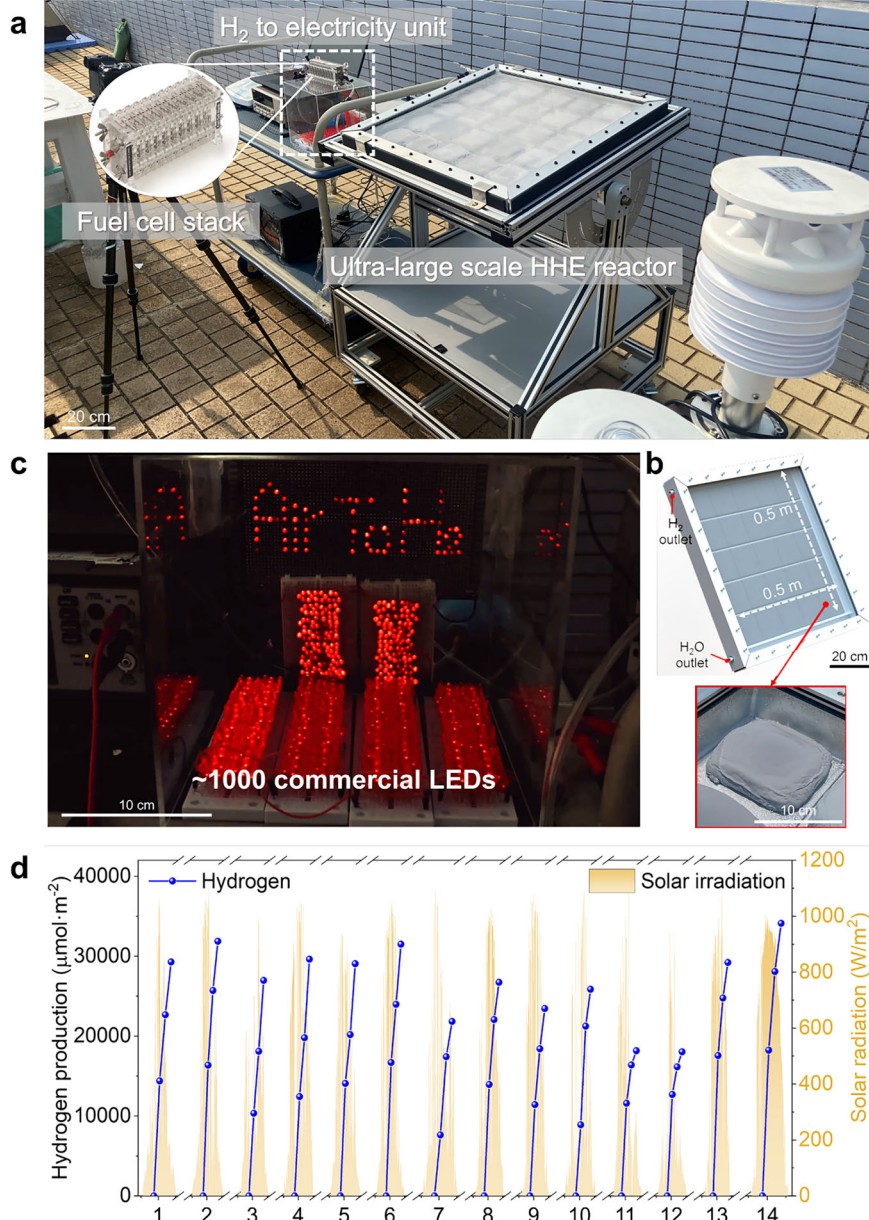

**Fig. 5 | Large scale air-to-hydrogen system. a** Outdoor set-up of large scale HHE system for sustainable $H_2$ production, powering ~1000 commercial LEDs. **b** The dimension of large scale HHE reactor and the distribution of HHE NFMs. **c** ~1000 commercial LED panels lighted by HHE system. **d** Operational status of the large-scale HHE system over a 14-day period. $H_2$ production and solar radiation intensity during the 14-day period.

steel autoclave and then followed by heating at 220 °C for 8 h. After being cooled to room temperature naturally, the green precipitates were obtained by centrifugation and washed several times with DI water, DMF, and hot ethanol. Finally, the product was dried in a vacuum oven at 150 °C for 12 h.

### Synthesis of MIL@PAN NFMs

A precursor solution for fabricating the MIL@PAN NFMs was prepared by ultrasonically dispersing 20 wt% MIL-101(Cr) in DMF for 2 h, followed by the addition of 5 wt% PAN and vigorous magnetic stirring of the blend for 12 h. This solution was electrospun into nanofibers using a commercial electrospinning apparatus (Huizhi Electrospinning Co., Ltd., China). Key electrospinning parameters, including a 15 cm working distance, a 1 mL·h$^{-1}$ flow rate, and a 20 kV applied voltage, were meticulously controlled. The electrospinning time was adjusted across

multiple hours to achieve varying membrane thicknesses under constant ambient conditions (25 ± 1 °C, 50 ± 3% RH). The obtained light green NFMs[27], exhibiting a multilayer honeycomb morphology, were then vacuum-dried at 100 °C for 12 h.

### Preparation of Pt/TiO₂@PVDF NFMs and Au/TiO₂@PVDF NFMs

10 wt% Pt/TiO$_2$ nanoparticles and 8 wt% of PVDF was dissolved in DMF under vigorous stirring for 12 h at 65 °C after 2 h of ultrasonic treatment. Resultant solution was electrospun under 1 mL·h$^{-1}$ feeding rate, 20 kV voltage, and the tip-roller distance of 15 cm was maintained. The as-prepared Pt/TiO$_2$@PVDF NFMs was deposited on a glossy paper-covered roller. In addition, the ambient temperature and RH were 25 ± 1 °C and 50 ± 3%, respectively. Finally, the gray Pt/TiO$_2$@PVDF NFMs was dried at 100 °C under vacuum for 12 h. Similarly, the Au/TiO$_2$@PVDF NFMs were prepared using the same process.

## Preparation of HHE NFMs

Preparation of Pt/TiO$_2$@PVDF NFMs as described above. Differently, multilayer MIL@PAN NFMs were covered on the spinning receiving disc for the electrospinning of Pt/TiO$_2$@PVDF. After several hours, the obtained bilayer NFMs were dried at 100 °C for 12 h under vacuum. Finally, the prepared bilayer NFMs were denoted as HHE NFMs.

## Characterization

A comprehensive suite of characterization techniques was employed to analyze the materials. The sample morphology was examined using scanning electron microscopy (SEM, Zeiss EVO MA10), with elemental mapping conducted on a TESCAN MIRA LMS system equipped for energy-dispersive X-ray spectroscopy (EDX). High-resolution transmission electron microscopy (HR-TEM) and corresponding EDX analysis were performed on a JEOL JEM-2100F. Crystal structure was determined by X-ray diffraction (XRD) on a PANalytical X'Pert3 powder diffractometer using Cu Kα radiation (λ = 1.54184 Å). Optical properties were assessed via ultraviolet-visible (UV-Vis) spectroscopy on a Shimadzu UV-3600 spectrophotometer. Surface temperature profiles and thermal images were captured with an FLIR EX-8 infrared camera. Chemical functional groups were identified by Fourier transform infrared (FTIR) spectroscopy using a Bruker Tensor 27 spectrometer. Textural properties, including specific surface area and pore volume, were measured via N$_2$ physisorption at 77 K on a Micromeritics ASAP 2020 system. Gas composition was monitored in real-time using a ThermoStar mass spectrometer.

## Indoor moisture adsorption measurement

Indoor moisture adsorption measurements were conducted following a standardized procedure. First, square samples (2 cm in length) were thoroughly dried in an oven at 100 °C until their mass stabilized. Subsequently, the samples were transferred to a constant temperature and humidity testing machine (QHP-150BE, LICHEN Technology) set to 25 °C and varying RH levels. The mass change during testing was recorded by an electronic balance (YOUSHENG, 0.1 mg) integrated with the system. To evaluate cyclic stability, the samples underwent repeated 6-h adsorption periods at 90% RH and 25 °C, interspersed with 1-h desorption periods at 100 °C. The adsorption capacity for each cycle was calculated according to Eq. (1).

$$C_{ads} = \frac{\Delta m}{m_0} \tag{1}$$

where $C_{ads}$ is the moisture adsorption capacity based on unit weight of hygroscopic layer g$_{water}$ · g$_{layer}^{-1}$, $\Delta m$ is the moisture adsorption quantity (g), and $m_0$ is the weight of dried hygroscopic layer (g).

## The relationship of absolute humidity, relative humidity and effective relative humidity

The ambient relative humidity (RH) is the amount of water vapor in the air as a percentage of the total amount that could be held at its current temperature, and can be expressed as:

$$RH = \frac{P_v}{p_a^*(T_a)} \tag{2}$$

where $p_v$ is the is the partial pressure of water vapor in the air, called absolute humidity, independent of temperature, and $P_a^*(T_a)$ is the saturated vapor pressure of water vapor at the current temperature. Correspondingly, the effective relative humidity near the sorbent (RH$_e$) is expressed as:

$$RH_e = \frac{P_v}{p_e^*(T_{surface})} \tag{3}$$

where $p_e^*(T_{surface})$ is the saturated vapor pressure of water vapor after temperature drops by 6.01 °C.

## Outdoor radiative cooling enhanced temperature and moisture adsorption measurement

The temperature tests for the outdoor radiative cooling enhanced AWH are presented in Fig. S10. The experiments were conducted on the rooftop of AC2 at City University of Hong Kong in Kowloon Tong on March 12, 2024. HHE NFMs samples were placed in an insulated cavity while exposed to the clear night sky. A polyethylene (PE) film served as a windbreak to minimize convective heat loss. Due to the excellent infrared transmittance of this PE film, it does not hinder the HHE NEFs from radiating heat into space[38]. Ambient temperature was measured using two thermocouples, while the sample temperature was monitored with a K-type thermocouple positioned at the yellow marker in Fig. S11.

The moisture adsorption tests for the outdoor radiative cooling enhanced AWH are presented in Fig. S12. The experiment took place on August 8, 2024, at the AC2 rooftop of City University of Hong Kong, located in Kowloon Tong, Hong Kong. HHE NFMs with an approximate area of 100 cm$^2$ were positioned on electronic balances with a precision of ±0.1 mg. For the radiative cooling-enhanced group (left panel of Fig. S12), the electronic balance surface was covered with a PE film to minimize wind interference and leave the window for radiative cooling. In contrast, the control group without radiative cooling (right panel of Fig. S12) was enclosed within an acrylic chamber wrapped with aluminum foil to block thermal radiation to outer space. Both of them was left enough gaps for air to get in. Prior to the commencement of the experiment, the test samples were subjected to a vacuum oven at 100 °C to determine their dry mass. Throughout the experiment, the mass of the sample, ambient temperature, and relative humidity were recorded every minute. The water absorption quantity was calculated using Eq. (1).

## Indoor theoretical radiative cooling enhanced water adsorption test

Simulation tests were conducted in a constant temperature and humidity chamber with controlled temperature and relative humidity. First, calculate the corresponding effective relative humidity after the average temperature drops by 6.01 °C by using Eq. (3), and use the result as the environment in the constant temperature and humidity box. Subsequently, a square sample with a side length of 2 cm was dried in a vacuum oven at 100 °C for 2 h, and then placed in a box to absorb moisture. At the same time, an electronic balance with an accuracy of 0.1 mg was used to record the mass change of the sample.

## Photocatalytic hydrogen production measurement in triphasic system

For H$_2$ production measurement of Pt/TiO$_2$ particles in DI water, 20 ml of deionized water was added to the transparent reactor chamber. To prepare the catalyst suspension, 20 mg of Pt/TiO$_2$ nanoparticles were immersed in water and subsequently added to a stirring rotor. Prior to initiating the photocatalytic reaction, the solution in the reactor underwent ultrasound treatment to achieve a homogeneous catalyst suspension. The light source used in the experiment is a 300 W xenon lamp, which is coupled with an AM 1.5 G filter (Fig. S21). The height of the light source is adjusted to ensure that the light intensity reaching the solution surface is maintained at 1000 W·m$^{-2}$. An injection of argon gas was carried out for a duration of 30 minutes to purge and remove as much air as possible from the reactor. During the photocatalytic reaction, gas samples were collected from the reactor at hourly intervals. The gas composition of the samples was subsequently quantitatively measured using meteorological chromatography (GC) to determine the H$_2$ yield within the reactor. Similarly, the H$_2$ production of Au/TiO$_2$ were measured using the same process.

## Indoor hydrogen production measurement of 0.0004 m² HHE NFMs

The indoor H₂ production measurement of HHE NFMs was carried out similarly to the triphasic system measurement. 0.0004 m² of the HHE NFMs, previously saturated with water at 25 °C and 90% RH, was put on the bottom of the transparent reactor chamber without adding any liquid water. The remaining testing process in the HHE system was similar to that of the triphasic system H₂ production.

## Indoor hydrogen production measurement of a biphasic system with vapor feeding

The biphasic system with vapor feeding is designed based on methods reported in the literature[39,40] (schematic and actual setup are shown in Fig. S23A). Specifically, 20 mg of Pt/TiO₂ was dispersed in 1 mL of ethanol solution and sonicated for 30 minutes. Then, the mixture was drop-cast onto a 4 cm² frosted glass plate placed on a hotplate at 60 °C. After solvent evaporation, the Pt/TiO₂ catalyst remained deposited on the glass plate, which was subsequently dried overnight in an oven at 80 °C. The catalyst-coated glass plate was then placed in a quartz reactor, with 20 mL of DI water positioned beneath the sample holder to serve as a vapor source. This setup maintained a relative humidity above 80% RH inside the reactor (as confirmed in Fig. S23B). The reactor was purged with argon for 30 minutes to remove air prior to irradiation under AM 1.5 G simulated sunlight (100 mW/cm²) for H₂ production test.

## Indoor solar evaporation experiment

Indoor solar evaporation experiments were performed under stable ambient conditions (~25 °C, ~45% RH). Prior to illumination, the HHE NFM sample (4 cm²) was first brought to moisture saturation by exposure to 90% RH at 25 °C. Desorption and evaporation were then driven by simulated AM1.5 G solar irradiation at an intensity of 1000 W·m⁻². The mass change of the sample was recorded at 5 min intervals throughout the one-sun illumination period. These time-resolved mass data were subsequently used to calculate the water evaporation rate (kg·m⁻²·h⁻¹).

## Isotope tracer experiment

Firstly, 2 cm × 2 cm HHE NFMs was placed in a vacuum oven and dried at 100 °C for 12 h. Then the HHE NFMs was quickly removed and placed in a glass reactor for sealing. At the same time, dry Ar is introduced into liquid phase pure $D_2O$. Then the wetted Ar is used to purge for 6 h to ensure that the hygroscopic layer of HHE NFMs can completely adsorb $D_2O$. The reactor was then exposed to AM 1.5 G light for 6 h, and then the gas products after reaction were introduced into the mass spectrometer to detect the gases composition. Similarly, the gas composition test of using $H_2O$ was measured by using the same process. For the isotope tracer experiment of oxygen, the liquid phase pure $H_2^{18}O$ is replace $D_2O$ and $N_2$ is used to take the $H_2^{18}O$ vapor into the reactor.

## Outdoor hydrogen production measurement of 0.002 m² HHE NFMs

The HHE NFMs, with an area of 0.002 m², were placed in a stainless steel reactor (Fig. 4a, middle). After sealing the reactor, it was purged with argon gas for 30 min and subsequently exposed to natural light. A weather station recorded solar radiation intensity, outdoor temperature, and humidity in real time. The amount of H₂ produced was measured using gas chromatography.

## Outdoor hydrogen production measurement of 0.01 m² HHE NFMs

Four HHE NFMs, each with an area of 0.01 m², were placed in an aluminum alloy reactor (Fig. 4a, right). After sealing, the reactor was purged with argon gas for 1 h. The reaction system was then positioned on the rooftop of AC2 at City University of Hong Kong in Kowloon Tong and exposed to natural light. A weather station recorded solar radiation intensity, outdoor temperature, and humidity in real time. The amount of H₂ produced was measured using gas chromatography.

## Outdoor daytime surface temperature measurement of 0.01 m² HHE NFMs

Two pieces of HHE NFMs (one has PTC layer and the other does not) were positioned inside a plastic box without the quartz glass covering under outdoor condition. The high walls around the plastic box are used to block wind and prevent it from affecting the experiment. Both of the two HHE NFMs were wrapped with insulating foam on all sides and underneath. An infrared thermal imager was used to record the upper surface temperature distribution of HHE NFMs exposed to outdoor sunlight.

## Outdoor hydrogen production measurement of a large-scale HHE system

Twenty-five 0.01 m² HHE NFMs were arranged in an array and filled into a large aluminum alloy reactor (Fig. S32). After complete sealing, the closed reaction system was purged with argon gas for 4 h. The reaction system was then positioned on the rooftop of AC2 at City University of Hong Kong in Kowloon Tong and exposed to natural light. A weather station was used to record solar radiation intensity, outdoor temperature, and humidity in real time. The amount of H₂ produced was measured using a GC system. After the outdoor H₂ production test, the reactor have to be opened to make sure that the HHE NFMs were exposed to the night sky for radiative cooling and to adsorb the moisture from the air for the next daytime H₂ production test. The quartz glass cover plate is part of a modular design with a reusable rubber ring system and quick-install clamp mechanism, ensuring consistent sealing after repeated reactor disassembly and reassembly (as shown in Fig. S42). The process of screwing the large-scale air-to-H₂ reactor together serves a dual purpose: securing the quick-install clamp assembly while applying compression to the rubber ring and quartz glass to create a tight fit and good sealing. Rigorous gas monitoring protocols were implemented: initial gas composition measurements were taken before each H₂ production experiment, followed by bihourly sampling to detect potential leaks through nitrogen (N₂) level monitoring by using gas chromatography (GC). Throughout extensive outdoor testing of the large-scale system, no gas leakage incidents were observed.

## STH efficiency

The STH efficiency of HHE system was calculated using Eq. (4):

$$STH = \frac{\text{Energy output as } H_2}{\text{Incident light energy}} = \frac{r_{H_2} \times \Delta G^0}{P_{\text{light}} \times A} \qquad (4)$$

where $r_{H_2}$ is the experimental H₂ production rate, $\Delta G^0$ is the standard Gibbs free energy, taken as 237 kJ·mol⁻¹ for liquid water and 228.6 kJ·mol⁻¹ for gas-phase water splitting at 298 K, and $P_{\text{light}}$ is the incident light intensity, and A is the irradiated area.

## 3D Numerical simulation

The simulations based on the finite element method were performed using COMSOL Multiphysics version 6.0. Three modules, namely laminar flow, humid air heat transfer, and moisture transport in the air, are used to investigate the concentration, temperature, and velocity distribution of evaporation process. Here, the simulation of water vapor photocatalytic H₂ production is ignored because its energy consumption only accounts for 0.1% of the incident solar energy, which is too small compared to other energies. Given the complexity of coupling reaction kinetics and phase-transport phenomena in the H₂ production process, fully integrating these effects would significantly increase computational demands. In this COMSOL study, our

primary objective was to validate that increasing the size of HHE NFMs (while maintaining a constant height) enhances the water vapor transport rate from the PTC layer, thereby supplying more reactants for the photothermal catalytic reaction. To maintain computational tractability, we simplified the model by focusing on the evaporation dynamics under illumination. 3D components were constructed for simulation, and the geometric parameters of the model are shown in Fig. S47. Free triangular elements are used in meshing the model. The relative tolerance in the steady-state solver was set to 0.01[41].

Our numerical simulations were conducted using COMSOL Multiphysics 6.0, employing a finite element approach to model the system. The primary goal of this study was to computationally verify that upscaling the HHE NFMs at a constant thickness augments the water vapor transport flux from the PTC layer, thereby enhancing reactant supply for the subsequent photothermal catalysis. To this end, we utilized three core modules, including laminar flow, humid air heat transfer, and moisture transport in air, to resolve the temperature, concentration, and velocity fields during water evaporation under one-sun illumination. Given the considerable computational cost associated with fully coupling complex reaction kinetics with phase-transport phenomena, and considering that the energy consumption of the photocatalytic step is negligible, we simplified the model by focusing exclusively on the evaporation dynamics. The 3D geometry, detailed in Fig. S47, was discretized using free triangular elements, and a relative tolerance of 0.01 was set in the steady-state solver to ensure computational efficiency.

The water vapor flow in the open area is simulated by solving Eq. (5) and Eq. (6):

$$\frac{\partial c_v}{\partial t} - D\nabla^2 c_v + u \cdot \nabla c_v = R_v \tag{5}$$

$$c_v = \varphi c_{sat} \tag{6}$$

where $c_v$ represents the concentration of water vapor, $D$ is the diffusion coefficient, $u$ denotes the velocity field induced by vapor diffusion. The saturation concentration is given by $c_{sat}$, $\varphi$ is the relative humidity, and $R_v$ is steam generation rate. Furthermore, the diffusion of water vapor, $g_w$, within the gas phase is defined by the following equation:

$$g_w = -\rho_g D_{eff} \nabla w_v \tag{7}$$

Here, the effective diffusion coefficient is given by $D_{eff} = D_{va}\varepsilon^{4/3}s_g^{10/3}$, and $D_{va}$ is the vapor-air diffusion coefficient with a value of $2.6 \times 10^{-5}\,m^2/s$. The moisture content within the HHE NFMs was determined experimentally. A correlation fitted from the measured water adsorption capacity at various humidity levels (Fig. S48) was incorporated into the model to define the internal moisture distribution. Furthermore, ambient RH variations influence the evaporation energy demand of the HHE NFMs by altering the environmental vapor pressure (and thus vapor density). The accompanying heat transfer was simulated by solving the following energy conservation equations:

$$\rho C_p \frac{\partial T}{\partial t} - \rho C_p \nabla T - k\nabla^2 T = Q_v \tag{8}$$

where $T$ is temperature, $\rho$, $C_p$, and $k$ represent the density, specific heat capacity, and thermal conductivity, respectively. To account for the endothermic process of evaporation, the associated latent heat absorption is incorporated as a volumetric heat source term $Q_v$, given by:

$$Q_v = -H_v M_w R_v \tag{9}$$

where $H_v$ and $M_w$ are the latent heat of evaporation and molecular weight of water, respectively. Incident illumination was modeled as a heat flux boundary condition on the top surface of HHE NFMs. The assembly was treated as a single domain, justified by the PTC layer's minimal thickness (130 μm) relative to the 1.1 cm hygroscopic layer. The model incorporates surface heat losses from convection and radiation to a 298.15 K environment and includes the temperature dependence of $c_{sat}$. The detailed simulated material properties and boundary conditions are cataloged in Tables S3 and S4.

## DFT simulation

All spin-polarized DFT computations were performed by using the Vienna Ab initio Simulation Package (VASP 6.4)[42]. The exchange-correlation interaction was described by the Generalized gradient approximation (GGA) with Perdew-Burke-Ernzerhof (PBE) functional[43]. Projector augmented-wave (PAW) potential with a kinetic energy cut-off of 400 eV was employed to treat the ion-electron interaction. To account for van der Waals (VdW) interactions, Grimme's DFT-D3 method with the Becke-Johnson damping function was used[44]. The GGA + $U$ approach was applied with a $U$–$J$ value of 4.0 eV to account for the Ti 3 d electrons[45–47].

Convergence criteria for the structural optimization were set to ensure that the total energies per atom and forces on each atom reached values less than $10^{-5}$ eV and 0.02 eV Å$^{-1}$, respectively. The anatase TiO$_2$ model (tetragonal, space group I41/amd) with lattice parameters $a = b = 3.78$ Å, $c = 9.62$ Å, was sourced from an open-access database (The Materials Project). Next, a four-layered TiO$_2$ (101) slab with $5 \times 3$ supercell was constructed based on X-ray diffraction (XRD) results, where the bottom layer was fixed during structural optimization. A 24-atom Pt cluster (Pt$_{24}$) was initially placed on TiO$_2$ (101) surface. All slab structures included a vacuum region of 20 Å. For Brillouin zone sampling, gamma k-point was utilized. The optimized structures of Pt$_{24}$/TiO$_2$(101) and the adsorbed H$_2$O and H atom on Pt$_{24}$ are shown in Fig. S49. The Gibbs free energy of adsorption was computed based of the following equation:

$$\Delta G = \Delta E + \Delta E_{ZPE} - T\Delta S + \Delta G_{pH} \tag{10}$$

where $\Delta E$ is the electronic energy difference between the free-standing and adsorption states of the intermediates; $\Delta E_{ZPE}$ and $\Delta S$ are the changes in zero-point energies (ZPE) and entropy, respectively, which are obtained from the vibrational frequency calculations; $T$ is the temperature; $\Delta G_{pH}$ is the free energy correction to account for the effect of solution pH value, which can be calculated from the following equation:

$$\Delta G_{pH} = K_{BT} \times pH \times \ln 10 \tag{11}$$

In this study, free-standing H$_2$ and H$_2$O served as the reference states, respectively. The solvent effect was taken into account by using the implicit solvation model based on the VASPsol[48,49].

The relative permittivity of liquid water and gaseous water were set to be 80 and 1, respectively. The Gibbs free energies in the bi- and tri-phasic systems are dependent on the environment (bi- and tri-phasic system), temperature (from 298 K to 373 K) and the entropy change $\Delta S$. The thermodynamic properties of water and H$_2$ were referenced from the Handbook of Fundamentals of Classical Thermodynamics[50].

## Calculation of the potential of H$_2$ production from HHE systems across terrestrial regions from 60 °N to 60 °S latitude throughout the year

Firstly, based on the light intensity and hydrogen production data collected from long-term outdoor testing of the air-to-H$_2$ system, we calculated the average hydrogen production rate $R_{H_2}$ (mL · m$^{-2}$ · h$^{-1}$)

under different average solar radiation ($P_{light}$). And based on these data, a relationship curve between average solar radiation and average hydrogen production rate was fitted (Fig. S54A), and the fitting formula is:

$$R_{\mathrm{H_2}} = 3.73 \left( \frac{P_{light}}{12} \right)^{0.53} \tag{12}$$

Secondly, based on the data obtained from the radiative cooling enhanced atmospheric water harvesting, we also established the relationship between RH and the water adsorption saturation ($\alpha$) of HHE NFMs (Fig. S54B), and fitted a curve with the following formula:

$$\alpha = -\frac{1.00189}{1 + (\mathrm{RH}/0.33612)^{6.25821}} + 0.9996 \tag{13}$$

Therefore, based on the above two fitting curves, the annual average hydrogen production per square meter of land ($H_{2,year}$/ ($\mathrm{L \cdot m^{-2} \cdot year^{-1}}$)) can be obtained:

$$\mathrm{H}_{2,year} = \alpha \frac{t R_{\mathrm{H_2}}}{1000} \cdot \mathrm{day} \tag{14}$$

Here, the direct solar irradiation all over the world can be obtained from the Global Solar Atlas (https://globalsolaratlas.info) (Fig. S55). And the Global nighttime average humidity distribution can be obtained from the Asia-Pacific Data-Research Center, Reanalysis Data, NCEP Reanalysis (http://apdrc.soest.hawaii.edu/las/v6/dataset?catitem=16801) (Fig. S56). Finally, the calculated annual average $H_2$ production data will be visualized on a map for reference, and further calculations will be made to determine the amount of $H_2$ available for the air-to-$H_2$ system from 0.1% of the land.

## Data availability
All data are available in the main text or the supplementary information. Source data are provided with this paper.

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

## Acknowledgements
We acknowledge the financial support from the Research Grants Council of Hong Kong, the Guangdong Basic and Applied Basic Research Foundation (2023A1515110920 and 2024A1515012307), and the Sichuan Science and Technology Program (2024NSFSC1141).

## Author contributions
Conceptualization: S.W., Y.H.N. Methodology: Q.X., X.Y., H.Y.C., X.L., X.C.Z., Y.H.N., and S.W. Investigation: Q.X., X.Y., H.Y.C., X.L., Z.Z. (Zhi Z), Z.Z. (Zhenwen Z), W.K.L., Y.Z., X.C.Z., Y.H.N., and S.W. Visualization: Q.X., X.Y. Funding acquisition: Y.H.N., S.W. Project administration: Y.H.N., S.W. Supervision: X.C.Z., Y.H.N., and S.W. Writing—original draft: Q.X., X.Y., Y.H.N., and S.W. Writing—review & editing: Q.X., X.Y., X.C.Z., Y.H.N., and S.W.

## Competing interests
The authors declare no competing interests.
