## [Transparent Peer Review file · Nature Communications]

Electricity-Free Hydrogen Production from the Air

Corresponding Author: Professor Steven Wang

Version 0:

Reviewer comments:

Reviewer #1

(Remarks to the Author)

Journal : Nature Communications

Manuscript ID : NCOMMS-25-00566-T

Title : Electricity-Free Hydrogen Production from the Air

The study proposes to capture moisture from the air and produce hydrogen without the need for electricity or liquid water. The demonstration unit is based on a radiative cooling-enhanced atmospheric water harvesting (AWH) with a photothermal catalytic (PTC) process to extract and split water. A multilayer nanofiber membrane composed of a hygroscopic bottom layer (MIL@PAN NFMs) that captures water from the air and a photothermal catalytic top layer (Pt/TiO₂@PVDF NFMs) that utilizes sunlight to drive the hydrogen evolution reaction has been used. The following are the specific comments that need to be addressed,

1. Page 5, line 15, the claim that “demonstrating nearly 10% improvement in performance compared to the sample without the radiative cooling enhanced one (0.82 gwater-1 ·glayer-1)”. It is not clear whether the same setup has been used to perform the experiments without radiative cooling or it has been taken from literature.
2. The manuscript claims that a hydrogen production rate of 6467.55 $\mu\text{mol}\cdot\text{m}^{-2}\cdot\text{h}^{-1}$ under outdoor conditions, which is nearly six times higher than indoor experiment. The plausible factors enhancing the hydrogen production rate for outdoor experiments compared to indoor experiments must be mentioned.
3. Page 18, line 10, mentions “The GGA + U method was used with U-J value of 4.0 for Ti 3d electrons”. The .xyz file of the catalyst slab used for the computation may be included in the Electronic Supplement.
4. Is numerical vibrational frequency computation was done on optimized catalyst slab, for estimating delta G.
5. If it is so which space group and arrangement of original unit cell was used, must be mentioned.
6. The claim that “The system can theoretically produce $\sim 4.5 \times 10^4$ GL (gigaliters) of hydrogen annually across 0.1% of global land areas.” is a very tall claim considering the amount of catalytic material and their production requirement.
7. Hydrogen production happens only when sunlight is available. Cloudy, rainy days and nighttime will not produce hydrogen. Though moisture collection happens at night, there will be no hydrogen production.
8. The solar-to-hydrogen (STH) efficiency of 0.1%–0.085% is reported in manuscript is very much lower than electrolysis-based hydrogen production, which is around 15–20%. It is not clear how the system may compete against electrolysis.
9. The hygroscopic and photocatalytic materials (Pt/TiO₂@PVDF and MIL@PAN NFMs) need to maintain their performance over long-term outdoor exposure. Potential degradation due to UV exposure, environmental pollutants, or material fatigue over time could reduce efficiency. The cycling stability of the moisture adsorption-desorption process must be further tested for real-world applications.
10. Is there any other material than platinum Pt and Au which can be used which is cheaper and abundant. A selection of these two elements must be justified.
11. Page 17, the results from 3D numerical simulations must be presented.
12. The system is a breakthrough in sustainable hydrogen production, but low efficiency, weather dependence, and scalability challenges must be addressed before commercial deployment.
13. Despite these limitations, the study presents a promising off-grid hydrogen production technology for arid and water-scarce regions, contributing to sustainable energy transitions.

Based on the above comments, the manuscript is “recommended with minor revisions” for publication after incorporation of comments.

Reviewer #2

(Remarks to the Author)

This manuscript presents a hydrogen production device that captures moisture from ambient air using MIL@PAN nanofiber membranes (NFM) with a radiative cooler (Pt/TiO₂@PVDF) during nighttime, and the collected water is subsequently converted into hydrogen via a photothermal catalytic process using hygroscopic H₂ evolution nanofiber membranes (HHE NFM) during daylight hours. While the research topic is interesting, the manuscript requires significant revisions based on the following comments.

1) Please provide the energy dispersive spectroscopy data and the specific surface area (m²/g) of the honeycomb-like structure composed of MIL@PAN.

2) The authors should evaluate the energy balance of the hydrogen production device for both indoor and outdoor experiments.

Optical characteristics: The photothermal catalytic (PTC) top layer, consisting of Pt/TiO₂@PVDF NFM, absorbs supplied energy as heat, with an absorbance of 0.95 in the ultraviolet range and 0.7 in the visible-near infrared spectrum. It simultaneously achieves radiative cooling, with an emissivity of 0.95 in the mid-infrared range.

In the indoor experiment under AM 1.5G conditions (1,000 W/m²), the surface temperature of the hydrogen production device was maintained at 68 °C, and the evaporation rate was 0.298 kg/m² h. The net radiative power absorbed by the device was calculated as 713 W/m² by integrating the product of absorbance and supplied power over the wavelength. Thermal radiation within the indoor environment is negligible. Latent heat absorption from water desorption was measured at 187 W/m². The remaining power, 526 W/m², corresponds to the sum of heat transferred to the surrounding air and the endothermic heat of the hydrogen conversion process.

However, in the outdoor experiment, the surface temperature of the device increased to 82.2 °C. This raises questions about the energy balance, when the net absorbed power under 1,000 W/m² solar flux is 713 W/m², while the thermal emission of the device is calculated as 858.9 W/m² at 82.2 °C. Additionally, water desorption from the porous MIL@PAN material absorbs latent heat at 2,260 kJ/kg-water, alongside the endothermic reaction heat of hydrogen production. This discrepancy suggests that the energy balance cannot be fully explained, even though the system is described as entirely insulated. It is also noted that the outdoor solar flux is measured as 786 W/m², significantly lower than the indoor value of 1,000 W/m². This further complicates the energy balance, as the heat transfer coefficient of the system with outdoor air should logically be higher than that with indoor air. The authors should clarify these discrepancies and provide a detailed analysis of the energy balance under outdoor conditions.

3) The numerical simulation conducted using COMSOL Multiphysics lacks sufficient details. Specifically, when desorbed water molecules undergo gas-phase water splitting for hydrogen production via a photothermal catalytic process in the indoor experiment, this endothermic reaction likely impacts the overall energy balance, which was not explained in the manuscript. Please discuss the effect of this phenomenon. Additionally, more comprehensive information regarding the initial and boundary conditions, as well as the analytical scheme used, should be included.

4) Why was MIL-101(Cr) used for water harvesting from ambient air? The moisture sorption performance (both kinetics and capacity) of MIL-101(Cr) is significantly lower than that of other materials under low humidity conditions. While the radiative cooler was utilized to enhance the moisture sorption performance of MIL-101(Cr) under low-humidity conditions, it also reduces net energy absorption during daylight hours due to thermal radiation. Alternatively, materials with higher sorption capacity at low humidity could be considered. Please discuss the advantages of combining the MIL@PAN layer with the radiative cooler in this context.

5) It is recommended to describe the performance of the hydrogen production device under outdoor conditions from an energy perspective. For example, the energy consumption required to produce a unit mass of hydrogen (J/g or J/mol) could be calculated and compared with other studies.

6) The hydrogen production device used for outdoor experiments appears to be insulated. The authors should discuss the insulation materials utilized and calculate the overall thermal resistances of the device in each direction.

7) Although the net energy absorbed by the hydrogen production device under outdoor conditions is lower than that under indoor conditions, how can the hydrogen production rate increase? Please consider the following factors:

Outdoor experiments: Solar flux of 786 W/m², thermal emission from the surface (absorbance/emissivity of 0.95), and a higher heat transfer coefficient due to windy conditions.

Indoor experiments: Power flux of 1,000 W/m² from the solar simulator, regulated thermal emission due to the indoor environment, and a lower heat transfer coefficient.

When the energy flux is high and a high surface temperature is achieved, improved efficiency can be explained thermodynamically using the Gibbs energy relation. However, this does not align with the lower energy environment described in outdoor experiments.

8) One potential reason for higher efficiency under outdoor conditions could relate to the wavelength distribution and intensity of the solar simulator used in the indoor tests. Please provide specifications for the solar simulator, including its wavelength range and intensity distribution.

Reviewer #3

(Remarks to the Author)

The author discusses a method that enhances atmospheric water harvesting using radiative cooling and achieves hydrogen production via photothermal and photocatalytic water splitting in the vapor phase. I have three main concerns regarding this paper:

Regarding Radiative Cooling

The author states that the PTC layer has a high emissivity, enabling radiative cooling at night. Figure 2g shows that the surface temperature of the HHE NMFs is approximately 6°C lower than the ambient air temperature. However, the author does not explicitly mention that this result was obtained with the HHE NMFs directly exposed to air—this detail can only be found in Supplementary Figure S10. In subsequent system experiments, the PTC layer and the water-absorbing layer were placed under a transparent window, as can be seen in the video materials. Under these conditions, the radiative cooling effect should be significantly weaker than what is presented in Figure 2g. Similarly, I suspect that the infrared temperature measurements shown in Figure 4d were also taken without the transparent window, although the author does not clarify this. These results, therefore, do not reflect the conditions with a transparent window. Given these concerns, the data presented in the earlier sections of the paper do not fully support the analysis and discussion of the system operation results in the later sections.

Regarding Hydrogen Production Rate under Outdoor Natural Conditions

The author exaggerates the superiority of the outdoor experimental results over those obtained under indoor illumination in both the abstract and main text. This comparison is unnecessary because the two conditions differ significantly in terms of operating temperatures and geometric characteristics, making them fundamentally incomparable.

Regarding Scalability

The author's estimation of hydrogen production based on 0.1% of the global land area lacks credible data support and has no scientific value.

Given the inconsistencies between fundamental data and system results, as well as the arbitrary nature of comparisons and estimations, I recommend rejecting the paper.

Version 1:

Reviewer comments:

Reviewer #1

(Remarks to the Author)

As my comments have been successfully incorporated, the manuscript is recommended for publication.

Reviewer #2

(Remarks to the Author)

The authors answered all the comments correctly, so I recommend this manuscript for publication in Nature Communications.

Reviewer #3

(Remarks to the Author)

Although the author has addressed my questions, the responses are clearly not very persuasive. I still have reservations about this paper.

1. Although the author added a description that the quartz glass cover plate was intentionally removed during nighttime operations to allow direct exposure of the HHE NMFs to ambient air for moisture adsorption, I would like to inquire how the frequent removal and reinstallation of the quartz glass cover in this integrated hydrogen production and water collection system maintains its seal. The revised version in Fig. 4b shows the hydrogen production rate under typical weather conditions during the day, while Fig. 5d only presents hydrogen production results. Fig. 2 is a standalone analysis of radiation cooling water collection, but I cannot find any experimental results in the entire paper that demonstrate the continuous operation of the large-scale air-to-hydrogen system (as shown in Fig. 5a) from daytime hydrogen production to nighttime water collection. Therefore, I believe that the author's statement about the removal of the quartz glass cover plate at night lacks credibility.

2. Regarding the explanation of the HHE NFM surface temperature records in Fig. 4g, the author conducted a theoretical analysis of the radiation cooling power with and without the quartz glass in order to account for the temperature differences of the PTC surface. In the response on page 42, the first five lines confirm that there is indeed a difference between the two configurations. However, the next sentence immediately concludes that the temperature difference between the two is minimal, and its overall impact on the thermal behavior is negligible. This conclusion seems overly forced and illogical. Additionally, the revised version of the figure does not contain Fig. 4g, which further heightens the reviewer's doubts.

Version 2:

Reviewer comments:

Reviewer #4

(Remarks to the Author)

I have carefully read the Rev #3's comments and the authors' responses, and I personally believe that the authors have well addressed the concerns raised by the Rev #3.

This is an engineering system work that contains many engineering details. The discussion between the Rev #3 and the authors on these details is conducive to the improvement of engineering experiments and the enhancement of reproducibility.

This work proposes a dual-functional composite nanofiber membrane (HHE NFMs) system that does not require electrical energy input and relies on air water vapor to directly produce hydrogen, integrating the hygroscopic and photothermal-photocatalytic synergistic hydrogen production mechanisms driven by radiation cooling. The paper is ingeniously conceived and novelly designed, with obvious originality and application potential, especially suitable for green hydrogen energy solutions in arid or water-scarce areas. The authors conducted systematic material construction, structural characterization, climate adaptation testing, and theoretical simulations, and the research content is rich. However, although the study proposed important concepts and demonstrated significant experimental results, some experiments still need to be further improved. The following are specific suggestions :

1. The experimental conditions for the Pt/TiO₂@PVDF nanofiber membranes (NFMs) and MIL-101(Cr)@PAN nanofiber membranes used in the article need further explanation. For example: Why was the Pt loading of 1 wt% chosen? Have you done any experiments to optimize the loading? What is the basis for selecting the loading and membrane thickness of MIL-101(Cr)? Is there any optimization experimental data to support this ?
2. It is recommended that the authors further improve the description of the experimental conditions, such as the specific definition of different weather conditions (sunny, partly cloudy, and overcast) and the frequency of recording environmental parameters, to facilitate other researchers to repeat the experiment.
3. The data presented in the chart is relatively complete, but it is recommended to add error bars or standard deviations of repeated experiments to the graph to reflect the stability and reliability of the experimental results.
4. The author mentioned that the humidity adsorption capacity of MIL-101(Cr) varies greatly at different humidity levels (for example, it decreases significantly at 50% humidity). Can you further provide adsorption data at lower humidity conditions (such as 20%, 30%, 40%) to clarify the applicable humidity lower limit of this material?
5. Some of the terms used in the article are too subjective, such as "ingeniously combine" and "we are surprised to find". We suggest using more neutral terms, such as "systematically integrate" and "it is noteworthy that".
6. To determine whether there are microstructural changes (such as agglomeration or detachment of MOF particles) in MIL-101(Cr)@PAN nanofibers after multiple adsorption-desorption cycles, it is recommended to supplement with TEM or SEM high-magnification characterization images after long-term cycling.

Reviewer #5

(Remarks to the Author)

I have carefully reviewed the revisions and acknowledge that the authors have addressed Reviewer #3's concerns by supplementing structural evidence to demonstrate sealing integrity during cover removal and by providing quantitative thermal analysis showing the negligible impact of the quartz glass on the overall reactor thermal performance.

In this manuscript, the authors report on spectral selective absorbing/emitting HHE nanofiber membranes (NFMs) employed as a self-sufficient, electricity-free air-to-H₂ system. This system combines radiative cooling with advanced photothermal catalytic processes. The H₂ production rate of scale-up air-to-H₂ system under outdoor natural light reaches 6467.55 μmol·m⁻² h⁻¹. However, some analytical results are not sufficiently convincing, and the study suffers from insufficient in-depth mechanistic investigations, which hinders a comprehensive assessment of its practical application potential. Consequently, I cannot recommend this paper for publication in Nature Communications in its current form (see below the details).

1. As shown in Fig. 3d, the schematic diagram of PTC vapor decomposition reactions indicates H₂ and O₂ as the primary products. However, the authors have not provided experimental evidence of O₂. Additionally, the author should present evidence regarding the formation of other potential oxidation products. The ambiguity surrounding the anodic reaction pathway introduces risks of unaccounted side reactions.
2. Can the authors provide quantitative evidence (e.g. interfacial adsorption energies, charge-transfer measurements) to enhance readers' understanding of why the biphasic system is better than the triphasic system?
3. It is better to compare this system with other recently reported photocatalytic air-to-H₂ systems in the literature, including parameters such as STH and quantum efficiency.
4. Specially, regarding nighttime operation: does the system store sufficient water to sustain daytime hydrogen production?
5. Please supplement data on the hygroscopic and hydrogen production performance under varying humidity levels (30%–90% RH).
6. Please conduct supplementary cyclic tests to evaluate the hygroscopic and hydrogen production stability of the composite system.
7. While radiative cooling is claimed to enhance hygroscopicity, the moisture saturation time is only 1 hour. How is this purported advantage substantiated?
8. Please explicitly state the functional contribution of each component within the main text.
9. The original DFT calculations used energy barrier data at 100°C. However, since the actual maximum operational temperature did not reach 100°C, this data is unreliable. Please provide supplementary DFT calculations at the actual operational temperatures to demonstrate the trend of decreasing reaction energy barriers with increasing temperature.
10. Please assess the solar energy utilization efficiency of the system.

11. Please compare the hydrogen production performance between the two-phase system and the system after in-situ moisture absorption, rather than the three-phase system under conditions of abundant water supply.

12. The mechanism of thermal-photocatalytic synergy remains ambiguous. It is not elucidated how infrared photothermal effects facilitate the separation/transfer of photogenerated charges.

Version 3:

Reviewer comments:

Reviewer #4

(Remarks to the Author)

Any science can only benefit the world if it is transformed into engineering. I think engineering-related works should also be eligible for publication in Nature Communicationse and I will no longer able to review this manuscript.

Reviewer #5

(Remarks to the Author)

NCOMMS-25-00566-T

Point-to point response to reviewers' comments on the manuscript

'Electricity-Free Hydrogen Production from the Air'

We wish to thank the three Reviewers for their comments. First, we have carefully
5 studied all these comments. Then, we made a major revision of our initial manuscript
by accounting for the comments of the three Reviewers as much as possible and in a
constructive way. We sincerely apologize for any repetition in our responses, as some
of the reviewers' comments addressed overlapping aspects of the work. Please rest
assured that each response has been carefully tailored to address the specific concerns
10 raised. We greatly appreciate your kind understanding and patience in reviewing this
material. We believe that this revision leads to a substantial improvement of our initial
manuscript.

Below, we explain how all the comments of the three Reviewers have been taken into
account in our revision. In this response letter, the reviewers' comments to our original
15 manuscript are provided in **black**, and **our point-by-point responses as well as the
corresponding changes are shown in blue.**

Reviewer #1:

[General comment]: The study proposes to capture moisture from the air and produce hydrogen without the need for electricity or liquid water. The demonstration unit is based on a radiative cooling-enhanced atmospheric water harvesting (AWH) with a photothermal catalytic (PTC) process to extract and split water. A multilayer nanofiber membrane composed of a hygroscopic bottom layer (MIL@PAN NFMs) that captures water from the air and a photothermal catalytic top layer (Pt/TiO₂@PVDF NFMs) that utilizes sunlight to drive the hydrogen evolution reaction has been used. The following are the specific comments that need to be addressed.

[Response]: We sincerely thank the reviewer for reviewing our work and giving the valuable comments! We have addressed all the comments through further experimentation and have made revisions on the manuscript, as specified below. We hope our response satisfies you.

[Comment 1]: Page 5, line 15, the claim that “demonstrating nearly 10% improvement in performance compared to the sample without the radiative cooling enhanced one (0.82 g_{water}⁻¹ · g_{layer}⁻¹)”. It is not clear whether the same setup has been used to perform the experiments without radiative cooling or it has been taken from literature.

[Response]: Thank you very much for your careful reading and useful comment. We apologize for the lack of clarity in our original manuscript. Here, we aim to demonstrate that the radiative cooling enhanced HHE NFMs exhibit a 10% improvement in water uptake capacity compared to their non-radiative cooling enhanced counterparts which comes from our own HHE NFMs. The same setup has been used to perform the experiments with and without radiative cooling HHE NFMs. As the reviewer rightly pointed out, we have supplemented relevant information in the revised manuscript and method part to prevent potential ambiguity (revised manuscript, page 5, line 13-17).

[Comment 2]: The manuscript claims that a hydrogen production rate of 6467.55 μmol·m⁻²·h⁻¹ under outdoor conditions, which is nearly six times higher than indoor

experiment. The plausible factors enhancing the hydrogen production rate for outdoor experiments compared to indoor experiments must be mentioned.

[Response]: We sincerely thank the reviewer for these insightful comments. While the reasons for the enhanced H₂ production rate were mentioned on page 9, line 17 of the original manuscript, we acknowledge that our explanation may not have been sufficiently clear. Here, we provide a more detailed analysis to clarify why the hydrogen production rate of the air-to-H₂ system under outdoor conditions is 6 times higher than that observed in indoor conditions. We analyze that this enhancement arises from the combined synergistic **thermodynamic** and **kinetic** effects.

1. Thermodynamic perspective:

A comprehensive energy analysis of the photothermal catalytic (PTC) layer surface reveals that under outdoor conditions, the upper surface absorbs significantly more solar energy, leading to elevated surface temperatures. This thermal effect thermodynamically favors the hydrogen evolution reaction. The specific mechanisms are elaborated as follows:

(1) Discrepancies in spectral intensity distribution between indoor simulated sunlight and outdoor natural light

First, we provide detailed specifications of the simulated solar light source used in indoor experiments. The xenon lamp system (Newport Model 67005 with Power Supply 69911) equipped with an AM1.5G filter was employed. Fig. R1 presents the manufacturer's spectral irradiance profile and our measured spectral distribution when using the AM1.5G filter. Notably, comparative analysis reveals that, in the ultraviolet range and visible range (625-800 nm), the filtered xenon lamp exhibits higher irradiance than natural sunlight. However, in the 450-625 nm of visible light, filtered xenon lamp exhibits lower irradiance. While these spectral mismatches were shown in the figure, these differences alone cannot fully explain the six times increase in her performance observed under outdoor conditions.

Fig. R1 A) The spectral irradiance of various Arc Lamps. (Here the 628 300W Xe OZONE FREE is what we used). B) The spectral irradiance of natural sunlight and 300W Xe light with AM1.5g filter.

5 (2) Light scattering by condensed liquid films on glass reactor surfaces

The solar energy absorbed by the surface (P_{sun}) of HHE NFMs under outdoor conditions and indoor conditions can be calculated by¹:

$$P_{sun} = \int_0^{\infty} [\varepsilon_s(\lambda, \theta_{sun}) I_{AM1.5}(\lambda)] d\lambda$$

where $\varepsilon_s(\lambda, \theta_{sun})$ represents the absorptance/emittance of the surface of HHE NFMs, which is PTC layer, and $I_{AM1.5}(\lambda)$ represents the AM1.5 solar irradiance. In the indoor H₂ production tests, the HHE NFMs were encapsulated within a **quartz glass reactor** to maintain a sealed environment and prevent hydrogen leakage. Theoretically, the P_{sun} calculation should be based on the standard simulated solar radiation of 1000 W/m². However, under indoor conditions, the temperature of the reactor's top glass window and sidewalls remained close to ambient temperature (~25°C), and water vapor released during evaporation condensed on these surfaces, forming liquid droplets beneath the glass window. Optical characterization reveals that these condensate droplets induce notable light scattering and absorption. Consequently, the actual photon flux reaching the PTC layer was reduced. To validate this conclusion, we set the simulated solar irradiance to 1000 W/m², as shown in Fig. R2A. Under illumination, Fig. R2B shows that liquid film rapidly formed and persisted on the underside of the quartz glass cover plate. To evaluate the impact of this liquid film on light intensity, a quartz plate with an analogous liquid film was positioned above a radiometer at the same height as the

reactor's quartz cover plate. The results from Fig. R2C demonstrated a significant irradiance reduction due to light scattering by the liquid film, decreasing from 1000 W/m² to **745 W/m²**. Consequently, the effective solar irradiance absorbed by the HHE NFMs surface within the reactor under indoor conditions ($P_{sun, indoor}$) was calculated as **534.1 W/m²**. The presence of the liquid film in the enclosed reactor thus reduced the solar irradiance absorbed by the PTC layer, leading to both (i) lower surface temperature compared to the open-environment measurement under indoor conditions and (ii) lower photoexcitation of photocatalyst to drive water splitting.

10 Fig. R2 A) The test figure of setting the indoor Xe lamp to 1000W/m² with AM1.5G filter. B) Photo of H₂ production test under indoor Xe lamp and the condensed water vapor under the quartz glass. C) The test photo of solar irradiance when the quartz glass is covered by condensed water vapor.

However, in the outdoor air-to-H₂ system, the Video S1 reveals that although a thin liquid film initially forms on the quartz glass surface, it rapidly dissipates under natural solar irradiation, resulting in negligible light-scattering effects. Under normal outdoor midday sunlight (1000 W/m²), the power received by the PTC layer surface is about **713 W/m²**, which is larger than indoor condition of **534.1 W/m²**. Even under an **average outdoor solar irradiance of 786 W/m²**, the effective irradiance absorbed by the PTC layer ($P_{sun, outdoor}$) reaches **569.1 W/m²**. This demonstrates that $P_{sun, outdoor}$ (**569.1 W/m²**) > $P_{sun, indoor}$ (**534.1 W/m²**). The superior solar energy absorption capacity of HHE NFMs under outdoor conditions is therefore confirmed.

Besides, most researchers intuitively assume that higher wind speeds in outdoor environments enhance convective heat transfer. However, in our outdoor air-to-H₂ system, the reactor body is constructed using aluminum (Al). This Al reactor undergoes

slight heating under sunlight, resulting a temperature rise to slightly above ambient temperature. Therefore, it can be inferred that the solar radiation absorbed by the reactor body is offset by external natural convective cooling, with negligible impact on the surface temperature of PTC layer.

5 Consequently, in the outdoor air-to-H₂ system, the HHE NFMs exhibit both higher energy absorption thus higher surface temperature of PTC layer (revised manuscript Fig. 4d). As demonstrated by our DFT calculations (revised manuscript Fig. 3h), the increased reaction temperature within the biphasic system lowers the energy barrier for H₂ formation, thereby accelerating the hydrogen evolution rate. Thus, from a thermodynamic perspective, the enhanced surface temperature of HHE NFMs under
10 outdoor conditions contributes significantly to the improvement in hydrogen production rates compared to indoor conditions.

2. Kinetic perspective

The kinetic enhancement of H₂ production rates in HHE NFMs primarily originates
15 from scaling effects. In indoor condition, a 4×10^{-4} m² HHE NFMs was enclosed within a glass reactor, whereas the outdoor system employed a 0.01 m² HHE NFMs array integrated into a 0.25 m² reactor. COMSOL simulations of the photothermal water release process for HHE NFMs of varying sizes revealed that moisture in the hygroscopic layer escapes not only from the top PTC layer but also from the sides of
20 the hygroscopic layer (revised manuscript Fig. 4e). However, as the HHE NFMs size increases, a greater proportion of vapor escapes through the PTC layer (revised manuscript Fig. 4f). This indicates that larger HHE NFMs dimensions enhance reactant (i.e. water molecules) availability by increasing vapor contact with catalysts on the PTC layer, thereby elevating vapor concentration in the PTC layer and kinetically enhancing
25 hydrogen production rates.

In summary, under natural outdoor sunlight irradiation, both thermodynamic and kinetic factors synergistically promote H₂ production. Consequently, the large-scale outdoor air-to-H₂ system achieves significantly higher hydrogen production rates. Additionally, we have refined the descriptions of H₂ production rate enhancements in

the main text and supplementary materials (revised manuscript page 9, line 16 to page 10, line 7, and revised Note S1 and Note S2) to better elucidate why the large-scale outdoor air-to-H₂ system exhibits superior performance.

We sincerely appreciate the reviewer's constructive feedback, which has enabled us to improve the clarity and rigor of our mechanistic explanations in the revised manuscript.

[Comment 3]: Page 18, line 10, mentions "The GGA + U method was used with U-J value of 4.0 for Ti 3d electrons". The .xyz file of the catalyst slab used for the computation may be included in the Electronic Supplement.

[Response]: Thank you for your kind suggestion! We have put the structural files of the catalyst into the Electronic Supplementary Materials for reference by both reviewers and readers.

[Comment 4]: Is numerical vibrational frequency computation was done on optimized catalyst slab, for estimating delta G.

[Response]: Thank you very much for your kind comments. We have done the numerical vibration frequency computation on optimized catalyst slab. The process was described in the calculation details (revised manuscript page 19, line 21-22).

[Comment 5]: If it is so which space group and arrangement of original unit cell was used, must be mentioned.

[Response]: We sincerely appreciate the reviewer's careful reading and constructive feedback. In response to these valuable comments, we have incorporated additional computational details in the revised manuscript (page 19, line 11-14) for enhanced clarity and reproducibility. The supplementary content is presented as follows:

'The model of anatase TiO₂ (tetragonal, space group I41/amd), with lattice parameters $a = b = 3.78 \text{ \AA}$, $c = 9.62 \text{ \AA}$, was obtained from open-access database The Materials Project. Then, a 4 layered (101) surface with 5×3 supercell was built based on XRD result, where the bottom layer was fixed during optimization.'

[Comment 6]: The claim that “The system can theoretically produce $\sim 4.5 \times 10^4$ GL (gigaliters) of hydrogen annually across 0.1% of global land areas.” is a very tall claim considering the amount of catalytic material and their production requirement.

5 **[Response]:** We appreciate the reviewer's kind suggestions and insights. The reviewer has a profound understanding of the annual hydrogen production, required materials, and preparation processes. In the current stage, this paper utilizes Pt/TiO₂ as the photocatalyst (Pt content is 1%wt), with a catalyst usage of 40g per square meter of HHE NFMs. Considering a land usage of 0.1%, which is approximately 1.49×10^{11} m²,
10 the required amount of catalyst is estimated to be around 5.9×10^6 tons of Pt/TiO₂. Furthermore, we integrate water-absorbing materials and Pt/TiO₂ using electrospinning technology to prepare HHE NFMs. As electrospinning is a relatively cost-effective and rapid technique, it enables us to swiftly fabricate HHE NFMs within a limited timeframe. Therefore, we concur with the reviewer's viewpoint that while
15 implementing our proposed air-to-H₂ system would demand lots of catalyst material with 0.1% of global land usage, the manufacturing process might be relatively straightforward.

Additionally, the statement in our original text, "The system can theoretically produce $\sim 4.5 \times 10^4$ GL (gigaliters) of hydrogen annually across 0.1% of global land areas," is
20 based on ideal conditions. Our quantitative projections and system-level analyses of global air-to-H₂ system deployment not only highlight its scalability but also provide intuitive feeling of the potential of H₂ production by using atmospheric moisture, which is a practical prospect for the future use of the system. At the same time, it can guide us to select a reasonable location (such as location with operatable relative humidity (RH)
25 and humidity) and let us clearly understand where to build the air-to-H₂ system we proposed to obtain better benefits. Furthermore, in the supporting materials, we have elaborated on and illustrated our calculation process for the reviewer and readers to reference (revised supplementary materials Note S3).

[Comment 7]: Hydrogen production happens only when sunlight is available. Cloudy, rainy days and nighttime will not produce hydrogen. Though moisture collection happens at night, there will be no hydrogen production.

[Response]: We sincerely appreciate the reviewer's thoughtful comments. Based on our experimental results, the system demonstrates optimal H₂ production efficiency under clear-sky conditions. While the HHE NFMs can still generate hydrogen under overcast conditions (with UV radiation and relatively weak visible light), the H₂ production rate decreases proportionally to the available photon flux (revised manuscript Fig. 4b). We concur with the reviewer's observation that H₂ production ceases during nighttime or rainy periods, a limitation shared by most solar-driven systems (photovoltaic^{2,3}, photoelectrochemical^{4,5}, solar driven seawater desalination^{6,7}). However, this work specifically focuses on enabling photothermal catalytic H₂ production under liquid water-free conditions by utilizing atmospheric moisture and natural sunlight. During nighttime or precipitation events, the elevated RH facilitates passive water harvesting from air, which subsequently enhances daytime photocatalytic activity. This dual-functional mechanism allows simultaneous energy and water production in arid regions.

While hydrogen production pauses at night, the system actively prepares for daytime operation through water storage. Furthermore, the radiative cooling capability during nighttime can enhance atmospheric water harvesting by dissipating heat to outer space, highlighting the unique advantage of our system compared to conventional photocatalytic^{8,9} or photo-electrocatalytic^{4,5} approaches.

We thank the reviewer for raising this important point, which has allowed us to clarify the system's operational strategy.

25

[Comment 8]: The solar-to-hydrogen (STH) efficiency of 0.1%–0.085% is reported in manuscript is very much lower than electrolysis-based hydrogen production, which is around 15–20%. It is not clear how the system may compete against electrolysis.

[Response]: We sincerely thank the reviewer for the detailed review. Here, we have

created a table to highlight the differences between our system and electrocatalytic approaches, along with a comparative analysis.

First of all, the proposed air-to-H₂ system is **not intended to compete with electrocatalysis but rather to complement it**. Large-scale centralized H₂ production near water sources and electricity supply may prioritize electrocatalysis, while our system provides decentralized solutions for water-scarce regions, contributing to sustainable hydrogen economies.

Secondly, as we emphasized throughout our work, the key strength of this system lies in its ability to produce H₂ without requiring liquid water and an electrical grid. By directly utilizing atmospheric moisture and solar energy, it overcomes the reliance of traditional electrolysis H₂ production on purified water/alkaline electrolytes and stable power supplies.^{10,11} Additionally, air-to-H₂ system is particularly suitable for off-grid scenarios (e.g., arid/semi-arid regions like the Sahara Desert, where nighttime humidity averages ~35% RH), where electrocatalytic systems struggle to operate.

Last but not least, our system produces **green H₂** with zero carbon emissions during operation, which is totally a passive way as no electricity is needed in the intermediate stage. In contrast, grid-dependent electrocatalytic hydrogen production may generate indirect emissions unless powered entirely by renewable energy.

We thank the reviewer for raising these critical points, which have allowed us to better articulate the unique value and applicability of our work.

[Comment 9]: The hygroscopic and photocatalytic materials (Pt/TiO₂@PVDF and MIL@PAN NFMs) need to maintain their performance over long-term outdoor exposure. Potential degradation due to UV exposure, environmental pollutants, or material fatigue over time could reduce efficiency. The cycling stability of the moisture adsorption-desorption process must be further tested for real-world applications.

[Response]: We sincerely thank the reviewer for this constructive comment. The cycling stability of HHE NFMs is indeed critical for the long-term operation of the system. To address this, we conducted additional extensive cyclic tests involving

prolonged illumination and repeated water adsorption/desorption processes.

1. Indoor cycling tests:

After each indoor H₂ production experiment, the HHE NFMs were exposed to ambient air for natural moisture adsorption for subsequent cycles of H₂ production. In fact, we performed multiple water adsorption and desorption processes (at least 10 times) using the same HHE NFMs, as shown in Fig. R3A. Meanwhile, repeated indoor cycling tests (Fig. R3B) demonstrated consistent H₂ production without noticeable performance decay (at least 8 times).

2. Outdoor cycling tests:

During the two-week outdoor H₂ production tests, no significant degradation in H₂ production performance was observed (revised manuscript Fig. 5d and revised supplementary materials Fig. S33). The observed fluctuations correlated with daily weather variations (e.g., cloud cover, humidity), confirming that the efficiency remained stable under operational conditions. After a long time of outdoor experiments, as shown in Fig. R3C, we took out the HHE NFMs after outdoor cycling reaction for independent testing, and found that its water adsorption and release capacity and H₂ production capacity did not significantly decline, which verifies that the outdoor environment has negligible influence on our HHE NFMs. We also added this data to the supporting materials for reference (revised supplementary materials Fig. S34).

3. Material Characterization

SEM analysis revealed intact honeycomb-like porous structures in the hygroscopic layer after prolonged outdoor exposure, with MIL-101(Cr) uniformly distributed on PAN nanofibers as shown in Fig. R3D, which verifies the structure stability of hygroscopic layer.

Meanwhile, the PTC layer maintained its structural integrity, with catalyst nanoparticles evenly dispersed on PVDF nanofibers after long time outdoor test as shown in Fig. R3E (added to revised supplementary materials Fig. S35). No detectable changes were observed in XRD patterns before and after testing, confirming material stability (Fig. R3F).

These comprehensive analyses validate the cycling stability of HHE NFMs under real-world operational conditions. We thank the reviewers for highlighting this important aspect, which has allowed us to strengthen the stability discussion in the revised manuscript page 10, line 13-17.

5

10

Fig. R3 A) Cycling stability of moisture adsorption–desorption of MIL@PAN and pure PAN NFMs at 25 °C and 90% RH (desorption at 100 °C). B) Time course of H₂ during the indoor cyclic testing of HHE NFMs. C) The water adsorption and desorption test and hydrogen production test after long time outdoor experiment. D) The SEM and EDS mapping of hygroscopic layer after long time outdoor experiment. E) The TEM photo of PTC layer after long time outdoor experiment. F) The XRD test result of PTC layer before and after long time outdoor experiment.

15

[Comment 10]: Is there any other material than platinum Pt and Au which can be used which is cheaper and abundant. A selection of these two elements must be justified.

[Response]: We sincerely appreciate the reviewer’s insightful comments. In the

proposed air-to-H₂ system, TiO₂ was selected as the photocatalyst for water vapor splitting due to its chemical stability, low cost, and environmental friendliness. However, the rapid electron-hole recombination in TiO₂ limits its photocatalytic activity and quantum efficiency.^{12,13} To address this, strategies such as metal loading
5 have been widely adopted to suppress recombination rates and enhance charge carrier mobility, thereby improving the overall H₂ evolution performance^{14,15}. Potential cocatalysts include noble metals (e.g., Pt, Au, Ag, Pd) and non-noble metals (e.g., Fe, Cu, Ni, Zn).¹⁵ While non-noble metals are more cost-effective, they exhibit certain limitations compared to noble metals.

10 First of all, Noble metal nanoparticles (e.g., Au, Pt) exhibit strong surface plasmon resonance (SPR) effect, which localize incident light and generate high-energy hot electrons.¹⁶ This significantly broadens the light-responsive range of TiO₂ and enhances photothermal conversion efficiency, while maintaining high stability against oxidation^{17,18}. These properties are critical for daytime water desorption and hydrogen
15 production in the air-to-H₂ system. Some non-noble metals (e.g., Cu, Co) may exhibit SPR effects but suffer from oxidation-induced deactivation under high-oxygen environments.¹⁹

Besides, noble metals (Pt, Au) possess high work functions, forming Schottky barriers at the TiO₂ interface.^{20,21} This effectively traps photogenerated electrons, suppresses
20 electron-hole recombination, and prolongs carrier lifetimes. Concurrently, noble metals provide high-density active sites to enhance hydrogen evolution rates.

Based on these considerations, Pt and Au were selected as cocatalysts to demonstrate the feasibility of the concept of this work. Their synergistic advantages, efficient electron trapping and SPR-driven photothermal effects, outperform the single-function
25 optimization achievable with non-noble metals. We acknowledge that further research on non-noble metals could improve material stability and balance performance with cost. Future work will explore such approaches to develop lower-cost hydrogen production solutions.

[Comment 11]: Page 17, the results from 3D numerical simulations must be presented.

[Response]: We sincerely appreciate the reviewer's thoughtful reminder. The purpose of this analysis is to illustrate the water vapor transport dynamics across HHE NFMs surfaces at varying scales (revised manuscript Fig. 4e). To clearly visualize these phenomena, we selected cross-sectional computational results for presentation in the original manuscript. As requested, the full 3D simulation results have been included in the revised supplementary materials (revised supplementary materials Fig. S30) for reference. Additionally, the 3D computational results are provided below for the reviewer's convenience.

Fig. R4 The velocity distribution of water vapor streamlines and concentration cloud maps of HHE NFMs (3D simulation results)

[Comment 12]: The system is a breakthrough in sustainable hydrogen production, but low efficiency, weather dependence, and scalability challenges must be addressed before commercial deployment.

[Response]: We sincerely thank the reviewer for this recognition of our work and the profound insights into the critical challenges. The issues of efficiency improvement, weather dependency, and scalability raised by the reviewers are indeed core obstacles that must be addressed for the commercialization of this technology. We fully acknowledge these limitations and will conduct further in-depth research on the challenges highlighted by the reviewer. For example, regarding H₂ production

efficiency, enhancement of the water adsorption capacity of the hygroscopic layer by incorporating hygroscopic salts into the MOF materials to form a hierarchical water-absorbing structure can be addressed. Additionally, photothermal catalysts with broader light absorption ranges can be developed to maximize solar energy utilization. To address weather dependency, more works can be done by adding heat energy storage systems to harvest excess heat during sunny periods, which could supply clean water to arid regions during cloudy or rainy conditions, even if H₂ production is paused. Regarding scalability, as mentioned in our previous response, we could focus on balancing performance and cost by utilizing more affordable and readily available catalysts to reduce overall system costs. We fully agree that achieving commercial deployment requires a better balance among efficiency, stability, and cost. The primary goal of this work is to provide a proof-of-concept and preliminary optimization pathways for this emerging field, while elucidating the effects of surface temperature and size effect on H₂ production rates. The reviewer's valuable feedback will significantly guide our future research, and we look forward to collaborating with the broader scientific community to advance this technology toward maturity.

[Comment 13]: Despite these limitations, the study presents a promising off-grid hydrogen production technology for arid and water-scarce regions, contributing to sustainable energy transitions.

[Response]: We sincerely thank the reviewer for the strong affirmation of our work's significance, particularly the recognition of the potential contributions to off-grid hydrogen production in arid, water-scarce regions and the sustainable energy transition. The highlighted limitations (e.g., efficiency and scalability challenges) align closely with our own priorities, and they serve as a strong impetus for further exploration of viable pathways toward practical implementation. Once again, we deeply appreciate your constructive feedback and encouragement.

Reviewer #2:

[General comment]: This manuscript presents a hydrogen production device that captures moisture from ambient air using MIL@PAN nanofiber membranes (NFMs) with a radiative cooler (Pt/TiO₂@PVDF) during nighttime, and the collected water is subsequently converted into hydrogen via a photothermal catalytic process using hygroscopic H₂ evolution nanofiber membranes (HHE NFMs) during daylight hours. While the research topic is interesting, the manuscript requires significant revisions based on the following comments.

[Response]: We sincerely thank the reviewer for reviewing our work and giving the valuable comments! We have addressed all the comments through further experimentation and have made revisions on the manuscript, as specified below. We hope our response will satisfy you.

[Comment 1]: Please provide the energy dispersive spectroscopy data and the specific surface area (m²/g) of the honeycomb-like structure composed of MIL@PAN.

[Response]: We sincerely thank the reviewer for these valuable suggestions. As recommended, the energy-dispersive spectroscopy (EDS) data and specific surface area analysis of the honeycomb-like MIL@PAN structure have been incorporated into the revised manuscript and supplementary materials. The EDS elemental mapping of the honeycomb structure is presented in Fig. R5.

The C and N elements from the PAN nanofibers are primarily distributed along the walls of the honeycomb structure, with minor amounts observed in the central regions, confirming the presence of nanofibers throughout the structure, consistent with the morphology shown in revised manuscript Fig. 2b. Similarly, the Cr, O, and F elements from MIL-101(Cr) are localized along the walls, with partial distribution in the central regions, further verifying the uniform dispersion of MIL-101(Cr) along the nanofibers and the formation of a highly porous architecture.

Brunauer–Emmett–Teller (BET) surface area measurements reveal that MIL@PAN exhibits a high specific surface area of **1790.8 m²/g**, suggesting the abundant accessible

surface area.

Fig. R5 SEM and elements mapping images of honeycomb-like structure

To address the reviewer's comments, we have made necessary changes and included
5 the updated results in the revised supplementary materials Fig. S3.

[Comment 2]: The authors should evaluate the energy balance of the hydrogen production device for both indoor and outdoor experiments.

Optical characteristics: The photothermal catalytic (PTC) top layer, consisting of
10 Pt/TiO₂@PVDF NFMs, absorbs supplied energy as heat, with an absorptance of 0.95 in the ultraviolet range and 0.7 in the visible-near infrared spectrum. It simultaneously achieves radiative cooling, with an emissivity of 0.95 in the mid-infrared range.

In the indoor experiment under AM 1.5G conditions (1,000 W/m²), the surface temperature of the hydrogen production device was maintained at 68 °C, and the
15 evaporation rate was 0.298 kg/m² h. The net radiative power absorbed by the device was calculated as 713 W/m² by integrating the product of absorptance and supplied power over the wavelength. Thermal radiation within the indoor environment is negligible. Latent heat absorption from water desorption was measured at 187 W/m². The remaining power, 526 W/m², corresponds to the sum of heat transferred to the
20 surrounding air and the endothermic heat of the hydrogen conversion process.

However, in the outdoor experiment, the surface temperature of the device increased to 82.2 °C. This raises questions about the energy balance, when the net absorbed power

under 1,000 W/m² solar flux is 713 W/m², while the thermal emission of the device is calculated as 858.9 W/m² at 82.2 °C. Additionally, water desorption from the porous MIL@PAN material absorbs latent heat at 2,260 kJ/kg-water, alongside the endothermic reaction heat of hydrogen production. This discrepancy suggests that the energy balance cannot be fully explained, even though the system is described as entirely insulated.

It is also noted that the outdoor solar flux is measured as 786 W/m², significantly lower than the indoor value of 1,000 W/m². This further complicates the energy balance, as the heat transfer coefficient of the system with outdoor air should logically be higher than that with indoor air. The authors should clarify these discrepancies and provide a detailed analysis of the energy balance under outdoor conditions.

[Response]: We sincerely appreciate the reviewer’s careful reading and constructive feedback. A rigorous energy conservation analysis of the HHE NFMs under outdoor solar conditions is indeed critical. Below, we provide a detailed analysis of the heat transfer processes involved. Here, we further divide the experiments under indoor and outdoor conditions (tests in open environments and tests in closed reactor) for separate calculations to verify energy conservation.

1) Indoor energy balance on the surface of HHE NFMs

In the indoor surface temperature test, the HHE NFMs was placed on an electronic balance for surface temperature test and water evaporation test, where the electronic balance has the glass wall on the surrounding to protect the HHE NFMs from being affected by the wind. Therefore, as we shown in revised manuscript Fig. 3C, the surface temperature of PTC layer increased from ~25 °C to ~58 °C within 10 min and finally arrived around 68 °C. Therefore, in energy balance calculation, we choose to use the surface temperature of 58 °C in the calculation, as under this temperature the evaporation rate is the highest. Therefore, the P_{sun} can be calculated by:

$$P_{sun} = \int_0^{\infty} [\epsilon_s(\lambda, \theta_{sun}) I_{AM1.5}(\lambda)] d\lambda$$

Here, P_{sun} is 718.9 W/m².

Under indoor condition, the passive cooling to the universe can be ignored. Therefore,

under the indoor condition,

$$P_{rad}(T_s) = \varepsilon \times \sigma \times (T_s^4 - T_{amb}^4) = 221.8 \text{ W/m}^2$$

The calculation of non-radiative cooling follows:

$$P_{non-radiative,indoor} = h_{non-radiative,indoor}(T_{s,indoor} - T_{amb,indoor})$$

5 Here, due to the surface area of HHE NFMs under indoor condition is $4 \times 10^{-4} \text{ m}^2$, the $h_{non-radiative,indoor} = 10.2 \text{ W}/(\text{m}^2 \cdot \text{K})$. Therefore, $P_{non-radiative,indoor} = 336.6 \text{ W/m}^2$.

According to the original manuscript, the evaporation rate under 1 sun illumination in the open indoor environment is approximately equivalent to that in the outdoor
10 environment.

$$P_{evaporation} = h_{evaporation} \dot{m} = 187 \text{ W/m}^2.$$

Therefore, under indoor open area,

$$P_{evaporation} + P_{non-radiative,indoor} + P_{rad,indoor} + P_{reaction,indoor} = 746.4 \text{ W/m}^2.$$

This value of 746.4 W/m² closely aligns with $P_{sun}=718.9 \text{ W/m}^2$. The extra energy loss
15 may come from slight inaccuracy of the measured indoor temperature, because under the simulated sunlight, the space inside the electronic balance also heats up, so $P_{non-radiative,indoor}$ may be overestimated here. **Therefore, the calculation results validate the energy balance at the PTC layer surface under indoor open area condition.**

20 **2) Outdoor energy balance on the surface of HHE NFMs under open area**

In the outdoor experiments, the HHE NFMs were placed in an **open plastic enclosure (without quartz glass shielding) for surface temperature test**, effectively operating under windless conditions due to the enclosure's walls. As shown in the temperature distribution data in the original manuscript, the maximum surface temperature reached
25 82°C. While this peak temperature was used in the initial radiative cooling power calculations in the comment, we acknowledge that using the average surface temperature of 76°C would yield a more representative energy balance. Furthermore, the solar irradiance during peak temperature conditions (noon) was measured as $\sim 970 \text{ W/m}^2$ (close to 1 sun intensity), whereas the value of 760 W/m^2 cited in the original

manuscript represents the daily average irradiance between 10:00 AM and 4:00 PM. Regarding the radiative cooling power calculation, after thorough literature review and validation, we have refined the energy balance framework. Under outdoor conditions, as shown in Fig. R6A, the energy conservation equation for the HHE NFMs surface should be expressed as:

$$P_{sun} = P_{rad}(T_s) - P_{atm}(T_{amb}) + P_{non-radiative} + P_{evaporation} + P_{reaction}$$

where P_{sun} , $P_{rad}(T_s)$, $P_{atm}(T_{amb})$, $P_{non-radiative}$, $P_{evaporation}$ and $P_{reaction}$ is the absorbed solar power, radiated power by the PTC layer, absorbed radiation power from the atmosphere, non-radiative cooling power, power absorbed by water evaporation process, and power for hydrogen production. T_s and T_{amb} is the temperature of PTC layer surface exposed to the sky and ambient temperature, which is 76°C and 35°C, respectively.

When the PTC layer is exposed to the sunlight, the P_{sun} can be calculated by¹:

$$P_{sun} = \int_0^{\infty} [\varepsilon_s(\lambda, \theta_{sun}) I_{AM1.5}(\lambda)] d\lambda$$

where $I_{AM1.5}(\lambda)$ represents the AM1.5 solar irradiance. When the clouds move to not block the sun, the light intensity is about 1sun, and the actual value is about 970 W/m². Therefore, the $P_{sun}=697.33$ W/m².

Due to the high infrared emissivity of PTC layer, it can spontaneously radiate heat through the atmospheric window. Under the condition of zero humidity, no clouds and clean air atmosphere (no aerosol, etc.), the theoretical value of surface radiation ($P_{rad,ideal}(T_s)$) to the sky should be calculated according to the following equation:²²

$$P_{rad,ideal}(T_s) = \int_{\Omega} \int_0^{\infty} [\varepsilon_s(\lambda, \theta) I_{BB}(\lambda, T_s)] d\lambda \cos\theta d\Omega$$

where $\int_{\Omega} d\Omega = \int_0^{\pi/2} \sin\theta d\theta \int_0^{2\pi} d\phi$ is the hemisphere angular integral, and $\varepsilon_s(\lambda, \theta)$ is the spectral and angular emissivity of the PTC layer. $I_{BB}(\lambda, T_s) = \frac{2hc^2}{\lambda^5} \frac{1}{e^{\frac{hc}{\lambda k_B T_s}} - 1}$ indicates the spectral radiance of a blackbody, where h is Planck's constant, c is the speed of light in vacuum, and k_B is the Boltzmann constant.

In this theoretical condition, the theoretical maximum radiation power of our PTC layer is:

$$P_{rad,ideal}(T_s) = 499.6 \text{ W/m}^2.$$

However, in practice, the atmospheric humidity, aerosol content, and cloud coverage significantly influence radiative cooling power. First, water vapor exhibits strong absorption of infrared radiation beyond wavelength of 16 μm .²³ As humidity increases, the 16–22 μm radiative window becomes substantially attenuated or even closed. Specifically, when environmental RH rises from 20% to 100%, the peak atmospheric transmittance in the primary radiative window (8–13 μm) decreases from 90% to 60%. Compared to dry conditions, most secondary radiative windows either disappear (e.g., 16–22 μm) or narrow (e.g., 2.5–5 μm) under high humidity.^{22,24} Additionally, atmospheric aerosols (suspended solid/liquid particles) induce Rayleigh and Mie scattering across both solar and thermal infrared spectra.^{25,26} Higher humidity exacerbates aerosol scattering, leading to increased infrared attenuation. In Hong Kong, a coastal humid city where our outdoor experiments were conducted, summer aerosols are particularly pronounced. At typical experimental conditions (average RH = 60~80%), aerosol scattering suppresses the atmospheric transmittance in the primary window by up to 20%.²³

To conservatively account for the combined effects of humidity and aerosols on radiative cooling under cloudless skies, we introduced a correction factor R in our calculation.²³ Generally, the smaller the R is, the greater the impact of the environment on the radiation cooling effect. In order to simplify the calculation and comprehensively consider the influence of RH (60%-80% RH) and aerosol effect in the air, we adopted a conservative value of $R = 0.8$ in our calculations. Therefore, the actual radiative cooling power can thus be expressed as:

$$P_{rad,T_s} = R \times P_{rad,ideal} = 0.8 \times 499.6 = 399.68 \text{ W/m}^2.$$

The power calculation of PTC layer absorbing radiation from the environment without cloud occlusion follows:²²

$$P_{atm,noncloud}(T_{amb}) = \int_{\Omega} \varepsilon_s(\lambda, \theta) \int_0^{\infty} [\varepsilon_{atm}(\lambda, \theta) I_{BB}(\lambda, T_{abm})] d\lambda \cos\theta d\Omega$$

where $\varepsilon_{atm}(\lambda, \theta) = 1 - t(\lambda)^{1/\cos\theta}$, is the emissivity of atmosphere, $t(\lambda)$ is the

transmittance of the atmosphere.

Furthermore, the presence of clouds significantly impacts radiative cooling performance.²⁷ Clouds often behave as near-blackbody emitters in the mid-infrared spectrum, becoming opaque to outgoing thermal radiation. This enhances atmospheric re-radiation and blocks heat dissipation from the Earth's surface, thereby reducing radiative cooling power. As demonstrated by Zhao et al.²⁸, the decline in radiative cooling power is proportional to the increased cloud cover duration as compared to clear-sky conditions. Besides, the presence of cloud layers leads to increased atmospheric re-radiation, resulting in a higher $P_{atm}(T_{amb})$. Taking June 2024 as an example (when outdoor experiments were conducted), Hong Kong experienced predominantly cloudy conditions with an average cloud cover of 86% (f_c) (The cloud cover data comes from Hong Kong Observatory). Even on clear days, abundant floating cloud layers persisted, which reduced the radiative cooling power of the PTC layer and increased the environmental radiation incident on the PTC. Here, we do not account for the cooling power reduction caused by cloud cover but focus solely on its effect on enhancing P_{atm} .

In the case of cloud cover, the received ambient radiation $P_{atm, undercloud}$ is:²³

$$P_{atm, undercloud} = \int_{\Omega} \varepsilon_s(\lambda, \theta) \int_0^{\infty} [\varepsilon_{cloud}(\lambda, \theta) I_{BB}(\lambda, T_{amb}) \tau_{atm}(\lambda, RH, T_{amb}, \theta)] d\lambda \cos\theta d\Omega$$

where $\varepsilon_{cloud}(\lambda, \theta)$ is the emissivity of cloud, here we take $\varepsilon_{cloud}(\lambda, \theta) = 1$.

$\tau_{atm}(\lambda, RH, T_{amb}, \theta)$ is the emissivity of atmosphere under cloud. Therefore,

$$P_{atm}(T_{amb}) = f_c P_{atm, undercloud} + (1 - f_c) P_{atm, noncloud} = 257.3 \text{ W/m}^2.$$

The calculation of non-radiative cooling follows:

$$P_{non-radiative} = h_{non-radiative} (T_s - T_{amb})$$

Where $h_{non-radiative}$ represents non-radiative heat transfer coefficient. In this setup, thermal insulation materials were applied to the sidewalls, and a windbreak layer was implemented to minimize additional heat loss caused by wind. Consequently, heat transfer in this configuration is dominated by natural convection. Following established natural convection heat transfer correlations, the calculated non-radiative heat transfer

coefficient is $h_{non-radiative} = 8.0 \text{ W}/(\text{m}^2 \cdot \text{K})$. Therefore,

$$P_{non-radiative} = h_{non-radiative}(T_s - T_{amb}) = 328 \text{ W/m}^2$$

According to the original manuscript, the evaporation rate under 1 sun illumination in the open indoor environment is approximately equivalent to that in the outdoor environment.

$$P_{evaporation} = h_{evaporation}\dot{m} = 187 \text{ W/m}^2.$$

It is mentioned in the draft that under outdoor conditions, the STH in sunny days is 0.1%. Therefore,

$$P_{reaction} = STH \cdot P_{AM1.5} = 1 \text{ W/m}^2.$$

As a result,

$$P_{rad}(T_s) - P_{atm}(T_{amb}) + P_{non-radiative} + P_{evaporation} + P_{reaction} = 658.38 \text{ W/m}^2$$

This value of 658.38 W/m² closely aligns with $P_{sun}=697.33 \text{ W/m}^2$. The residual energy discrepancy likely arises from unaccounted energy losses. **This calculation results validates the energy balance at the PTC layer surface under outdoor open area condition.**

3) Outdoor energy balance on the surface of HHE NFMs in the large scale reactor

When the HHE NFMs were placed in the outdoor large scale reactor, **the quartz glass window can block most of the radiative cooling into the universe under sunny day**, due to its low transmittance in atmospheric radiation window, as shown in Fig. R6C.

However, its transmittance is extremely high in the UV and visible light range, so the light intensity here is still calculated based on natural light:

$$P_{sun} = \int_0^{\infty} [\varepsilon_s(\lambda, \theta_{sun}) I_{AM1.5}(\lambda)] d\lambda$$

Here, we take the midday outdoor natural sunlight as example, because the surface temperature was tested under midday. So, $P_{sun} = 697.33 \text{ W/m}^2$.

When HHE NFMs are enclosed in a reactor, the calculation of their radiative heat dissipation should follow:

$$P_{rad}(T_s) = \varepsilon \times \sigma \times (T_{s,outdoor}^4 - T_{amb,outdoor}^4)$$

Here, as it is difficult to get the temperature inside the reactor, the surface temperature of PTC layer and ambient temperature also taken as 76°C and 35 °C. Therefore,

$$P_{rad}(T_s) = 314.3 \text{ W/m}^2.$$

Most researchers intuitively assume that higher wind speeds in outdoor environments enhance convective heat transfer. However, in our outdoor air-to-H₂ system, the reactor body is constructed of aluminum (Al). This Al reactor undergoes slight heating under sunlight (moderate temperature rises, slightly above ambient temperature, here taken as 35°C). Therefore, it can be inferred that the solar radiation absorbed by the reactor body is offset by external natural convective cooling, with negligible impact on the surface temperature of PTC layer. For energy analysis focusing solely on the PTC layer, since the HHE NFMs are enclosed within reactors in both indoor and outdoor configurations, natural convective heat transfer occurs at the PTC layer surface in all cases.

$$P_{non-radiative} = h_{non-radiative}(T_s - T_{amb}) = 328 \text{ W/m}^2.$$

Furthermore, when the HHE NFM was placed in a reactor (either under outdoor conditions or indoor conditions), the evaporation rate of the HHE NFM was greatly reduced in a closed environment as shown in Fig. R6B (revised supplementary materials Fig. S34). The obviously lower evaporation rate in the closed system is mainly because (i) the significantly higher relative humidity in the closed system inhibits evaporation,^{29,30} and (ii) the convection in the closed system is much less than that in the open system.^{31,32}

Therefore, the evaporation power was calculated as:

$$P_{evaporation} = h_{evaporation}\dot{m} = 32.5 \text{ W/m}^2.$$

It is mentioned in the manuscript that under outdoor conditions, the STH in sunny days is 0.1%. Therefore,

$$P_{reaction} = STH \cdot P_{AM1.5} = 1 \text{ W/m}^2.$$

Therefore, in outdoor large scale reactor,

$$P_{evaporation} + P_{non-radiative,indoor} + P_{rad,indoor} + P_{reaction,indoor} = 675.8 \text{ W/m}^2$$

This value of 675.8 W/m² closely aligns with P_{sun}=697.3 W/m² within the outdoor large scale reactor. The residual energy discrepancy likely arises from unaccounted energy losses.

Therefore, based on our calculations, HHE NEMs verify energy conservation in

both outdoor and indoor conditions.

Fig. R6 A) Schematic diagram and heat transfer processes of PTC layer under outdoor open area. B) The water adsorption and desorption test and hydrogen production test with the humidity, temperature, and solar radiation intensity under outdoor fixed reactor condition. C) The transmittance of quartz glass window.

[Comment 3]: The numerical simulation conducted using COMSOL Multiphysics lacks sufficient details. Specifically, when desorbed water molecules undergo gas-phase water splitting for hydrogen production via a photothermal catalytic process in the indoor experiment, this endothermic reaction likely impacts the overall energy balance, which was not explained in the manuscript. Please discuss the effect of this phenomenon. Additionally, more comprehensive information regarding the initial and boundary conditions, as well as the analytical scheme used, should be included.

[Response]: We sincerely thank the reviewer for these insightful comments. As noted, the photothermal catalytic water vapor splitting process is endothermic, and the absorbed energy would indeed influence the overall energy flow in simulations. However, as calculated in our previous response, the energy consumed by the H_2

evolution reaction accounts for only 0.10% of the total solar input (approximately 1 W/m²), which is negligible compared to other energy pathways. Given the complexity of coupling reaction kinetics and phase-transport phenomena in the hydrogen production process, fully integrating these effects would significantly increase computational demands. In this COMSOL study, our primary objective was to validate that increasing the size of HHE NFMs (while maintaining a constant height) enhances the water vapor transport rate from the PTC layer, thereby supplying more reactants (i.e. water molecules) for the photothermal catalytic reaction. To maintain computational tractability, we simplified the model by focusing on the evaporation dynamics under illumination.

As suggested, we have clarified this rationale in the revised manuscript (page 17, line 19-28) and expanded the description of initial/boundary conditions in the revised supplementary materials Table S3. In addition, we have added a more detailed simulation scheme in the revised manuscript for the reference of reviewers (page 17, line 16-19 and page 18, line10-15).

[Comment 4]: Why was MIL-101(Cr) used for water harvesting from ambient air? The moisture sorption performance (both kinetics and capacity) of MIL-101(Cr) is significantly lower than that of other materials under low humidity conditions. While the radiative cooler was utilized to enhance the moisture sorption performance of MIL-101(Cr) under low-humidity conditions, it also reduces net energy absorption during daylight hours due to thermal radiation. Alternatively, materials with higher sorption capacity at low humidity could be considered. Please discuss the advantages of combining the MIL@PAN layer with the radiative cooler in this context.

[Response]: We sincerely thank the reviewer for the constructive feedback. In atmospheric water harvesting systems, MIL-101(Cr) has been widely adopted due to its high porosity, exceptional adsorption capacity, and broad operational humidity range.^{33,34} The rationale for selecting MIL-101(Cr) as the hygroscopic material in this work is as follows:

(1) MIL-101(Cr) exhibits ultrahigh water adsorption capacity (1.2–1.4 g/g) under moderate to high humidity conditions (RH > 60%),^{34,35} significantly outperforming silica gel (0.3–0.4 g/g) and zeolites (0.2–0.3 g/g).³⁶ Its rapid adsorption kinetics further ensure efficient water vapor capture.³⁷

5 (2) The material demonstrates outstanding regeneration performance, requiring a low desorption temperature of 60~80°C,³⁴ which aligns well with solar thermal energy availability.

(3) MIL-101(Cr) maintains structural stability under operational conditions and long-term stable water adsorption capacity.³⁸

10 We fully acknowledge the limitation raised by the reviewer regarding MIL-101(Cr)'s reduced efficacy under low-humidity conditions (RH < 30%), where its adsorption capacity drops to 0.1–0.2 g/g due to its larger pore size.³⁷ To address this, we have included a comparative Table R1 to analyze the advantages and limitations of various hygroscopic materials under low-humidity conditions.

15 Table R1 The comparison between HHE NFMs and other atmospheric water harvesting materials.

Materials	Water uptake		Advantages	Disadvantages
	capacity, RH=20%	(g/g)		
MIL-101(Cr)	0-0.15 ^{34,35,39}		Wide moisture absorption range, large capacity under high humidity, ³⁹ high cycle stability and low regeneration energy consumption ^{38,39}	Water uptake capacity is relatively low under low humidity conditions
MOF-801 ⁴⁰⁻⁴²	0.25-0.3 ⁴⁰⁻⁴²		Ultra microporous structure (pore size 0.6 nm), designed for low humidity conditions, fast adsorption kinetics under low humidity conditions ⁴²	The synthesis cost is high, and the adsorption capacity is limited under high humidity conditions.

CAU-10-H ⁴³⁻⁴⁵	0.2-0.25 ⁴³⁻⁴⁵	Modified with amino group, the material has strong hygroscopic capacity (0.1g/g) under extreme low humidity conditions (RH=10%) ^{44,45}	Poor water uptake capacity under high humidity conditions and high water desorption temperature (80-100 °C) ^{45,46}
Hygroscopic salt (taking LiCl as an example) ³⁶	0.3-0.5 ^{36,47}	High water absorption capacity at low humidity ⁴⁷	Easy deliquescence, poor circulation stability, and the deliquescent solution is corrosive to the equipment and difficult to control. ^{36,47}
MIL@PAN (This work)	0.1-0.2	PAN fiber enhances hydrophilicity, grading holes accelerate mass transfer, and radiation cooling can improve local humidity to enhance water adsorption under low humidity conditions	Composite materials

As demonstrated in the table, most hygroscopic adsorbents or absorbents with strong water uptake capabilities under low-humidity conditions struggle to maintain high adsorption capacities at elevated humidity levels. Notably, hygroscopic salts exhibit strong water adsorption across both low- and high-humidity ranges⁵¹. However, they are prone to deliquescence after moisture absorption, leading to severe cycling instability. The resulting liquid salt solutions are difficult to control and may cause equipment corrosion⁴⁷. Furthermore, our statistical analysis reveals that nighttime average humidity exceeds 40% across most global land areas. This ensures that the operational environment for our air-to-H₂ system predominantly falls within the

moderate-to-high humidity regime ($RH \geq 40\%$), where the HHE NFMs can achieve atmospheric water harvesting even in relatively low-humidity environments through radiative cooling.

Therefore, in this work, we propose integrating MIL-101(Cr) with radiative cooling to develop the MIL@PAN layer, which offers the following advantages:

(1) Multi-layer honeycomb-like structure

The electrospinning technique ensures uniform dispersion of MIL-101(Cr) nanoparticles on PAN nanofibers, preventing particle agglomeration and preserving their high specific surface area. Additionally, the introduction of PAN enhances hydrophilicity and promoting water molecule adsorption and diffusion under low-humidity conditions.

(2) Radiative cooling enhanced atmospheric water harvesting

By coupling the MIL@PAN layer with the PTC layer, radiative cooling is achieved. During nighttime, the HHE NFMs surface temperature is reduced through radiative cooling to outer space, thereby increasing the effective surface relative humidity (RH). This effect is particularly critical in low-humidity environments, elevating ambient RH from suboptimal levels (e.g., 30% RH) to values favorable for adsorption (e.g., 45% RH), significantly enhancing water uptake capacity of MIL@PAN.

(3) Optimized energy utilization

Although the high emissivity of PTC layer may cause partial energy loss during daytime, the remaining energy is sufficient to drive the desorption process in the hygroscopic layer. However, due to the quartz glass covering above the HHE NFMs, the daytime radiative cooling will be decreased, which will minimize the energy loss by radiative cooling during the daytime. This mechanism also establishes a stable, continuous vapor flux under outdoor conditions to sustain the photothermal H_2 evolution reaction.

(4) Improved system stability and scalability

The PAN matrix acts as a flexible substrate, enhancing mechanical stability. Furthermore, the MIL@PAN composite can be mass-produced via electrospinning, facilitating large-scale manufacturing and practical deployment of the system.

[Comment 5]: It is recommended to describe the performance of the hydrogen production device under outdoor conditions from an energy perspective. For example, the energy consumption required to produce a unit mass of hydrogen (J/g or J/mol) could be calculated and compared with other studies.

[Response]: We sincerely appreciate the reviewer’s insightful suggestion. In this study, we employed the solar-to-H₂ (STH) efficiency to quantify the conversion efficiency from solar energy to H₂ energy in the photocatalytic process. And the STH efficiency is inversely proportional to the ‘energy consumption required to produce a unit mass of hydrogen’. Following the recommendation of reviewer, we have included a comprehensive comparison between our air-to-H₂ system with other photocatalytic H₂ production technologies as shown in Table R2 (revised supplementary materials Table S1).

Table R2 The comparison of air-to-H₂ system with other photocatalytic H₂ production technologies.

Photocatalyst	STH Efficiency	Solar energy required to produce one mol of hydrogen (J/mol)	Feedstock	Light source	Ref
Pt/TiO ₂	0.10%	2.3×10 ⁸	Vaper harvested for the air (without using sacrificial agent)	Natural sunlight	This work

			Vapor		
RhCrO _x -Al:SrTiO ₃	0.09%	2.63×10 ⁸	evaporated from pure liquid water	Natural sunlight	48
CoOOH/Rh loaded SrTiO ₃ :Al coated with TiO _x	0.4%	5.93×10 ⁷	Pure vapor feeding	300 W Xe lamp AM 1.5G	49
Pt/TiO ₂	0.002%	1.19×10 ¹⁰	Liquid seawater	500 W Hg lamp	50
Ru-modified SrTiO ₃ : La, Rh/Au colloid (40 wt%)/BiVO ₄ :Mo	0.1%	2.37×10 ⁸	Bulk pure water	300 W Xe lamp	51
g-C ₃ N ₄ /ITO/Co-BiVO ₄	0.028%	8.46×10 ⁸	Bulk pure water	300 W Xe lamp AM 1.5 G	52
Rh/CrO _x Au/CoO _x modified PbTiO ₃ /BiVO ₄	0.053%	4.47×10 ⁸	Liquid water containing Fe ²⁺ /Fe ³⁺	300 W Xe lamp λ > 420 nm	53
BiVO ₄ /Au/CdS	0.054%	4.39×10 ⁸	Bulk pure water	300W Xe lamp AM 1.5 G	54
CoO _x /CdTe/V-In ₂ S ₃	1.31%	1.87×10 ⁷	Bulk pure water	300 W Xe lamp λ ≥ 300 nm	55
Pt/CoO _x /ZnIn ₂ S ₄ /WO ₃	1.52%	1.56×10 ⁷	Bulk pure water	300 W Xe lamp AM 1.5 G	56
			Methanol		
Cu/TiO ₂	0.002%	1.19×10 ¹⁰	aqueous solution	Natural sunlight	57
			Methanol		
Cu/TiO ₂	0.074%	3.2×10 ⁸	aqueous solution	300W Xe AM1.5G	57

PdPSA-CdS	0.1%	2.3×10^8	Ethanol aqueous solution	365 mW LED $\lambda=420$ nm	58
Pt-IrO ₂ /3D-g-C ₃ N ₄	0.06%	3.95×10^8	Bulk pure water	300 W Xe lamp $\lambda \geq 420$ nm	59
Ni-Ag/CN _x	0.02%	1.19×10^9	Ethanol aqueous solution	300 W Xe lamp AM 1.5 G	60

[Comment 6]: The hydrogen production device used for outdoor experiments appears to be insulated. The authors should discuss the insulation materials utilized and calculate the overall thermal resistances of the device in each direction.

5 **[Response]:** We sincerely thank the reviewer for raising this important consideration. In our outdoor large-scale reactor setup, a thin foam layer (0.5 cm thickness) was applied to the periphery of the reactor. However, this foam serves as a protective cushioning layer to prevent quartz glass breakage during transportation and handling, rather than functioning as a thermal insulation barrier.

10 The reactor system comprises two distinct components: reactor body and reactor support frame. The reactor body is composed of aluminum alloy reaction baseplate, quartz glass window, stainless steel spacers, and rubber gaskets. And the reactor frame is composed of extruded aluminum alloy profiles. Crucially, the aluminum reactor body remains uninsulated and is directly exposed to ambient air. Therefore, thermal
15 resistance beneath the HHE NFMs is around $8.77 \times 10^{-5} \text{ m}^2 \cdot \text{K/W}$ (attributed to the 2 cm-thick aluminum plate) and the thermal resistance of the reactor periphery is around $0.143 \text{ m}^2 \cdot \text{K/W}$ (resulting from the combined 3 cm aluminum casing and 0.5 cm rubber insulation layer).

20 **[Comment 7]:** Although the net energy absorbed by the hydrogen production device under outdoor conditions is lower than that under indoor conditions, how can the

hydrogen production rate increase? Please consider the following factors:

Outdoor experiments: Solar flux of 786 W/m², thermal emission from the surface (absorptance/emissivity of 0.95), and a higher heat transfer coefficient due to windy conditions.

5 Indoor experiments: Power flux of 1,000 W/m² from the solar simulator, regulated thermal emission due to the indoor environment, and a lower heat transfer coefficient. When the energy flux is high and a high surface temperature is achieved, improved efficiency can be explained thermodynamically using the Gibbs energy relation. However, this does not align with the lower energy environment described in outdoor
10 experiments.

[Response]: We sincerely thank the reviewers for these insightful comments. While the reasons for the enhanced hydrogen production rate were mentioned on the original manuscript, we acknowledge that our explanation may not have been sufficiently clear. Here, we provide a more detailed analysis to clarify why the hydrogen production rate
15 of the air-to-H₂ system under outdoor conditions is 6 times higher than that observed in indoor conditions. We hypothesize that this enhancement arises from the combined synergistic **thermodynamic** and **kinetic** effects.

1) Thermodynamic perspective:

A comprehensive energy analysis of the PTC layer surface reveals that under outdoor
20 conditions, the upper surface absorbs significantly more solar energy, leading to elevated surface temperatures. This thermal effect thermodynamically favors the hydrogen evolution reaction. The specific mechanisms are elaborated as follows:

(1) Light scattering by condensed liquid films on glass reactor surfaces

The solar energy absorbed by the surface (P_{sun}) of HHE NFMs under outdoor
25 conditions and indoor conditions can be calculated by:²²

$$P_{sun} = \int_0^{\infty} [\varepsilon_s(\lambda, \theta_{sun}) I_{AM1.5}(\lambda)] d\lambda$$

where $\varepsilon_s(\lambda, \theta_{sun})$ represents the absorptance/emittance of the surface of HHE NFMs, which is PTC layer, and $I_{AM1.5}(\lambda)$ represents the AM1.5 solar irradiance. In the indoor hydrogen production tests, the HHE NFMs were encapsulated within a quartz glass

reactor to maintain a sealed environment and prevent hydrogen leakage. Theoretically, the P_{sun} calculation should be based on the standard simulated solar radiation of 1000 W/m². However, under indoor conditions, the temperature of the reactor's top glass window and sidewalls remained close to ambient temperature (~25°C), and water vapor released during evaporation condensed on these surfaces, forming liquid droplets beneath the glass window. Optical characterization reveals that these condensate droplets induce notable light scattering and absorption. Consequently, the actual photon flux reaching the PTC layer was reduced. To validate this conclusion, we set the simulated solar irradiance to 1000 W/m², as shown in the Fig. R7A. Fig. R7B shows that, under illumination, a liquid film rapidly formed and persisted on the underside of the quartz glass cover plate. To evaluate the impact of this liquid film on light intensity, a quartz plate with an analogous liquid film was positioned above a radiometer at the same height as the reactor's quartz cover plate. The results from Fig. R7C demonstrated a significant irradiance reduction due to light scattering by the liquid film, decreasing from 1000 W/m² to **745 W/m²**. Consequently, the effective solar irradiance absorbed by the HHE NFMs surface within the reactor under indoor conditions ($P_{sun, indoor}$) was calculated as **534.1 W/m²**. The presence of the liquid film in the enclosed reactor thus reduced the solar irradiance absorbed by the PTC layer, leading to a lower surface temperature compared to the open-environment measurement under indoor conditions.

Fig. R7 A) The test figure of setting the indoor Xe lamp to 1000W/m² with AM1.5G filter. B) Photo of H₂ production test under indoor Xe lamp and the condensed water vapor under the quartz glass. C) The test photo of solar irradiance when the quartz glass is covered by condensed water vapor.

However, in the outdoor air-to-H₂ system, the Supplementary Video 1 reveals that

although a thin liquid film initially forms on the quartz glass surface, it rapidly dissipates under natural solar irradiation, resulting in negligible light-scattering effects.

Under normal outdoor midday sunlight (1000 W/m²), the power received by the PTC layer surface is about 713 W/m², which is larger than indoor condition of

5 **534.1 W/m².** Even under an average outdoor solar irradiance of 786 W/m², the effective irradiance absorbed by the PTC layer ($P_{sun, outdoor}$) reaches 569.1 W/m². This demonstrates that $P_{sun, outdoor}$ (**569.1 W/m²**) > $P_{sun, indoor}$ (**534.1 W/m²**). The superior solar energy absorption of HHE NFMs under outdoor conditions is therefore confirmed.

Besides, most researchers intuitively assume that higher wind speeds in outdoor
10 environments enhance convective heat transfer. However, in our outdoor air-to-H₂ system, the reactor body is constructed using aluminum (Al). This Al reactor undergoes slight heating under sunlight, resulting a temperature rise to slightly above ambient temperature. Therefore, it can be inferred that the solar radiation absorbed by the reactor body is offset by external natural convective cooling, with negligible impact on the
15 surface temperature of PTC layer.

Consequently, in the outdoor air-to-H₂ system, the HHE NFMs exhibit both higher solar energy absorption and higher surface temperature of PTC layer (revised manuscript Fig. 4d). As demonstrated by our DFT calculations (revised manuscript Fig. 3h), the increased reaction temperature within the biphasic system lowers the energy barrier for
20 H₂ formation, thereby accelerating the hydrogen evolution rate. Thus, from a thermodynamic perspective, the enhanced surface temperature of HHE NFMs under outdoor conditions contributes significantly to the improvement in hydrogen production rates compare to the indoor conditions.

2) Kinetic perspective

25 The kinetic enhancement of hydrogen production rates in HHE NFMs primarily originates from scaling effects. In indoor condition, a 4×10⁻⁴ m² HHE NFMs was enclosed within a glass reactor, whereas the outdoor system employed a 0.01 m² HHE NFMs array integrated into a 0.25 m² reactor. COMSOL simulations of the photothermal water release process for HHE NFMs of varying sizes revealed that

moisture in the hygroscopic layer escapes not only from the top PTC layer but also from the sides of the hygroscopic layer (revised manuscript Fig. 4e). However, as the HHE NFMs size increases, a greater proportion of vapor escapes through the PTC layer (revised manuscript Fig. 4f). This indicates that larger HHE NFMs dimensions enhance reactant (i.e. water molecules) availability by increasing vapor contact with catalysts on the PTC layer, thereby elevating vapor concentration in the PTC layer and kinetically enhancing hydrogen production rates.

In summary, under natural outdoor sunlight irradiation, both thermodynamic and kinetic factors synergistically promote H₂ production. Consequently, the large-scale outdoor air-to-H₂ system achieves significantly higher H₂ production rates. Additionally, we have refined the descriptions of H₂ production rate enhancements in the revised manuscript and revised supplementary materials (revised manuscript page 9, line 16 to page 10, line 7, and supplementary materials Note S1) to better elucidate why the large-scale outdoor air-to-H₂ system exhibits superior performance.

[Comment 8]: One potential reason for higher efficiency under outdoor conditions could relate to the wavelength distribution and intensity of the solar simulator used in the indoor tests. Please provide specifications for the solar simulator, including its wavelength range and intensity distribution.

[Response]: We sincerely thank the reviewer for the kind comments and suggestion. The Xenon lamp system (Newport Model 67005 with Power Supply 69911) equipped with an AM1.5G filter was employed. Fig.R8 presents the spectral irradiance of Xenon lamp we used and our measured spectral distribution when using the AM1.5G filter. Notably, comparative analysis reveals that, in the ultraviolet and visible range (625-800 nm), the filtered xenon lamp exhibits higher irradiance than natural sunlight. However, in 450-625 nm of visible light, filtered xenon lamp exhibits lower irradiance. While these spectral mismatches were shown in the figure, these differences alone cannot fully explain the increase in the performance observed under outdoor conditions.

Fig. R8 A) The spectral irradiance of various Arc Lamps. (Here the 628 300W Xe OZONE FREE is what we used). B) The spectral irradiance of natural sunlight and 300W Xe light with AM1.5g filter.

Reviewer #3:

[General comment]: The author discusses a method that enhances atmospheric water harvesting using radiative cooling and achieves hydrogen production via photothermal and photocatalytic water splitting in the vapor phase. I have three main concerns regarding this paper:

[Response]: We sincerely appreciate the reviewer for the time and valuable feedback on our work. We explained our experimental process and results in detail, and made careful revisions based on the reviewer's comments, as specified below. We hope our response will sufficiently address the concerns.

10

[Comment 1]: Regarding Radiative Cooling

The author states that the PTC layer has a high emissivity, enabling radiative cooling at night. Figure 2g shows that the surface temperature of the HHE NMFs is approximately 6°C lower than the ambient air temperature. However, the author does not explicitly mention that this result was obtained with the HHE NMFs directly exposed to air—this detail can only be found in Supplementary Figure S10. In subsequent system experiments, the PTC layer and the water-absorbing layer were placed under a transparent window, as can be seen in the video materials. Under these conditions, the radiative cooling effect should be significantly weaker than what is presented in Figure 2g. Similarly, I suspect that the infrared temperature measurements shown in Figure 4d were also taken without the transparent window, although the author does not clarify this. These results, therefore, do not reflect the conditions with a transparent window. Given these concerns, the data presented in the earlier sections of the paper do not fully support the analysis and discussion of the system operation results in the later sections.

[Response]: We sincerely thank the reviewer for the careful reading and constructive comments, and we apologize for any misunderstandings caused by insufficient clarity in the original manuscript.

While we mentioned the radiative cooling-enhanced atmospheric water harvesting process during nighttime in the text, we did not explicitly emphasize the experimental

protocol. The requirement for exposing the HHE NFMs to the night sky for radiative cooling was detailed in the Methods section under ‘Outdoor radiative cooling-enhanced temperature and moisture adsorption measurement’ (page 14, lines 3–10). In these experiments, the HHE NFMs were covered with a polyethylene (PE) film and equipped with K-type thermocouples to record temperature changes, as shown in revised supplementary materials Fig. S11. The PE film was selected for its high infrared transmittance (ensuring unimpeded radiative cooling at night) and its ability to minimize wind-induced convective heat transfer, which is a standard practice in radiative cooling studies.¹ Experimental results demonstrated that the high emissivity of the PTC layer reduced the surface temperature by an average of 6°C compared to samples without the PTC layer under identical conditions.

We further clarify that subsequent experiments investigating radiative cooling enhanced atmospheric water harvesting are detailed in the Methods section (Page 14, lines 16–20). In these experiments, two identical HHE NFMs were placed on electronic balances to monitor nighttime mass changes. For the radiative cooling-enhanced group (left panel of Fig. S11), the electronic balance surface was covered with a PE film to minimize wind interference. In contrast, the control group without radiative cooling (right panel of Fig. S11) was enclosed within an acrylic chamber wrapped with aluminum foil to block thermal radiation to outer space. Both of them were left enough gaps for air to get in. Results demonstrated that the radiative cooling-enhanced HHE NFMs exhibited a 10% enhancement in total water uptake compared to the non-radiative cooling control group.

In outdoor experiments, the HHE NFMs were indeed enclosed within the large-scale air-to-H₂ reactor. This design choice was necessitated by the requirement to maintain a sealed environment during daytime H₂ production tests to prevent H₂ leakage. Regarding the reviewer’s concern about the **quartz glass impacting radiative cooling performance**, we clarify that the **quartz glass cover plate was intentionally opened during nighttime operations** to allow direct exposure of the HHE NFMs to ambient air for moisture adsorption. Under these conditions, **the quartz glass does not interfere**

with radiative cooling. We apologize for not explicitly detailing this operational protocol in the original manuscript. To address this omission, we have emphasized the nighttime radiative cooling exposure process in the revised manuscript, specifically stating that the HHE NFMs are exposed to the night sky to enable radiative cooling.

5 Detailed descriptions of the nighttime water adsorption procedures for the large-scale system have been added to the '*Methods: Outdoor hydrogen production measurement of large-scale HHE system*' subsection to provide the reviewer with full experimental transparency (page 17, line 4-6).

Furthermore, regarding the reviewer's query about the temperature recordings of the HHE NFMs surface in Fig. 4g, we have added detailed outdoor temperature testing methods in the revised manuscript (page 16, line 22-28) for the reviewers' reference.

10 Actually, when the HHE NFMs were placed within the reactor and isolated with a quartz glass, we were unable to capture their surface temperature using an infrared camera.

Therefore, during the temperature measurement process, we positioned the HHE NFMs in an open environment (without quartz glass covering) inside a plastic box. Due to the enclosure by the box's walls, the setup was essentially under conditions without external airflow. In this setup, our HHE NFMs film could radiate heat to the space. In contrast, when the HHE NFMs were placed inside a sealed reactor, the quartz glass would hinder heat radiation towards space due to its low transmittance in infrared spectrum.

15 Consequently, the temperature distribution in revised manuscript Fig. 4g may have underestimated the surface temperature of the PTC layer inside the sealed reactor. To address this disparity, we conducted an analysis of the radiative cooling process on the surface of the PTC layer under these two distinct conditions.

(1) Daytime radiative cooling power of PTC layer under outdoor open area condition
25 **(without quartz glass covering)**

Due to the high infrared emissivity of PTC layer, it can spontaneously radiate heat through the atmospheric window. Under the condition of zero humidity, no clouds and clean air atmosphere (no aerosol, etc.), the theoretical value of surface radiation ($P_{rad,ideal}(T_s)$) to the sky should be calculated according to the following equation:²²

$$P_{rad,ideal}(T_s) = \int_{\Omega} \int_0^{\infty} [\varepsilon_s(\lambda, \theta) I_{BB}(\lambda, T_s)] d\lambda \cos\theta d\Omega$$

where $\int_{\Omega} d\Omega = \int_0^{\pi/2} \sin\theta d\theta \int_0^{2\pi} d\phi$ is the hemisphere angular integral, and $\varepsilon_s(\lambda, \theta)$ is the spectral and angular emissivity of the PTC layer. $I_{BB}(\lambda, T_s) = \frac{2hc^2}{\lambda^5} \frac{1}{e^{\frac{hc}{\lambda k_B T_s}} - 1}$ indicates the spectral radiance of a blackbody, where h is Planck's constant, c is the speed of light in vacuum, and k_B is the Boltzmann constant. T_s and T_{amb} is the temperature of PTC layer surface exposed to the sky and ambient temperature, which is 76°C (here we take the average temperature in the open area rather than the maximum temperature because the average temperature is more representative of the surface temperature of PTC layer) and 35°C, respectively.

10 In this theoretical condition, the theoretical maximum radiation power of our PTC layer is:

$$P_{rad,ideal}(T_s) = 499.6 \text{ W/m}^2.$$

At the same time, the PTC layer also receives radiation from the environment, and its calculation method is:

$$15 \quad P_{atm}(T_{amb}) = \int_{\Omega} \varepsilon_s(\lambda, \theta) \int_0^{\infty} [\varepsilon_{atm}(\lambda, \theta) I_{BB}(\lambda, T_{amb})] d\lambda \cos\theta d\Omega$$

where $\varepsilon_{atm}(\lambda, \theta) = 1 - t(\lambda)^{1/\cos\theta}$, where $\varepsilon_{atm}(\lambda, \theta) = 1 - t(\lambda)^{1/\cos\theta}$, is the emissivity of atmosphere, $t(\lambda)$ is the transmittance of the atmosphere. Therefore, the theoretical radiation received from the environment is $P_{atm}(T_{amb}) = 136 \text{ W/m}^2$.

Therefore, the heat power taken away by radiation is:

$$20 \quad P_{rad,w/o \ window} = P_{rad,ideal}(T_s) - P_{atm}(T_{amb}) = 363.6 \text{ W/m}^2.$$

(2) Radiative cooling power of PTC layer in large outdoor reactor (**covered by quartz glass**)

Due to the quartz glass blocking radiative heat dissipation to outer space, the PTC layer can only radiate heat to the surrounding environment within the enclosed reactor. Under these conditions,

$$25 \quad P_{rad,w \ window} = \varepsilon \times \sigma \times (T_s^4 - T_{amb}^4) = 314.4 \text{ W/m}^2.$$

Indeed, $P_{rad,w/o\ window} > P_{rad,w\ window}$, which results in a slightly higher surface temperature of the HHE NFMs inside the enclosed reactor compared to those operating in open environments without quartz glass covering. However, the temperature difference between the two configurations is minimal, leading to negligible practical impact on the overall thermal behavior.

In our subsequent analysis, the primary focus was on demonstrating that temperature elevation enhances H₂ production rates. A slight underestimation of the PTC layer's surface temperature under enclosed conditions does not affect the validity of this conclusion, as the relative temperature trends remain consistent. **Consequently, the use of open-environment test results (without glass covering) for system performance analysis does not compromise the conclusion of our study.**

We appreciate the reviewer's diligence in ensuring methodological rigor and have incorporated the relevant changes into the revised manuscript.

[Comment 2]: Regarding Hydrogen Production Rate under Outdoor Natural Conditions.

The author exaggerates the superiority of the outdoor experimental results over those obtained under indoor illumination in both the abstract and main text. This comparison is unnecessary because the two conditions differ significantly in terms of operating temperatures and geometric characteristics, making them fundamentally incomparable.

[Response]: We sincerely appreciate the reviewer's meticulous review and constructive feedback. Through careful consideration of the comments, we strongly believe it is essential to enhance the original manuscript based on the reviewer's insights, particularly emphasizing the different environments of indoor and outdoor experiments, as well as the impact of varying sizes of HHE NFMs on the entire H₂ production process to avoid reader confusion. Besides, we have deleted the comparison of H₂ production rate under indoor and outdoor conditions in the abstract (revised manuscript page 1, line 29-30). In addition, we have emphasized the reasons for the increase in H₂ production rate under outdoor conditions and the comparison with indoor environments

in the revised manuscript.

Additionally, we intend to provide further explanations to elucidate the significance and criteria behind contrasting outdoor experiments with indoor experiments.

5 First of all, the comparison between indoor and outdoor experiments holds profound engineering implications. Historically, photocatalysis research has predominantly focused on catalyst modification^{52,61}, with limited attention to reactor design and system-level integration^{8,62,63}. However, transitioning laboratory-scale photocatalysis to real-world applications necessitates rigorous outdoor validation under natural sunlight. Such comparisons reveal the catalyst's true performance beyond the constrained
10 wavelength ranges of simulated solar light, thereby guiding catalyst development toward practical outdoor compatibility.⁶² This principle extends beyond solar H₂ production. Systems for solar energy conversion or atmospheric water harvesting similarly require outdoor testing to bridge the gap between laboratory efficiency metrics and field performance, ultimately informing scalable engineering solutions.

15 Secondly, our discovery that the air-to-H₂ system exhibits superior -H₂ production rates under outdoor conditions compared to controlled indoor environments represents a significant departure from prior studies. To our knowledge, existing research on photocatalytic systems consistently reports performance degradation when transitioning from indoor to outdoor conditions^{8,48,57,63}. Besides, in other large-scale
20 photocatalytic reactions, increasing the size typically results in a decrease in reaction rates when conducted under outdoor natural light.^{8,48,57} To elucidate the underlying mechanisms driving the observed performance enhancement in our air-to-H₂ system, we conducted a rigorous comparative analysis and mechanistic study. This investigation aims to advance fundamental understanding of gas-phase photothermal catalytic
25 systems and propose new performance enhancement pathways beyond traditional catalyst modification strategies. Based on our analyses, the dimension and environment of air-to-H₂ system are crucial. Conventional photocatalysis studies primarily investigate liquid-phase water splitting using particulate catalysts, where reactor scale has minimal impact on performance. In contrast, our work focuses on gas-phase

photothermal catalytic hydrogen production via HHE NFMs, which adsorb atmospheric moisture at night and generate steam for daytime photocatalytic splitting. Here, both the dimensions of HHE NFMs and their operational environment critically influence hydrogen evolution:

5 **Thermodynamic Enhancement:** Under outdoor conditions, higher surface temperature rise on HHE NFMs thermodynamically accelerates H₂ production.

Kinetic Enhancement: Larger HHE NFMs increase vapor flux through the PTC layer, elevating reactant concentrations at catalytic interfaces.

10 These findings highlight the importance of optimizing dimensions and environmental adaptability in practical applications, offering key insights for scaling air-to-H₂ systems.

Last but not least, we evaluated system performance using the widely adopted metric of area-normalized hydrogen production rate ($\mu\text{mol}\cdot\text{m}^{-2}\cdot\text{h}^{-1}$), enabling direct comparison with other photocatalytic H₂ production systems. A comprehensive benchmarking analysis against other systems has been added to the supplementary materials (Table 15 S1) to provide more detailed messages for the reviewers and readers.

We have to emphasize that we highly respect the patient reading and well intentioned reminders of the reviewers. Indeed, we should not only focus on the differences in indoor and outdoor performance, but neglect to emphasize the reasons behind this process. Therefore, we have deleted the strong contrast in the abstract. And in the 20 description of the revised manuscript, clearer explanations have been added to clarify the reasons for the differences in indoor and outdoor performance, so that readers can better understand the focus of our work (page 9, line 16 to page 10, line 7). We also put them here for your reference.

25 *‘Due to environmental variability and dimensional scaling of the HHE NFMs, the distinct H₂ production performance observed between indoor and outdoor conditions (contrary to trends reported in prior studies) demands a rigorous investigation into the underlying mechanisms.*

The enhancement of H₂ production performance under outdoor larger scale system can be attributed to synergistic thermodynamic and kinetic effects. From the

thermodynamic perspective, in the outdoor air-to-H₂ system, the HHE NFMs exhibit both higher energy input and higher surface temperature of PTC layer (Note S1 and Note S2). As shown in Fig. 4b, the temperature of the PTC layer under outdoor open area conditions can rise from 45.9°C to 71.8°C within 30 seconds (Fig. S29), reaching a maximum surface temperature of 82.2°C (average ~76 °C) (Fig. 4d), which is about 14°C higher than that of indoor conditions. As demonstrated by DFT calculations (Fig. 3h), the increased reaction temperature within the biphasic system lowers the energy barrier for H₂ formation, thereby accelerating the H₂ evolution rate. Thus, the enhanced surface temperature under outdoor conditions thermodynamically boosts hydrogen production rates compared to indoor systems. From the kinetic perspective, the enhancement of H₂ production rates in HHE NFMs originates from scaling effects. In the indoor condition, a 0.0004 m² HHE NFMs was enclosed within a glass reactor, whereas the outdoor system employed a 0.01 m² HHE NFMs array integrated into a 0.25 m² reactor. COMSOL simulations of the photothermal water desorption process for HHE NFMs of varying sizes revealed that moisture in the hygroscopic layer escapes not only from the top PTC layer but also from the sides of the hygroscopic layer (Fig. 4e and S31). However, as the HHE NFMs size increases, a greater proportion of vapor escapes through the PTC layer (Fig. 4f). This indicates that larger HHE NFMs dimensions enhance reactant availability by increasing vapor contact with catalysts on the PTC layer, thereby elevating vapor concentration in the PTC layer and kinetically enhancing hydrogen production rates.'

[Comment 3]: The author's estimation of hydrogen production based on 0.1% of the global land area lacks credible data support and has no scientific value.

[Response]: We sincerely thank the reviewer for this highly valuable suggestion and their meticulous review. We apologize for omitting the detailed estimation methodology for the global-scale application potential of the proposed air-to-H₂ system in the original manuscript. To address this, we have added a comprehensive description of the calculation process to the supplementary materials Note S3. For the reviewer's

convenience, we summarize the estimation process below:

Firstly, based on the light intensity and hydrogen production data collected from long-term outdoor testing of the air-to-H₂ system, we calculated the average H₂ production rate R_{H_2} (mL·m⁻²·h⁻¹) under different average solar radiation (P_{light}). And based on these data, as shown in Fig. R9A, a relationship curve between average solar radiation and average hydrogen production rate was fitted, and the fitting formula is:

$$R_{H_2} = 3.73 \left(\frac{P_{light}}{12} \right)^{0.53}$$

Secondly, based on the data obtained from the radiative cooling enhanced atmospheric water harvesting, we also established the relationship between RH and the water adsorption saturation (α) of HHE NFMs, as shown in Fig. R9B, and fitted a curve with the following formula:

$$\alpha = -\frac{1.00189}{1 + (RH/0.33612)^{6.25821}} + 0.99965$$

Therefore, based on the above two fitting curves, the annual average hydrogen production per square meter of land ($H_{2,year}$ /(L·m⁻²·year⁻¹)) can be obtained:

$$H_{2,year} = \alpha \frac{tR_{H_2}}{1000} \cdot \text{day}$$

Here, the direct solar irradiation all over the world can be obtained from *Global Solar Atlas*, as shown in Fig. R10 (revised supplementary materials Fig. S44). And Global nighttime average humidity distribution can be obtained from *Asia-Pacific Data-research Center, Reanalysis Data, NCEP Reanalysis*, as shown in Fig. R11 (revised supplementary materials Fig. S45).

Finally, the calculated annual average H₂ production data will be visualized on a map for reference, and further calculations will be made to determine the amount of H₂ available for the air-to-H₂ system from 0.1% of the land. Therefore, our calculations have reliable and sufficient information sources.

Our quantitative projections and system-level analyses of global air-to-H₂ system deployment not only highlight its scalability but also provide intuitive feeling of the potential of H₂ production potential by using atmospheric moisture, which is a practical prospect for the future use of the system. At the same time, it can guide us to select a

reasonable location and let us clearly understand where to build the system to obtain better benefits. Therefore, the calculation of this predicted value has strong scientific and engineering value. Besides, we have also rephrased the abstract of the manuscript and focused more on the potential of our electricity free air-to-H₂ system (page 1, line 30-32).

5

Fig. R9 A) Relationship curve between average solar radiation and average hydrogen production rate. B) The relationship between RH and the water adsorption saturation (α) of HHE NFMs.

10

Fig. R10 Direct solar irradiation all over the world. Source: World Bank Group, accessed on [05.2024], <https://globalsolaratlas.info>

Fig. R11 Global nighttime average humidity distribution. Source: Asia-Pacific Data-research Center, Reanalysis Data, NCEP Reanalysis, accessed on [05.2024], <http://apdrc.soest.hawaii.edu/las/v6/dataset?catitem=16801>

References

- 1 Yin, Y. *et al.* A Colored Temperature-Adaptive Cloak for Year-Round Building Energy Saving. *Advanced Energy Materials* (2024). <https://doi.org/10.1002/aenm.202402202>
- 2 Bhuckory, M. B. *et al.* Enhancing prosthetic vision by upgrade of a subretinal photovoltaic
5 implant in situ. *Nature communications* **16** (2025). <https://doi.org/10.1038/s41467-025-58084-y>
- 3 Jiang, E. *et al.* Organic photovoltaic mini-module providing more than 5000 V for energy autonomy of dielectric elastomer actuators. *Nature communications* **16** (2025). <https://doi.org/10.1038/s41467-025-57226-6>
- 10 4 Gao, R.-T. *et al.* Photoelectrochemical production of disinfectants from seawater. *Nature Sustainability* (2025). <https://doi.org/10.1038/s41893-025-01530-y>
- 5 Shi, X. *et al.* Unassisted photoelectrochemical water splitting exceeding 7% solar-to-hydrogen conversion efficiency using photon recycling. *Nature communications* **7** (2016). <https://doi.org/10.1038/ncomms11943>
- 15 6 Chiavazzo, E., Morciano, M., Viglino, F., Fasano, M. & Asinari, P. Passive solar high-yield seawater desalination by modular and low-cost distillation. *Nature Sustainability* **1**, 763-772 (2018). <https://doi.org/10.1038/s41893-018-0186-x>
- 7 Dong, Y. *et al.* Ceramic-carbon Janus membrane for robust solar-thermal desalination. *Nature communications* **16** (2025). <https://doi.org/10.1038/s41467-025-57888-2>
- 20 8 Nishiyama, H. *et al.* Photocatalytic solar hydrogen production from water on a 100-m² scale. *Nature* **598**, 304-307 (2021). <https://doi.org/10.1038/s41586-021-03907-3>
- 9 Goto, Y. *et al.* A Particulate Photocatalyst Water-Splitting Panel for Large-Scale Solar Hydrogen Generation. *Joule* **2**, 509-520 (2018). <https://doi.org/10.1016/j.joule.2017.12.009>
- 10 Tüysüz, H. Alkaline Water Electrolysis for Green Hydrogen Production. *Accounts of Chemical Research* (2024). <https://doi.org/10.1021/acs.accounts.3c00709>
- 25 11 Yu, M., Budiyo, E. & Tüysüz, H. Principles of Water Electrolysis and Recent Progress in Cobalt-, Nickel-, and Iron-Based Oxides for the Oxygen Evolution Reaction. *Angewandte Chemie International Edition* **61** (2021). <https://doi.org/10.1002/anie.202103824>
- 12 Qian, R. *et al.* Charge carrier trapping, recombination and transfer during TiO₂ photocatalysis: An overview. *Catalysis Today* **335**, 78-90 (2019). <https://doi.org/10.1016/j.cattod.2018.10.053>
- 30 13 Zhong, S. *et al.* Hybrid cocatalysts in semiconductor-based photocatalysis and photoelectrocatalysis. *Journal of Materials Chemistry A* **8**, 14863-14894 (2020). <https://doi.org/10.1039/D0TA04977H>
- 35 14 Linic, S., Aslam, U., Boerigter, C. & Morabito, M. Photochemical transformations on plasmonic metal nanoparticles. *Nature Materials* **14**, 567-576 (2015). <https://doi.org/10.1038/nmat4281>
- 15 Dey, A. *et al.* Hydrogen evolution with hot electrons on a plasmonic-molecular catalyst hybrid system. *Nature communications* **15**, 445 (2024). <https://doi.org/10.1038/s41467-024-44752-y>
- 16 Naldoni, A. *et al.* Pt and Au/TiO₂ photocatalysts for methanol reforming: Role of metal nanoparticles in tuning charge trapping properties and photoefficiency. *Applied Catalysis B: Environmental* **130-131**, 239-248 (2013). <https://doi.org/10.1016/j.apcatb.2012.11.006>
- 40 17 Lan, L. *et al.* Comparative study of the effect of TiO₂ support composition and Pt loading on the performance of Pt/TiO₂ photocatalysts for catalytic photoreforming of cellulose.

- International Journal of Hydrogen Energy* **46**, 31054-31066 (2021).
[https://doi.org:https://doi.org/10.1016/j.ijhydene.2021.06.043](https://doi.org/https://doi.org/10.1016/j.ijhydene.2021.06.043)
- 18 Yu, J., Muhetaer, A., Li, Q. & Xu, D. Solar Energy-Driven Reverse Water Gas Shift Reaction: Photothermal Effect, Photoelectric Activation and Selectivity Regulation. *Small* **20** (2024).
5 <https://doi.org:10.1002/sml.202402952>
- 19 Xu, J., Huang, X., Peng, J., Li, S. & Li, J.-F. Insights into plasmon-assisted chemical reactions: From fabrication to characterization. *eScience* **5**, 100312 (2025).
<https://doi.org:https://doi.org/10.1016/j.esci.2024.100312>
- 20 Park, J. Y., Lee, H., Renzas, J. R., Zhang, Y. & Somorjai, G. A. Probing Hot Electron Flow
10 Generated on Pt Nanoparticles with Au/TiO₂ Schottky Diodes during Catalytic CO Oxidation. *Nano Letters* **8**, 2388-2392 (2008). <https://doi.org:10.1021/nl8012456>
- 21 Ma, R. *et al.* Thermally-enhanced sono-photo-catalysis by defect and facet modulation of Pt-TiO₂ catalyst for high-efficient hydrogen evolution. *Ultrasonics Sonochemistry* **90**, 106222 (2022). <https://doi.org:https://doi.org/10.1016/j.ultsonch.2022.106222>
- 15 22 Raman, A. P., Anoma, M. A., Zhu, L., Rephaeli, E. & Fan, S. Passive radiative cooling below ambient air temperature under direct sunlight. *Nature* **515**, 540-544 (2014).
<https://doi.org:10.1038/nature13883>
- 23 Huang, J., Lin, C., Li, Y. & Huang, B. Effects of humidity, aerosol, and cloud on subambient radiative cooling. *International Journal of Heat and Mass Transfer* **186** (2022).
20 <https://doi.org:10.1016/j.ijheatmasstransfer.2021.122438>
- 24 Eldridge, R. G. Water vapor absorption of visible and near infrared radiation.
- 25 He, Q., Li, C., Mao, J., Lau, A. K. H. & Chu, D. A. Analysis of aerosol vertical distribution and variability in Hong Kong. *Journal of Geophysical Research: Atmospheres* **113** (2008).
<https://doi.org:10.1029/2008jd009778>
- 25 26 Mishra, A. K., Koren, I. & Rudich, Y. Effect of aerosol vertical distribution on aerosol-radiation interaction: A theoretical prospect. *Heliyon* **1**, e00036 (2015).
<https://doi.org:https://doi.org/10.1016/j.heliyon.2015.e00036>
- 27 Han, D., Ng, B. F. & Wan, M. P. Preliminary study of passive radiative cooling under Singapore's tropical climate. *Solar Energy Materials and Solar Cells* **206**, 110270 (2020).
30 <https://doi.org:https://doi.org/10.1016/j.solmat.2019.110270>
- 28 Zhao, B., Hu, M., Ao, X. & Pei, G. Performance evaluation of daytime radiative cooling under different clear sky conditions. *Applied Thermal Engineering* **155**, 660-666 (2019).
<https://doi.org:https://doi.org/10.1016/j.applthermaleng.2019.04.028>
- 29 Batishcheva, K. *Evaporation time of water droplets in an isolated chamber*. Vol. 2212 (2020).
- 35 30 Cioulachtjian, S., Launay, S., Bodaert, S. & Lallemand, M. Experimental investigation of water drop evaporation under moist air or saturated vapour conditions. *International Journal of Thermal Sciences* **49**, 859-866 (2010).
<https://doi.org:https://doi.org/10.1016/j.ijthermalsci.2009.12.014>
- 31 Xu, Z. *et al.* Ultrahigh-efficiency desalination via a thermally-localized multistage solar still. *Energy & Environmental Science* **13**, 830-839 (2020). <https://doi.org:10.1039/C9EE04122B>
- 40 32 Chen, K., Li, L. & Zhang, J. Elucidating differences in solar-driven interfacial evaporation between open and closed systems. *Desalination* **564**, 116791 (2023).
<https://doi.org:https://doi.org/10.1016/j.desal.2023.116791>
- 33 Zhang, Y., Wu, L., Wang, X., Yu, J. & Ding, B. Super hygroscopic nanofibrous membrane-based

- moisture pump for solar-driven indoor dehumidification. *Nature communications* **11**, 3302 (2020). <https://doi.org/10.1038/s41467-020-17118-3>
- 34 Cao, B., Tu, Y. & Wang, R. A Moisture-Penetrating Humidity Pump Directly Powered by One-Sun Illumination. *iScience* **15**, 502-513 (2019).
5 [https://doi.org:https://doi.org/10.1016/j.isci.2019.05.013](https://doi.org/https://doi.org/10.1016/j.isci.2019.05.013)
- 35 Canivet, J. *et al.* Structure–property relationships of water adsorption in metal–organic frameworks. *New J. Chem.* **38**, 3102-3111 (2014). [https://doi.org:10.1039/c4nj00076e](https://doi.org/10.1039/c4nj00076e)
- 36 Shan, H. *et al.* Hygroscopic salt-embedded composite materials for sorption-based atmospheric water harvesting. *Nature Reviews Materials* **9**, 699-721 (2024). [https://doi.org:10.1038/s41578-024-00721-x](https://doi.org/10.1038/s41578-024-00721-x)
- 10 37 Zhao, H., Li, Q., Wang, Z., Wu, T. & Zhang, M. Synthesis of MIL-101(Cr) and its water adsorption performance. *Microporous and Mesoporous Materials* **297**, 110044 (2020).
[https://doi.org:https://doi.org/10.1016/j.micromeso.2020.110044](https://doi.org/https://doi.org/10.1016/j.micromeso.2020.110044)
- 38 Ge, L., Feng, Y., Wu, J., Wang, R. & Ge, T. Performance evaluation of MIL-101(Cr) based
15 desiccant-coated heat exchangers for efficient dehumidification. *Energy* **289**, 130049 (2024).
[https://doi.org:https://doi.org/10.1016/j.energy.2023.130049](https://doi.org/https://doi.org/10.1016/j.energy.2023.130049)
- 39 Cai, J., Zheng, X., Pan, Q., Li, D. & Wang, W. Advances in hygroscopic metal-organic
frameworks for air, water & energy applications. *Applied Energy* **377**, 124362 (2025).
[https://doi.org:https://doi.org/10.1016/j.apenergy.2024.124362](https://doi.org/https://doi.org/10.1016/j.apenergy.2024.124362)
- 20 40 Kim, H. *et al.* Water harvesting from air with metal-organic frameworks powered by natural
sunlight. *Science* **356**, 430-434 (2017). [https://doi.org:doi:10.1126/science.aam8743](https://doi.org/doi:10.1126/science.aam8743)
- 41 Xu, W. & Yaghi, O. M. Metal-Organic Frameworks for Water Harvesting from Air, Anywhere,
Anytime. *ACS central science* **6**, 1348-1354 (2020). [https://doi.org:10.1021/acscentsci.0c00678](https://doi.org/10.1021/acscentsci.0c00678)
- 42 Furukawa, H. *et al.* Water adsorption in porous metal-organic frameworks and related materials.
25 *Journal of the American Chemical Society* **136**, 4369-4381 (2014).
[https://doi.org:10.1021/ja500330a](https://doi.org/10.1021/ja500330a)
- 43 Hanikel, N., Prevot, M. S. & Yaghi, O. M. MOF water harvesters. *Nature nanotechnology* **15**,
348-355 (2020). [https://doi.org:10.1038/s41565-020-0673-x](https://doi.org/10.1038/s41565-020-0673-x)
- 44 Lenzen, D. *et al.* Scalable Green Synthesis and Full-Scale Test of the Metal–Organic Framework
30 CAU-10-H for Use in Adsorption-Driven Chillers. *Advanced materials* **30** (2017).
[https://doi.org:10.1002/adma.201705869](https://doi.org/10.1002/adma.201705869)
- 45 Solovyeva, M. V. *et al.* Water Vapor Adsorption on CAU-10-X: Effect of Functional Groups on
Adsorption Equilibrium and Mechanisms. *Langmuir : the ACS journal of surfaces and colloids*
37, 693-702 (2021). [https://doi.org:10.1021/acs.langmuir.0c02729](https://doi.org/10.1021/acs.langmuir.0c02729)
- 35 46 Zhang, B. *et al.* Experimental study on the open adsorption performance of CAU-10-H and its
composite adsorbent. *Journal of Solid State Chemistry* **321**, 123930 (2023).
[https://doi.org:https://doi.org/10.1016/j.jssc.2023.123930](https://doi.org/https://doi.org/10.1016/j.jssc.2023.123930)
- 47 Zou, H. *et al.* Solar-driven scalable hygroscopic gel for recycling water from passive plant
transpiration and soil evaporation. *Nature Water* **2**, 663-673 (2024).
40 [https://doi.org:10.1038/s44221-024-00265-y](https://doi.org/10.1038/s44221-024-00265-y)
- 48 Pornrunroj, C. *et al.* Hybrid photothermal–photocatalyst sheets for solar-driven overall water
splitting coupled to water purification. *Nature Water* **1**, 952-960 (2023).
[https://doi.org:10.1038/s44221-023-00139-9](https://doi.org/10.1038/s44221-023-00139-9)
- 49 Suguro, T. *et al.* A hygroscopic nano-membrane coating achieves efficient vapor-fed

- photocatalytic water splitting. *Nature communications* **13**, 5698 (2022).
<https://doi.org/10.1038/s41467-022-33439-x>
- 50 Zhang, J. *et al.* Photocatalytic hydrogen production from seawater under full solar spectrum without sacrificial reagents using TiO₂ nanoparticles. *Nano Research* **15**, 2013-2022 (2021).
5 <https://doi.org/10.1007/s12274-021-3982-y>
- 51 Kato, H., Sasaki, Y., Shirakura, N. & Kudo, A. Synthesis of highly active rhodium-doped SrTiO₃ powders in Z-scheme systems for visible-light-driven photocatalytic overall water splitting. *Journal of Materials Chemistry A* **1** (2013). <https://doi.org/10.1039/c3ta12803b>
- 52 Dai, D. *et al.* g-C₃N₄/ITO/Co-BiVO₄ Z-scheme composite for solar overall water splitting.
10 *Chemical Engineering Journal* **433** (2022). <https://doi.org/10.1016/j.cej.2021.134476>
- 53 Kang, Y. *et al.* Ferroelectric polarization enabled spatially selective adsorption of redox mediators to promote Z-scheme photocatalytic overall water splitting. *Joule* **6**, 1876-1886 (2022). <https://doi.org/10.1016/j.joule.2022.06.017>
- 54 Xu, X. *et al.* Refined Z-scheme charge transfer in facet-selective BiVO₄/Au/CdS
15 heterostructure for solar overall water splitting. *International Journal of Hydrogen Energy* **46**, 8531-8538 (2021). <https://doi.org/10.1016/j.ijhydene.2020.12.047>
- 55 Zhang, Y. *et al.* Internal quantum efficiency higher than 100% achieved by combining doping and quantum effects for photocatalytic overall water splitting. *Nature Energy* **8**, 504-514 (2023).
<https://doi.org/10.1038/s41560-023-01242-7>
- 20 56 Wang, Y. *et al.* Sulfur-Deficient ZnIn₂S₄/Oxygen-Deficient WO₃ Hybrids with Carbon Layer Bridges as a Novel Photothermal/Photocatalytic Integrated System for Z-Scheme Overall Water Splitting. *Advanced Energy Materials* **11** (2021). <https://doi.org/10.1002/aenm.202102452>
- 57 Lee, W. H. *et al.* Floatable photocatalytic hydrogel nanocomposites for large-scale solar hydrogen production. *Nature nanotechnology* **18**, 754-762 (2023).
25 <https://doi.org/10.1038/s41565-023-01385-4>
- 58 Zhou, P. *et al.* Partially reduced Pd single atoms on CdS nanorods enable photocatalytic reforming of ethanol into high value-added multicarbon compound. *Chem* **7**, 1033-1049 (2021).
<https://doi.org/10.1016/j.chempr.2021.01.007>
- 59 Chen, X. *et al.* Three-dimensional porous g-C₃N₄ for highly efficient photocatalytic overall
30 water splitting. *Nano Energy* **59**, 644-650 (2019). <https://doi.org/10.1016/j.nanoen.2019.03.010>
- 60 Chai, Z., Mattsson, A., Tesfamhret, Y., Österlund, L. & Zhu, J. Ni-Ag Nanostructure-Modified Graphitic Carbon Nitride for Enhanced Performance of Solar-Driven Hydrogen Production from Ethanol. *ACS Applied Energy Materials* **3**, 10131-10138 (2020).
<https://doi.org/10.1021/acsaem.0c01838>
- 35 61 Nishioka, S., Osterloh, F. E., Wang, X., Mallouk, T. E. & Maeda, K. Photocatalytic water splitting. *Nature Reviews Methods Primers* **3**, 42 (2023). <https://doi.org/10.1038/s43586-023-00226-x>
- 62 Hisatomi, T. & Domen, K. Key Goals and Systems for Large-Scale Solar Hydrogen Production. 1331-1347 (2022). https://doi.org/10.1007/978-3-030-63713-2_43
- 40 63 Gao, M., Peh, C. K., Zhu, L., Yilmaz, G. & Ho, G. W. Photothermal Catalytic Gel Featuring Spectral and Thermal Management for Parallel Freshwater and Hydrogen Production. *Advanced Energy Materials* **10**, 2000925 (2020). <https://doi.org/10.1002/aenm.202000925>

NCOMMS-25-00566-A

Point-to point response to reviewers' comments on the manuscript

'Electricity-Free Hydrogen Production from the Air'

Reviewer #1 (Remarks to the Author):

As my comments have been successfully incorporated, the manuscript is recommended for publication.

[Response]: We sincerely thank the reviewer for the positive feedback. We are glad that our revisions have addressed the reviewer's concerns, and we are grateful for his/her endorsement to publish in Nature Communications.

Reviewer #2 (Remarks to the Author):

The authors answered all the comments correctly, so I recommend this manuscript for publication in Nature Communications.

[Response]: We thank the reviewer for the positive feedback. We are glad that our revisions have correctly answered the reviewer's comments, and we are grateful for his/her endorsement to publish in Nature Communications.

Reviewer #3 (Remarks to the Author):

Although the author has addressed my questions, the responses are clearly not very persuasive. I still have reservations about this paper.

[Response]: We sincerely appreciate the reviewer for the time and valuable feedback on our work. We explained our experimental process and results in detail, and made careful revisions based on the reviewer's comments, as specified below. We hope our response will sufficiently address the concerns.

Comment 1: Although the author added a description that the quartz glass cover plate was intentionally removed during nighttime operations to allow direct exposure of the HHE NFMs to ambient air for moisture adsorption, I would like to inquire how the frequent removal and reinstallation of the quartz glass cover in this integrated hydrogen production and water collection system maintains its seal.

The revised version in Fig. 4b shows the hydrogen production rate under typical weather conditions during the day, while Fig. 5d only presents hydrogen production results. Fig. 2 is a standalone analysis of radiation cooling water collection, but I cannot find any experimental results in the entire paper that demonstrate the continuous operation of the large-scale air-to-hydrogen system (as shown in Fig. 5a) from daytime hydrogen production to nighttime water collection. Therefore, I believe that the author's

statement about the removal of the quartz glass cover plate at night lacks credibility.

[Response]: We sincerely thank the reviewer for the careful reading and constructive comments. To clarify concerns regarding potential gas leakage during repeated reactor disassembly and reassembly, we provide a detailed description of the large-scale reactor's structural design and assembly process. The analysis results show that **our modular air-to-hydrogen reactor design enables frequent disassembly and reassembly to achieve continuous cyclic operation between nighttime radiative cooling enhanced atmospheric water harvesting and daytime photothermal vapor splitting for H₂ production, while maintaining excellent sealing integrity throughout the process.**

First of all, as shown in Fig. R1A, the large-scale air-to-H₂ system features **a modular design with a reusable rubber ring system and quick-install clamp mechanism to ensure airtightness.** The reactor consists of the following components from bottom to top: reactor base, rubber ring, transparent window, gasket, and quick-install clamp assembly. During installation, the rubber ring (with diameter approximately twice the groove depth) is first secured in the precisely machined groove of the reactor base. The quartz glass is then positioned above the gasket, followed by placement of the gasket along the glass perimeter. Finally, the clamp assembly is installed and fastened using threaded screws. The screw tightening process serves dual purposes: securing the clamp assembly while simultaneously applying compressive force to the gasket and quartz glass. **As shown in Fig. R1B, the rubber ring undergoes significant compression, forming a tight seal against the quartz glass to achieve sealing.** The entire assembly process requires approximately 3 minutes (excluding screw tightening time), as demonstrated in the supplementary installation video R1. Furthermore, rigorous gas monitoring protocols were implemented: initial gas composition measurements were taken before each experiment, followed by bihourly sampling to detect potential leaks through nitrogen (N₂) level monitoring by using gas chromatography (GC). Throughout extensive outdoor testing of the large-scale system, no gas leakage incidents were observed.

Besides, the nighttime atmospheric water harvesting process with the glass removed is illustrated in Fig. R1C. Under this condition, the reactor base is mounted on the support frame and covered with a polyethylene (PE) film, which effectively prevents interference from wind and potential precipitation while maintaining optimal radiative cooling performance. The reactor design incorporates quick-connect ports on the side panels that remain open during nighttime operation to facilitate sufficient airflow for moisture adsorption. These ports are connected to valved tubing during daytime operation to ensure reactor sealing (Fig. R1D).

Finally, in addition to presenting the nighttime radiative cooling-enhanced atmospheric water harvesting and daytime hydrogen production results in the revised manuscript, **we have conducted comprehensive 24-hour continuous water adsorption/desorption and H₂ production tests, as shown in the Fig. R1E (also included in Fig. S34 in the revised supplementary materials).** Due to technical

limitations in directly monitoring real-time mass changes of HHE NFMs within the large-scale reactor during operation, we implemented an alternative experimental approach. During nighttime water harvesting with the reactor cover open, we simultaneously conducted separate tests on individual HHE NFMs samples to characterize the complete daily water adsorption/desorption cycle, while the remaining HHE NFMs in the reactor were used for subsequent daytime H₂ production experiments. A statement on airtightness and nighttime water harvesting of large-scale system were also added to the revised manuscript page 17, line 4-15.

Fig. R1 A) 3D model of large-scale air-to-H₂ system. B) The rubber ring before and after tightening. C) Radiative cooling enhanced atmospheric water harvesting during the night. D) The reactor connected to pipes with valves. E) 24h-continuous moisture adsorption and desorption test and hydrogen production test.

Comment 2: Regarding the explanation of the HHE NFM surface temperature records in Fig. 4g, the author conducted a theoretical analysis of the radiation cooling power with and without the quartz glass in order to account for the temperature differences of the PTC surface. In the response on page 42, the first five lines confirm that there is indeed a difference between the two configurations. However, the next sentence immediately concludes that the temperature difference between the two is minimal, and its overall impact on the thermal behavior is negligible. This conclusion seems overly forced and illogical. Additionally, the revised version of the figure does not contain Fig. 4g, which further heightens the reviewer's doubts.

[Response]: We sincerely appreciate the reviewer's thoughtful comments. The Fig. 4g in the original response should be Fig.4d. In our previous analysis, we calculated the radiative heat transfer at the HHE NFMs surface under outdoor clear-sky conditions, comparing scenarios with and without the quartz glass cover plate. Due to the inability to measure the internal HHE NFMs surface temperature without compromising reactor airtightness, we assumed identical surface temperatures between the covered and uncovered configurations for radiative heat transfer calculations. The calculations yielded radiative heat transfer values of 363.6 W/m² (without glass cover) and 314.4 W/m² (with glass cover), showing a relatively small difference of 49.2 W/m².

However, when considering other heat dissipation mechanisms, particularly natural convection:

$$P_{non-radiative} = h_{non-radiative}(T_s - T_{amb}) = 328 \text{ W/m}^2,$$

where $h_{non-radiative}$ represents non-radiative heat transfer coefficient. In this setup, thermal insulation materials were applied to the sidewalls, and a windbreak layer was implemented to minimize additional heat loss caused by wind. Consequently, heat transfer in this configuration is dominated by natural convection. Therefore, calculated non-radiative heat transfer coefficient is $h_{non-radiative} = 8.0 \text{ W}/(\text{m}^2 \cdot \text{K})$. Under this condition, **the radiative heat transfer difference represents only 7% of the total heat dissipation.** This relatively small contribution supports our initial assessment that the effect of quartz glass on the overall thermal performance of HHE NFMs under daytime sunlight conditions is relatively small and therefore proposed to be negligible.

Furthermore, even if we take this effect into account, the quartz glass slightly reduces radiative cooling effect during the day, which theoretically leading to marginally higher surface temperatures in enclosed configurations. **However, this observation actually strengthens our primary finding, as the reactor-enclosed HHE NFMs would still maintain temperatures slightly above both the measured 76 °C in open outdoor conditions, and more certainly above the 58-68 °C range observed in indoor simulated sunlight tests.** Consequently, our fundamental conclusion that elevated surface temperatures under outdoor conditions enhance H₂ production rates remains fully supported.

NCOMMS-25-00566B-Z

Point-to point response to reviewers' comments on the manuscript

'Electricity-Free Hydrogen Production from the Air'

General response:

We wish to sincerely thank the two additional reviewers (Reviewer #4 and Reviewer #5) for their comments. Both reviewers explicitly mentioned that they agreed that we have satisfactorily addressed the comments from Reviewer #3 during the previous revision. Both reviewers provide additional new comments. First, we have carefully studied all these comments. We have made a major revision of our initial manuscript by accounting for the comments of the two Reviewers as much as possible and in a constructive way. We sincerely apologize for any repetition in our responses, as some of the reviewers' comments addressed overlapping aspects of the work. Please rest assured that each response has been carefully tailored to address the specific concerns raised. We greatly appreciate your kind understanding and patience in reviewing this material. We believe that this revision leads to a substantial improvement of our initial manuscript.

Below, we explain how all the comments of the two additional reviewers (Reviewer#4 and Reviewer #5) have been taken into account in our revision. In this response letter, the reviewers' comments to our original manuscript are provided in **black**, and **our point-by-point responses as well as the corresponding changes are shown in blue**.

Reviewer #4:

[General comment]:

I have carefully read the Rev #3's comments and the authors' responses, and I personally believe that the authors have well addressed the concerns raised by the Rev #3.

This is an engineering system work that contains many engineering details. The discussion between the Rev #3 and the authors on these details is conducive to the improvement of engineering experiments and the enhancement of reproducibility.

This work proposes a dual-functional composite nanofiber membrane (HHE NFMs) system that does not require electrical energy input and relies on air water vapor to directly produce hydrogen, integrating the hygroscopic and photothermal-photocatalytic synergistic hydrogen production mechanisms driven by radiation cooling. The paper is ingeniously conceived and novelly designed, with obvious originality and application potential, especially suitable for green hydrogen energy solutions in arid or water-scarce areas. The authors conducted systematic material construction, structural characterization, climate adaptation testing, and theoretical simulations, and the research content is rich.

[Response]: We sincerely thank the reviewer for providing the valuable comments! We have addressed all the comments through further experimentation and have made revisions on the manuscript, as specified below. We hope our response will satisfy you.

[Comment 1]: The experimental conditions for the Pt/TiO₂@PVDF nanofiber membranes (NFMs) and MIL-101(Cr)@PAN nanofiber membranes used in the article need further explanation. For example: Why was the Pt loading of 1 wt% chosen? Have you done any experiments to optimize the loading? What is the basis for selecting the loading and membrane thickness of MIL-101(Cr)? Is there any optimization experimental data to support this?

[Response]: We sincerely appreciate the reviewer's thorough review and valuable comments regarding the optimization of Pt loading and loading and membrane thickness of MIL-101(Cr).

Regarding the Pt loading optimization, we conducted a comprehensive literature survey prior to experimental design. While Pt deposition on TiO₂ surfaces significantly enhances photocatalytic H₂ production, this improvement does not scale infinitely with increasing Pt content¹⁻³. The literature consistently indicates an optimal Pt loading range of 0.5-2 wt%⁴⁻⁶. Accordingly, we systematically prepared and tested Pt/TiO₂ catalysts with 0.5, 1, and 2 wt% Pt loadings (denoted as 0.5Pt/TiO₂, 1Pt/TiO₂, and 2Pt/TiO₂) by adjusting the amount of H₂PtCl₆ solution during synthesis.

The H₂ production performance was evaluated using 20 mg of catalyst dispersed in 20 mL deionized water within a transparent reactor. The suspension was homogenized by ultrasonication before illumination with a 300 W xenon lamp equipped with an AM 1.5G filter, maintaining an irradiance of 1000 W·m⁻² at the solution surface. The reactor was purged with argon for 30 minutes, and gas samples were analyzed hourly by gas chromatography (GC). As shown in Fig. R1a, the H₂ production rate initially increased and then decreased with higher Pt loading, reaching an optimum at 1 wt% Pt. This trend aligns perfectly with established literature findings^{5,7}, confirming our selection of 1 wt% Pt as the optimal loading.

Concerning the MIL-101(Cr) loading in the hygroscopic layer, our design aimed to maximize water uptake capacity while maintaining structural integrity. However, excessive MIL-101(Cr) incorporation led to particle agglomeration (Fig. R1b) and non-uniform fiber formation, which would compromise both water uptake kinetics and total capacity. We therefore optimized the loading to achieve maximum MIL-101(Cr) content without sacrificing distribution homogeneity, as demonstrated in Fig. R1c. Furthermore, we optimized the hygroscopic layer thickness by fabricating HHE NFMs with varying thicknesses while maintaining identical photothermal catalytic layers. Since H₂ production rates decrease over time, we compared the initial hourly production rates across different thicknesses. As shown in Fig. R1d, the H₂ production rate first increased and then decreased with increasing hygroscopic layer thickness, peaking at approximately 11 mm. This optimal thickness was subsequently adopted for all further experiments. The above results have been included into the revised supplementary

materials (Fig. S14 and Fig. S22) and revised manuscript page 6, line 11.

Fig. R1 a) Optimization of Pt loading mass. b) SEM image of excessive MIL-101(Cr) incorporation led to particle agglomeration. c) SEM image of optimized the loading mass of MIL-101(Cr) content without sacrificing distribution homogeneity. d) Optimization of the thickness of HHE NFMs for H₂ evolution (H₂ production rate during the first hour).

[Comment 2]: It is recommended that the authors further improve the description of the experimental conditions, such as the specific definition of different weather conditions (sunny, partly cloudy, and overcast) and the frequency of recording environmental parameters, to facilitate other researchers to repeat the experiment.

[Response]: We sincerely thank the reviewer for their careful reading and constructive suggestions. We have now provided a more detailed description of the outdoor experimental conditions in the Supporting Information. During outdoor testing, we employed a portable weather station to systematically monitor ambient conditions, recording solar irradiance, temperature, and humidity at one-minute intervals.

Specifically, we characterized three distinct weather scenarios:

A) Sunny conditions (with average solar intensity is around $786.0 \text{ W}\cdot\text{m}^{-2}$): With minimal cloud coverage throughout the testing period.

B) Partially cloudy conditions (with average solar intensity around $716.5 \text{ W}\cdot\text{m}^{-2}$): Featuring intermittent cloud passage causing temporary shading of the reactor.

C) Overcast conditions (with average solar intensity around $397.6 \text{ W}\cdot\text{m}^{-2}$): With persistent cloud coverage frequently obstructing solar irradiation.

The specific definition of different weather conditions has been included in supplementary materials Nots S1.

Besides, the comprehensive time-resolved data for solar irradiance and temperature during each weather condition have been provided in revised manuscript Fig. 4b.

Additionally, we have included comprehensive time-resolved data for solar irradiance, temperature, and humidity during outdoor testing in the revised supplementary materials (Fig. R2 and revised Fig. S43). These datasets provide valuable reference for readers to understand the environmental parameters affecting system performance under real-world conditions.

Fig. R2 Operational status of the large-scale HHE system over a 14-day period. H_2

production, solar radiation intensity, H₂ production rate and STH efficiency at corresponding humidity and temperature during the 14-day period.

[Comment 3]: The data presented in the chart is relatively complete, but it is recommended to add error bars or standard deviations of repeated experiments to the graph to reflect the stability and reliability of the experimental results.

[Response]: We sincerely appreciate the reviewer's meticulous review. We would like to clarify that error bars were already included in the relevant figures of the original manuscript (as indicated in the revised manuscript Fig. 2f, and revised supplementary materials Fig. S22A and Fig. S29). Furthermore, to comprehensively demonstrate the reproducibility and stability of our HHE NFMs, we have conducted additional repeated tests and incorporated updated error bars in revised manuscript Fig. 4c and revised supplementary materials Fig. S22B, Fig. S32.

Besides, Fig. R3 also shows the new added figures in the manuscript. These enhancements provide stronger statistical validation of our experimental results.

Fig. R3 a) Comparison of average H₂ production rates between HHE NFMs of different sizes and varying reaction conditions. b) Optimization of the thickness of HHE NFMs for H₂ evolution (H₂ production rate during the first hour). c) H₂ production rate over 6h of Au/TiO₂ and Pt/TiO₂ in triphasic and biphasic system.

[Comment 4]: The author mentioned that the humidity adsorption capacity of MIL-101(Cr) varies greatly at different humidity levels (for example, it decreases significantly at 50% humidity). Can you further provide adsorption data at lower humidity conditions (such as 20%, 30%, 40%) to clarify the applicable humidity lower limit of this material?

[Response]: We sincerely thank the reviewer for their insightful comments. In revised manuscript Fig. 2j (also provided as Fig. R4a for reference), we have presented the moisture adsorption performance of the HHE NFMs. The dark-colored bars in the figure represent the water uptake capacity under conventional conditions (without radiative cooling enhancement). As correctly observed by the reviewer, the HHE NFMs exhibit limited performance at low humidity levels without radiative cooling enhancement: the adsorption capacity is $0.179 \text{ g}_{\text{water}}^{-1} \cdot \text{g}_{\text{layer}}^{-1}$ at 50% RH, merely $0.05 \text{ g}_{\text{water}}^{-1} \cdot \text{g}_{\text{layer}}^{-1}$ at 40% RH, and virtually negligible at 30% RH.

This limitation originates from the unique water adsorption mechanism of MIL-101(Cr). Although featuring exceptionally large pore sizes ($\sim 2.9 \text{ nm}$ and $\sim 3.4 \text{ nm}$) and outstanding moisture capture capacity at high humidity levels, its adsorption mechanism relies not on simple monolayer-multilayer adsorption but on cooperative adsorption. Individual water molecules exhibit weak interactions with the pore walls. The process requires initial water molecule adsorption at hydrophilic Cr sites to form nucleation centers^{8,9}. These nucleated water molecules then facilitate H₂ bonding with additional water molecules, ultimately triggering instantaneous capillary condensation that fills the entire pore cavity through a chain reaction-like process. Under low humidity conditions, insufficient water molecules are available to form stable nucleation centers, preventing the chain reaction from initiating and resulting in low adsorption capacity.

Through radiative cooling-enhanced atmospheric water harvesting, we effectively lower the surface temperature of the material during nighttime operation¹⁰. This localized cooling significantly increases the relative humidity at the HHE NFMs surface, thereby promoting the formation of nucleation centers at hydrophilic sites. Consequently, radiative cooling not only accelerates the moisture adsorption kinetics but also enhances the maximum water uptake capacity across various humidity levels, extending the effective operating range to below 30% RH. As shown in the yellow bars of Fig. R4a, the radiative cooling-enhanced HHE NFMs achieve a water adsorption capacity of $0.29 \text{ g}_{\text{water}} \cdot \text{g}_{\text{layer}}^{-1}$ even at 30% RH, with notable improvements observed

across all tested humidity levels. Unfortunately, due to equipment limitations in our laboratory, we were unable to test performance below 20% RH as we cannot reliably maintain such low humidity conditions. Additionally, we have included further experimental data on moisture adsorption under low humidity conditions without radiative cooling in the revised supplementary materials Fig. S8 (Fig. R4b and R4c) to provide comprehensive reference for the reviewers.

Fig. R4 a) The theoretical enhancement of water adsorption capacity due to the effects of radiation cooling. b) and c) Moisture adsorption kinetics of hygroscopic layer at 25 °C under various humidities.

[Comment 5]: Some of the terms used in the article are too subjective, such as "ingeniously combine" and "we are surprised to find". We suggest using more neutral terms, such as "systematically integrate" and "it is noteworthy that".

[Response]: We sincerely appreciate the reviewer's careful review and thoughtful feedback. We have revised the original manuscript accordingly, adopting more neutral language in the relevant descriptions as suggested. The modifications can be found on Page 1, line 23-24 and Page 10, line 4 of the revised manuscript.

[Comment 6]: To determine whether there are microstructural changes (such as agglomeration or detachment of MOF particles) in MIL-101(Cr)@PAN nanofibers after multiple adsorption-desorption cycles, it is recommended to supplement with TEM or SEM high-magnification characterization images after long-term cycling.

[Response]: We sincerely thank the reviewer for their thorough review and valuable suggestions. In the original manuscript, we have included SEM and elemental mapping

images of MIL-101(Cr)@PAN after long-term cycling tests in the supplementary materials Fig. S3A (as shown in Fig. R5a). After long-term outdoor testing, the honeycomb structure remained intact and did not undergo significant deformation. Additionally, as kindly suggested by the reviewer, we have provided further supplementary figures for reference. As shown in Fig. R5b (added in revised supplementary materials Fig. S3B), the TEM of MIL101(Cr)@PAN results demonstrate that MIL-101(Cr) remains firmly anchored on the nanofibers without any observed detachment or agglomeration, indicating excellent structural stability and cycling durability of the composite material. These findings further support the reliability of our proposed material system for long-term operational applications.

Fig. R5 a) The SEM and element mapping images of honeycomb-like structure after long time outdoor experiment. b) The TEM image of MIL101(Cr)@PAN.

Reviewer #5:

[General comment]: I have carefully reviewed the revisions and acknowledge that the authors have addressed Reviewer #3's concerns by supplementing structural evidence to demonstrate sealing integrity during cover removal and by providing quantitative thermal analysis showing the negligible impact of the quartz glass on the overall reactor thermal performance.

In this manuscript, the authors report on spectral selective absorbing/emitting HHE nanofiber membranes (NFMs) employed as a self-sufficient, electricity-free air-to-H₂ system. This system combines radiative cooling with advanced photothermal catalytic processes. The H₂ production rate of scale-up air-to-H₂ system under outdoor natural light reaches 6467.55 $\mu\text{mol}\cdot\text{m}^{-2}\cdot\text{h}^{-1}$. However, some analytical results are not sufficiently convincing, and the study suffers from insufficient in-depth mechanistic investigations, which hinders a comprehensive assessment of its practical application potential. Consequently, I cannot recommend this paper for publication in Nature Communications in its current form (see below the details).

[Response]: We sincerely thank the reviewer for reviewing our work and giving the valuable comments! We have addressed all the comments through further experimentation and have made revisions on the manuscript, as specified below. We hope our response will satisfy you.

[Comment 1]: As shown in Fig. 3d, the schematic diagram of PTC vapor decomposition reactions indicates H₂ and O₂ as the primary products. However, the authors have not provided experimental evidence of O₂. Additionally, the author should present evidence regarding the formation of other potential oxidation products. The ambiguity surrounding the anodic reaction pathway introduces risks of unaccounted side reactions.

[Response]: We sincerely thank the reviewer for their constructive comments. During indoor experimental testing, we monitored the gas generation throughout the HHE NFMs reaction process, with results shown in Fig. R6a. The ratio of generated H₂ to

oxygen (O_2) was consistently close to 2:1, with minor deviations attributed to minimal gas leakage during transfer and measurement procedures.

To further verify O_2 generation, we conducted experiments using $H_2^{18}O$ and analyzed the O_2 composition. Specifically, a 2 cm \times 2 cm HHE NFMs sample was first dried in a vacuum oven at 100°C for 12 hours. The sample was then rapidly transferred and sealed in a glass reactor. Simultaneously, dry nitrogen gas was bubbled through pure liquid $H_2^{18}O$, and this moisture-saturated nitrogen stream was introduced into the reactor for 6 hours to ensure complete adsorption of $H_2^{18}O$ by the hygroscopic layer of the HHE NFMs. Subsequently, the reactor was exposed to AM 1.5G simulated sunlight for 4 hours. The gaseous products were then introduced into a mass spectrometer for composition analysis. Parallel control experiments were performed using $H_2^{16}O$ under identical conditions. The results clearly demonstrated the detection of $^{18}O_2$ in the gaseous products from the $H_2^{18}O$ experiment, while no $^{18}O_2$ was detected in the $H_2^{16}O$ control experiment. Additional experiments using D_2O confirmed that the generated H_2 originated exclusively from the adsorbed water vapor. These findings provide conclusive evidence that the H_2 and O_2 produced by the HHE NFMs system derive from photocatalytic water splitting. Besides, the above results have been added into supplementary materials (Fig. S27 and S28) for the references of reviewers and readers.

Fig. R6 a) Time dependent gas production from HHE NFMs under AM1.5G simulated solar light. Mass spectrometer signals of gas generated from HHE NFMs after illumination by using b) H_2^{18}O , c) H_2^{16}O , and d) D_2O .

[Comment 2]: Can the authors provide quantitative evidence (e.g. interfacial adsorption energies, charge-transfer measurements) to enhance readers' understanding of why the biphasic system is better than the triphasic system?

[Response]: We sincerely appreciate the reviewer's valuable suggestion. We have calculated the electronic properties including charge density difference (CDD) and projected density of states (PDOS) to distinguish biphasic system and triphasic system. As shown in Fig. R7a, b and c, Pt cluster of biphasic system accept less electron from TiO_2 (101) substrate, leading to upshift of p orbital and d orbital center compared to that of triphasic system. As a results, the adsorption strength, and electron transfer of water molecule and hydrogen atom of biphasic system both are larger than those of triphasic system under the same temperature (298 K) (See Fig. R7d, e, f and g). The above results have been provided in the revised Supplementary Materials (Figure S33 and S34).

Fig. R7 The charge density difference and the corresponding electron transfer analyzed by Bader charge for a) biphasic system and b) triphasic system (The silver, blue and red balls indicate Pt, Ti and O, respectively. The yellow and blue colors represent charge accumulation and depletion. The isosurface values are set at $\pm 5.0 \times 10^{-3} \text{ e}/\text{\AA}^3$). c) PDOS for Pt/TiO₂ of biphasic system and triphasic. The charge density difference and the corresponding electron transfer analyzed by Bader charge for d) adsorbed H₂O* of biphasic system and e) adsorbed H₂O* of triphasic system, f) adsorbed H* of triphasic system and g) adsorbed H* of triphasic system (The silver, blue, red and white balls indicate Pt, Ti, O and H, respectively. The yellow and blue colors represent charge accumulation and depletion. The isosurface values are set at $\pm 5.0 \times 10^{-3} \text{ e}/\text{\AA}^3$).

[Comment 3]: It is better to compare this system with other recently reported photocatalytic air-to-H₂ systems in the literature, including parameters such as STH and quantum efficiency.

[Response]: We sincerely appreciate the reviewer's valuable suggestion. We have incorporated a comparative analysis between the proposed Air-to-H₂ system and other photocatalytic Air-to-H₂ systems in the revised manuscript (as shown in Table R1). Given the limited number of studies focusing specifically on photocatalytic air-to- H₂ systems, we have further expanded the comparison to include representative vapor-fed and moisture evaporated from pure water photocatalytic H₂ production systems. The comprehensive comparison, including key performance metrics and operational parameters, has been provided in the revised Supplementary Materials (Table S2).

Table R1 The comparisons of photocatalytic air-to-H₂ system.

Photocatalyst	Feedstock	STH efficiency	quantum efficiency	System scale	Light source	Ref.
Pt/TiO ₂	Moisture harvested from the air	0.10%	NA.	2.5 × 10⁻¹ m²	Natural sunlight	This work
TiO _x -coated	Moisture	0.08%	0.11%	1.075 × 10 ⁻³ m ²	Natural	¹¹

CoOOH/Cr ₂ O ₃ /Rh/SrTiO ₃ :Al	harvested from the air				sunlight and UV LED (380 nm) 300W Xe lamp (100 mW cm ⁻²)	11
TiO _x -coated CoOOH/Cr ₂ O ₃ /Rh/SrTiO ₃ :Al	Moisture harvested from the air	0.12%	0.11%	1.075×10 ⁻³ m ²	and UV LED (380 nm)	
Pt-TiO ₂	Moisture harvested from the air (with polyethylene glycol as sacrificial agent)	0.0859 %	NA.	2.83×10 ⁻³ m ²	Natural sunlight	12
Pt-TiO ₂	Moisture harvested from the air (with ethylene glycol as sacrificial agent)	0.14%	NA.	5×10 ⁻² m ²	Simulated sunlight (100 mW cm ⁻²)	13
SrTiO ₃ :Al-RhCrO _x	Moisture harvested from the air	NA.	0.34%	1.35×10 ⁻⁴ m ²	UV LED (365 nm,	14

CoO _y					14.27mW cm ² .)	
RhCrO _x - Al:SrTiO ₃	Moisture evaporated from pure liquid water	0.09%	NA.	2.5×10 ⁻³ m ²	Natural sunlight	15
Py- HMPA@Pt	Moisture evaporated from pure water	0.064%	7.9%	3.14×10 ⁻⁴ m ²	300 W Xenon lamp with filter (λ > 420 n m, 100 mW c m ²), UV LED (420 nm)	16
CoOOH/Rh loaded SrTiO ₃ :Al coated with TiO _x	Pure vapor feeding	0.4%	NA.	4×10 ⁻⁴ m ²	AM 1.5G simulated sunlight (~100 mW cm ²)	17
Pt-CN	Pure vapor feeding (flow rate 2 mL min ⁻¹)	0.26%	NA	4×10 ⁻⁴ m ²	Simulated sunlight (100 mW cm ⁻²)	18

[Comment 4]: Specially, regarding nighttime operation: does the system store sufficient water to sustain daytime hydrogen production?

[Response]: We sincerely appreciate the reviewer's thorough review and insightful

comments. Due to the good moisture harvesting capability of the HHE NFMs under nighttime conditions, the material can absorb sufficient amounts of water vapor even in low-humidity environments. The radiative cooling effect at the surface of the HHE NFMs further enhances this process by effectively reducing the local temperature and increasing the relative humidity nearby, thereby significantly improving the water uptake performance of the hygroscopic layer.

Based on our daytime tests, the amount of water adsorbed during the night is adequate to support H₂ production throughout the following day. Under cloudy or partially cloudy conditions, the system maintains continuous H₂ production over the entire daytime period. Even under clear-sky conditions, the stored moisture enables at least 6 hours of efficient H₂ production under effective illumination. To validate the stability of the material after long-term outdoor exposure as well as the reliability of the adsorption–desorption process, we have provided detailed data on both H₂ production and moisture changes of the HHE NFMs during extended outdoor testing in the Supporting Information. The results confirm that the stored water is gradually released over 6 hours of illumination, ensuring consistent H₂ production during the daytime (As shown in Fig. R8 and revised supplementary materials Fig. S44).

Fig. R8 24h-continuous moisture adsorption and desorption test and H₂ production test after long time outdoor experiment, with the humidity, temperature, and solar radiation

intensity.

[Comment 5]: Please supplement data on the hygroscopic and hydrogen production performance under varying humidity levels (30%–90% RH).

[Response]: We sincerely thank the reviewers for their valuable comments. We have supplemented the manuscript with comprehensive data on both hygroscopic performance and H₂ production efficiency across varying humidity levels (30%–90% RH), as now presented in Figure R9. These datasets, demonstrating the system's performance under different relative humidity conditions, have also been included in the Supporting Information to ensure full transparency and reproducibility. The above results have been provided in the revised Supplementary Materials (Fig. S8 and S25).

Fig. R9 a) and b) Moisture adsorption kinetics of hygroscopic layer at 25 °C under various humidities. c) H₂ production rate of HHE NFMs (0.0004 m²) under indoor simulated sunlight. d) Average H₂ production rate of 4h sunlight illumination.

[Comment 6]: Please conduct supplementary cyclic tests to evaluate the hygroscopic and hydrogen production stability of the composite system.

[Response]: We sincerely appreciate the reviewer's valuable suggestions. We have conducted long-term outdoor testing to evaluate the cycling stability of the large-scale air-to-H₂ system, as shown in revised supplementary materials Fig. S33. We also put the result in Fig. R10a for your references. Besides, the cycling stability of the hygroscopic layer has also been thoroughly examined, and the results are provided in Fig. R10b (included in the revised Supplementary materials Fig. S9). Additionally, we performed multiple cycling experiments under indoor conditions using 2 cm × 2 cm HHE NFMs (Fig. R10c and revised supplementary materials Fig. S31), which further confirm the stability of the integrated system.

Fig. R10 a) Operational status of the large-scale HHE system over a 14-day period. H₂ production, solar radiation intensity, H₂ production rate and STH efficiency at corresponding humidity and temperature during the 14-day period. b) Cycling stability of moisture adsorption–desorption of MIL@PAN and pure PAN NFMs at 25 °C and 90% RH (desorption at 100 °C). c) H₂ production cycling stability of HHE NFMs.

[Comment 7]: While radiative cooling is claimed to enhance hygroscopicity, the moisture saturation time is only 1 hour. How is this purported advantage substantiated?

[Response]: We sincerely thank the reviewer for raising this important question. The radiative cooling effect on the surface of the HHE NFMs effectively reduces the local temperature of the hygroscopic layer. This decrease in temperature significantly increases the local relative humidity around the HHE NFMs, thereby enhancing their moisture capture performance. As a result, radiative cooling contributes to two key improvements: **a) It accelerates the moisture adsorption rate of the HHE NFMs, and b) It increases the maximum water uptake capacity of the material.** In outdoor tests comparing HHE NFMs with and without the PTC layer, we observed that the samples with the PTC layer exhibited **faster moisture adsorption kinetics** (Fig. R11a) and achieved nearly **10% higher maximum water uptake capacity** (Fig. R11a and R11b).

Besides, following established methodologies reported in the literature^{10,19}, we replicated radiative cooling-like conditions under controlled laboratory settings to systematically evaluate the hygroscopic performance of HHE NFMs under idealized radiative cooling-enhanced scenarios. The results demonstrate that radiative cooling extends the effective operating range of the HHE NFMs and increases the maximum water uptake capacity under all RH (Fig. R11c). The minimum required RH is reduced from 40% RH to 30 % RH. This improvement significantly expands the potential application range of the air-to- H₂ system and enhances the maximum water supply available for H₂ production.

Fig. R11 a) Water uptake capacity comparison of HHE NFMs with/without radiative cooling effect as a function of time during the outdoor test. b) The maximum water uptake comparison of HHE NFMs with/without radiative cooling during the outdoor test. c) The theoretical enhancement of water adsorption capacity due to the effects of radiation cooling at an ambient temperature of 25°C during the indoor test.

[Comment 8]: Please explicitly state the functional contribution of each component within the main text.

[Response]: We sincerely appreciate the reviewer’s valuable suggestion. In the original manuscript (page 2, line 30 – page 3, line 13), we provided a detailed introduction to each component of the HHE NFMs and their respective functions. To present this information more clearly and avoid potential misunderstandings, we have carefully refined and elaborated the description in this section. The revised content is provided below and has been explicitly marked in the revised manuscript for easy reference by both the reviewers and readers.

‘This study introduces an innovative air-to-hydrogen system that employs multilayer hygroscopic hydrogen evolution (HHE) nanofiber membranes (NFMs) for efficient green H₂ production, with clean water generated as a byproduct (Fig. 1a).

Fabricated via electrospinning, the HHE NFMs consist of a photothermal catalytic

(PTC) top layer of Pt/TiO₂@PVDF and a porous hygroscopic bottom layer of MIL-101(Cr)@PAN (Fig. 1b). Capitalizing on the high solar absorptivity and high infrared emissivity of the PTC layer (Fig. 1c), the system achieves radiative cooling-enhanced atmospheric water harvesting (AWH) at night, and photocatalytic-driven H₂ production during the day by decomposing the water vapor released from the hygroscopic layer. Specifically, the distribution of MIL-101(Cr) within the highly porous hygroscopic layer facilitates rapid, spontaneous moisture adsorption from ambient air at night while simultaneously mitigating particle aggregation. Concurrently, the radiative cooling effect of the PTC layer elevates the local humidity, thereby accelerating moisture capture and extending the effective hygroscopic range (Fig. 1d). Under daytime solar irradiation, visible and near-infrared (NIR) wavelengths are converted into thermal energy to desorb water vapor from the hygroscopic layer, while ultraviolet (UV) light is absorbed to activate Pt/TiO₂ for the photocatalytic (PC) production of H₂.

[Comment 9]: The original DFT calculations used energy barrier data at 100°C. However, since the actual maximum operational temperature did not reach 100°C, this data is unreliable. Please provide supplementary DFT calculations at the actual operational temperatures to demonstrate the trend of decreasing reaction energy barriers with increasing temperature.

[Response]: We sincerely thank the reviewer for their highly valuable suggestion. We calculated the Gibbs energy of a photocatalytic reaction in the biphasic system over the Pt/TiO₂ surface at various temperature (including our reaction temperature around 60 °C under indoor conditions and around 80 °C under outdoor conditions) and reaction energy barriers of potential determining step (PDS) for the biphasic system at various temperature. As shown in Fig. R12 (b), the reaction energy barriers of PDS generally exhibit a linear relationship with temperature. These results have been included in the revised supplementary materials (Fig. S35) for the reference of reviewers and readers.

Fig. R12 a) Calculated Gibbs energy of a photocatalytic reaction in the biphasic system over the Pt/TiO₂ surface at various temperature. b) Reaction energy barriers of potential determining step (PDS) for the biphasic system at various temperature.

[Comment 10]: Please assess the solar energy utilization efficiency of the system.

[Response]: We sincerely thank the reviewer for their highly valuable suggestion. The solar-to-hydrogen (STH) conversion efficiency of the large-scale outdoor air-to-H₂ system has been calculated. Additionally, we have now specifically highlighted and annotated the solar energy utilization efficiency in the relevant figures as indicated in Fig. R13 (also provided in supplementary materials Fig. S43). For the solar efficiency calculations, we assumed that after each 6-hour outdoor test, the moisture within the HHE NFMs was completely released. This assumption formed the basis for determining the solar energy utilized during the water evaporation process. Therefore, the calculation of the power utilized by water evaporation ($P_{\text{evaporation}}$) can be calculated as:

$$P_{\text{evaporation}} = (h_{\text{evaporation}} + C_{p,w} \Delta T) \dot{m}$$

where, $h_{\text{evaporation}}$ is water vapor desorption isosteric heats of MIL-101(Cr), which is taken from the experimental results of Yan et.al (53 kJ/mol), $C_{p,w}$ is heat capacity of water ($4.182 \times 10^3 \text{ J} \cdot \text{kg}^{-1} \cdot \text{K}^{-1}$), ΔT is the temperature increased of HHE NFMs (here, we use the average temperature rise on the surface of HHE NFMs as the standard), and \dot{m} is the desorption rate of water vapor. $h_{\text{evaporation}}$ is taken from the experimental results of Yan et.al (53 kJ/mol). Therefore, the solar to water vapor efficiency (STW) could be

calculated by:

$$STW = \frac{\text{Energy output as H}_2\text{O}}{\text{Incident light energy}} = \frac{P_{\text{evaporation}}}{P_{\text{light}}}$$

Therefore, the solar utilization efficiency (EFF_{sun}) is calculated by:

$$EFF_{\text{sun}} = \text{STH} + \text{STW}$$

Fig. R13 Operational status of the large-scale HHE system over a 14-day period. H_2 production, solar radiation intensity, Average H_2 production rate, STH efficiency and Solar utilization efficiency at corresponding humidity and temperature during the 14-day period.

[Comment 11]: Please compare the hydrogen production performance between the two-phase system and the system after in-situ moisture absorption, rather than the three-phase system under conditions of abundant water supply.

[Response]: We sincerely thank the reviewer for their constructive suggestion. In the original manuscript, we compared the H_2 production performance of catalysts in liquid-

phase environments primarily to highlight the advantages of our air-to- H₂ system over traditional liquid-phase catalytic systems. For instance, our system eliminates the need for purified liquid water input, as it autonomously harvests moisture from air to provide a virtually unlimited clean water source. Additionally, our system demonstrates significantly higher H₂ production rates compared to conventional triphasic catalytic systems.

The reviewer's suggestion to include a biphasic system for comparison is highly valuable, as it would provide clearer insights into the catalyst's performance under biphasic conditions. Although it is challenging to maintain a continuous vapor feeding in our experimental setup, we designed an experimental condition simulating biphasic reactions based on methods reported in the literature^{11,16} (schematic and actual setup are shown in Fig. R14a). Specifically, 20 mg of Pt/TiO₂ was dispersed in 1 mL of ethanol solution and sonicated for 30 minutes. The mixture was then drop-cast onto a 4 cm² frosted glass plate placed on a hotplate at 60°C. After solvent evaporation, the Pt/TiO₂ catalyst remained deposited on the glass plate, which was subsequently dried overnight in an oven at 80°C. The catalyst-coated glass plate was then placed in a quartz reactor, with 20 mL of DI water positioned beneath the sample holder to serve as a vapor source. This setup maintained a relative humidity above 80% RH inside the reactor (as confirmed in Fig. R14b). The reactor was purged with argon for 30 minutes to remove air prior to irradiation under AM 1.5G simulated sunlight (100 mW/cm²) for H₂ production test. The results show that the H₂ production rate of biphasic system with vapor feeding is about 2543.2 μmol·m⁻²·h⁻¹ during the first hour of reaction, which is higher than HHE NFMs biphasic system (2035.2 μmol·m⁻²·h⁻¹), as shown in Fig. R14c. However, both of these two biphasic systems show four 4-4.5 times higher than that of the triphasic system. However, the H₂ production rate of HHE NFMs biphasic system decreases because of the gradual release of water vapor from hygroscopic layer. And the H₂ production rate of biphasic system with vapor feeding keeps nearly constant H₂ production rate (Fig. R14d). Therefore, after 4h of sunlight illumination, the average H₂ production rate of biphasic system with vapor feeding is about twice that of HHE NFMs

biphasic system under indoor condition. This enhancement can be attributed to the continuous vapor supply in the biphasic system with vapor feeding, which ensures stable H₂ production performance, further validating the advantages of two-phase systems over three-phase systems. These findings have been included in the revised supporting materials (Fig. S23 and 24) and revised manuscript page 7, line 15-29 for the reference of reviewers and readers.

Fig. R14 a) Schematic and actual setup of biphasic system with vapor feeding. b) The record of RH in the biphasic photoreactor. c) H₂ production rate comparison of triphasic and biphasic reaction system in the first hour. d) H₂ production comparison of HHE NFMs biphasic system and biphasic system with vapor feeding.

[Comment 12]: The mechanism of thermal-photocatalytic synergy remains ambiguous. It is not elucidated how infrared photothermal effects facilitate the separation/transfer of photogenerated charges.

[Response]: We sincerely thank the reviewer for their careful reading and constructive

comments. We would like to clarify and emphasize that the photothermal catalytic process described in this work refers to a synergistic process that combines photothermal and photocatalytic mechanisms for enhanced solar energy utilization. The **ultraviolet (UV) portion of the spectrum is primarily responsible for exciting the TiO₂ catalyst to generate electron-hole pairs** that drive the redox reaction, **with Pt acting as a cocatalyst to facilitate charge separation**⁶. Meanwhile, **visible and near-infrared (NIR) light, along with energy from the relaxation of electron-hole pairs generated above the bandgap energy, is converted into heat**. This thermal energy drives the release of water vapor from the hygroscopic layer, thereby supplying the reactant for the photocatalytic H₂ evolution reaction.

To validate this mechanism, we conducted controlled experiments. When the HHE NFMs were irradiated with only visible and NIR light, the release of steam was observed as condensation on the reactor walls, but no H₂ was detected (Fig. R15a and b). This confirms that photons in this energy range are insufficient to excite charge carriers in TiO₂ for redox reactions. Conversely, under UV irradiation alone, a measurable amount of H₂ was produced, though the yield was significantly lower than that under full-spectrum illumination. This demonstrates that UV light is essential for generating electron-hole pairs in TiO₂, while visible-NIR light contributes mainly through photothermal effects. Notably, full-spectrum irradiation substantially enhances the H₂ production performance of the air-to- H₂ system. In the Pt/TiO₂ catalytic system, Pt serves as an effective electron acceptor, promoting charge separation and reducing electron-hole recombination. The temperature increase induced by photothermal effects further **accelerates electron transport between Pt and TiO₂ and within Pt itself**, thereby **reducing the recombination** of photogenerated carriers and improving the overall photocatalytic H₂ evolution efficiency²⁰⁻²². Additionally, DFT calculations indicate that in the biphasic air-to-H₂ system, higher reaction temperatures lower the energy barrier for H₂ generation, thermodynamically favoring the reaction. Elevated temperatures also reduce H₂ transport resistance, facilitating desorption from active sites. These combined effects synergistically enhance the H₂ production efficiency of

the photocatalytic system.

In response to the reviewer's suggestion, we have not only elaborated on the promotional role of the photothermal process in the photocatalytic reaction in the revised manuscript but also added further explanation regarding its positive impact on charge carrier migration (page10, line 5 to line 31).

Fig. R15 a) H_2 production rate under different light conditions during the first hour. b) Photo of H_2 production test under Vis-NIR light condition with condensed water on the wall of reactor.

References

- 1 Matsubara, K., Inoue, M., Hagiwara, H. & Abe, T. Photocatalytic water splitting over Pt-loaded TiO₂ (Pt/TiO₂) catalysts prepared by the polygonal barrel-sputtering method. *Applied Catalysis B: Environmental* **254**, 7-14 (2019). <https://doi.org/10.1016/j.apcatb.2019.04.075>
- 2 Han, X. *et al.* Effect of loading of Pt at different positions on the photocatalytic hydrogen production performance of TiO₂ hollow spheres. *Separation and Purification Technology* **361**, 131586 (2025). <https://doi.org/10.1016/j.seppur.2025.131586>
- 3 Al-Madanat, O., AlSalka, Y., Dillert, R. & Bahnemann, D. W. Photocatalytic H₂ Production from Naphthalene by Various TiO₂ Photocatalysts: Impact of Pt Loading and Formation of Intermediates. *Catalysts* **11** (2021).
- 4 Jovic, V. *et al.* Photocatalytic H₂ Production from Ethanol–Water Mixtures Over Pt/TiO₂ and Au/TiO₂ Photocatalysts: A Comparative Study. *Topics in Catalysis* **56**, 1139-1151 (2013). <https://doi.org/10.1007/s11244-013-0080-8>
- 5 Bamwenda, G. R., Tsubota, S., Nakamura, T. & Haruta, M. Photoassisted hydrogen production from a water-ethanol solution: a comparison of activities of Au□TiO₂ and Pt□TiO₂. *Journal of Photochemistry and Photobiology A: Chemistry* **89**, 177-189 (1995). [https://doi.org/10.1016/1010-6030\(95\)04039-1](https://doi.org/10.1016/1010-6030(95)04039-1)
- 6 Sakthivel, S. *et al.* Enhancement of photocatalytic activity by metal deposition: characterisation and photonic efficiency of Pt, Au and Pd deposited on TiO₂ catalyst. *Water Research* **38**, 3001-3008 (2004). <https://doi.org/10.1016/j.watres.2004.04.046>
- 7 Jiang, Z. *et al.* Impact of Methanol Photomediated Surface Defects on Photocatalytic H₂ Production Over Pt/TiO₂. *ENERGY & ENVIRONMENTAL MATERIALS* **3**, 202-208 (2020). <https://doi.org/10.1002/eem2.12068>
- 8 Zhang, B., Zhu, Z., Wang, X., Liu, X. & Kapteijn, F. Water Adsorption in MOFs: Structures and Applications. *Advanced Functional Materials* **34**, 2304788 (2024). <https://doi.org/10.1002/adfm.202304788>
- 9 Shaharudin, M. R. b., Williams, C. D., Achari, A., Nair, R. R. & Carbone, P. Decoding the Interplay between Topology and Surface Charge in Graphene Oxide Membranes During Humidity Induced Swelling. *ACS nano* **17**, 21923-21934 (2023). <https://doi.org/10.1021/acsnano.3c08260>
- 10 Zhu, W. *et al.* Radiative cooling sorbent towards all weather ambient water harvesting. *Communications Engineering* **2**, 35 (2023). <https://doi.org/10.1038/s44172-023-00082-3>
- 11 Yang, W. *et al.* Synergistic Integration of Atmospheric Water Harvesting and Solar-Driven Hydrogen Production via Multifunctional Hygroscopic-Photocatalytic Hydrogel Nanocomposite. *Advanced Functional Materials* (2025). <https://doi.org/10.1002/adfm.202512738>
- 12 Huang, L. *et al.* Solar-driven hydrogen production based on moisture adsorption-desorption cycle. *Nano Energy* **128** (2024). <https://doi.org/10.1016/j.nanoen.2024.109879>
- 13 Liu, P. *et al.* Scalable hydrogen production by harvesting moisture from the air under natural sunlight. *Chemical Engineering Journal* **510** (2025). <https://doi.org/10.1016/j.cej.2025.161832>
- 14 Shearer, C. J., Hisatomi, T., Domen, K. & Metha, G. F. Gas phase photocatalytic water splitting of moisture in ambient air: Toward reagent-free hydrogen production. *Journal of Photochemistry and Photobiology A: Chemistry* **401**, 112757 (2020).

- <https://doi.org/10.1016/j.jphotochem.2020.112757>
- 15 Pornrunroj, C. *et al.* Hybrid photothermal–photocatalyst sheets for solar-driven overall water splitting coupled to water purification. *Nature Water* **1**, 952-960 (2023). <https://doi.org/10.1038/s44221-023-00139-9>
- 16 Liu, Y. *et al.* One-dimensional covalent organic frameworks with atmospheric water harvesting for photocatalytic hydrogen evolution from water vapor. *Applied Catalysis B: Environmental* **338** (2023). <https://doi.org/10.1016/j.apcatb.2023.123074>
- 17 Suguro, T. *et al.* A hygroscopic nano-membrane coating achieves efficient vapor-fed photocatalytic water splitting. *Nature communications* **13**, 5698 (2022). <https://doi.org/10.1038/s41467-022-33439-x>
- 18 He, L. *et al.* A Hybrid Photocatalytic System Splits Atmospheric Water to Produce Hydrogen. *Advanced Functional Materials* **34** (2024). <https://doi.org/10.1002/adfm.202313058>
- 19 Wang, Y. *et al.* Heterogeneous wettability and radiative cooling for efficient deliquescent sorbents-based atmospheric water harvesting. *Cell Reports Physical Science* **3** (2022). <https://doi.org/10.1016/j.xcrp.2022.100879>
- 20 Li, J. *et al.* Efficient photothermal catalytic hydrogen production via plasma-induced photothermal effect of Cu/TiO₂ nanoparticles. *International Journal of Hydrogen Energy* **48**, 6336-6345 (2023). [https://doi.org:https://doi.org/10.1016/j.ijhydene.2022.05.027](https://doi.org/https://doi.org/10.1016/j.ijhydene.2022.05.027)
- 21 Wei, H. *et al.* Spontaneous Chirality Transition-Mediated Piezo-Pyroelectric Synergy for Photocatalytic Hydrogen Production from Saturated Vapor of VOC Aqueous Solutions. *Advanced Functional Materials* **n/a**, e18394 (2025). <https://doi.org:https://doi.org/10.1002/adfm.202518394>
- 22 Qu, J. *et al.* Direct Thermal Enhancement of Hydrogen Evolution Reaction of On-Chip Monolayer MoS₂. *ACS nano* **16**, 2921-2927 (2022). <https://doi.org/10.1021/acsnano.1c10030>